# The synthetic lethal interaction between *CDS1* and *CDS2* is a vulnerability in uveal melanoma and across multiple tumor types

Pui Ying Chan[1,18], Diana Alexander [1,18], Ishan Mehta [1,18],
Larissa Satiko Alcantara Sekimoto Matsuyama [1,18], Victoria Harle[1],
Rebeca Olvera-León[1], Jun Sung Park [1], Fernanda G. Arriaga-González[1],
Louise van der Weyden [1], Saamin Cheema [1], Vivek Iyer[1], Victoria Offord[1],
David Barneda[2], Phillip T. Hawkins [2], Len Stephens[2], Zuza Kozik[3],
Michael Woods[4,5], Kim Wong[1], Gabriel Balmus [1,4,5,6], Alessandro Vinceti[7],
Nicola A. Thompson[1], Martin Del Castillo Velasco-Herrera [1],
Lodewyk Wessels [8,9], Joris van de Haar[8,9,10], Emanuel Gonçalves [11,12],
Sanju Sinha[13], Martha Estefania Vázquez-Cruz[14], Luisa Bisceglia[15],
Francesco Raimondi [16], Jyoti Choudhary [3], Sumeet Patiyal[13],
Anjan Venkatesh[16], Francesco Iorio [7], Colm J. Ryan [16,17] & David J. Adams [1]✉

Metastatic uveal melanoma is an aggressive disease with limited effective therapeutic options. To comprehensively map monogenic and digenic dependencies, we performed CRISPR–Cas9 screening in ten extensively profiled human uveal melanoma cell line models. Analysis involved genome-wide single-gene and combinatorial paired-gene CRISPR libraries. Among our 76 uveal melanoma-specific essential genes and 105 synthetic lethal gene pairs, we identified and validated the CDP-diacylglycerol synthase 2 gene (*CDS2*) as a genetic dependency in the context of low CDP-diacylglycerol synthase 1 gene (*CDS1*) expression. We further demonstrate that *CDS1/CDS2* forms a synthetic lethal interaction in vivo and reveal that *CDS2* knockout results in the disruption of phosphoinositide synthesis and increased cellular apoptosis and that re-expression of *CDS1* rescues this cell fitness defect. We extend our analysis using pan-cancer data, confirming increased *CDS2* essentiality in diverse tumor types with low *CDS1* expression. Thus, the CDS1/CDS2 axis is a therapeutic target across a range of cancers.

While the clinical benefit of immune checkpoint inhibition has revolutionized the treatment of cutaneous melanoma, survival outcomes for patients with metastatic uveal melanoma remain exceptionally poor[1,2]. Tebentafusp, the first approved systemic therapy, modestly improves survival but is limited to patients with uveal melanoma who carry the human leukocyte antigen (HLA)-A*02:01 allele[3,4]. Although our molecular understanding of uveal melanoma has improved, targeted therapies have shown limited antitumor responses[5–10]. Thus, new

therapeutic targets are urgently needed, and adopting a systematic approach is crucial to comprehensively identify new tumor-intrinsic vulnerabilities.

CRISPR–Cas9 screening of cancer cell lines using single-guide RNAs (sgRNAs) has enabled the high-throughput perturbation of individual genes on a genome-wide scale, leading to the discovery of cell-essential genes[11,12]. Synthetic lethal interactions, where combined perturbation of two genes results in a significant growth/fitness defect,

have traditionally been identified by performing large loss-of-function screens in cell lines that harbor defined genetic changes. A valuable source of synthetic lethal interactions are paralog gene pairs, which are derived from a common ancestral gene and often function in common or parallel pathways. Notably, shared functions between paralog gene pairs may enable cells to withstand the loss of one member of the pair[13]. Indeed, genes with identifiable paralogs are less likely to be essential compared with genes with no identifiable paralog[14–18]. Moreover, in situations where both genes are expressed, buffering between paralogs may prevent genes from being identified as essential in single-gene knockout studies[19]. Combinatorial CRISPR–Cas9 approaches have been developed to disrupt two genes simultaneously, and screens employing this technology have facilitated the identification of new synthetic lethal interactions between paralog pairs[19–23], highlighting them as potential therapeutic targets.

In this study, we perform a comprehensive CRISPR–Cas9 knockout screen in uveal melanoma, analyzing ten cell line models that we also subject to extensive profiling, including whole-genome and transcriptome sequencing and proteomic profiling. Using combinatorial paired guide RNA (pgRNA) and genome-wide sgRNA libraries, we map gene-pair and single-gene dependencies, respectively. Our analysis reveals a previously uncharacterized interaction between the paralogous CDP-diacylglycerol synthase 1 and 2 (*CDS1* and *CDS2*) genes, alongside previously described pairs. We show that low *CDS1* expression identifies a subset of tumors that are selectively sensitive to *CDS2* loss, both in uveal melanoma and other cancer types. We also show mechanistically that disruption of the CDS1/CDS2 axis causes precursor phosphatidic acid (PA) accumulation, lipid droplet accumulation, and decreased levels of the phosphoinositides phosphatidylinositol (PI) and phosphatidylinositol monophosphate (PIP), resulting from disruption of phosphoinositide synthesis. Thus, these findings reveal a potential therapeutic target for uveal melanoma and other malignancies.

## Results

### Genomic and proteomic analysis of uveal melanoma models

Few uveal melanoma models exist and none have been comprehensively characterized. We therefore collected and validated a panel of ten human uveal melanoma cell lines, which were subjected to whole-genome and deep-transcriptome sequencing. These models were also proteome profiled, with these data collectively generating rich catalogs of single-nucleotide variants (SNVs) and copy number/structural variant calls alongside RNA/protein expression profiles (Supplementary Figs. 1–3 and Supplementary Tables 1–3). Notably, our collection of models includes established driver events, and using both CELLector[24] and Celligner[25] and with the genome and transcriptome data of each cell line, respectively, we show these models align tightly with uveal melanoma data in The Cancer Genome Atlas (TCGA[26]; Supplementary Fig. 4). Of note, in keeping with the melanocytic origin of the lines, reverse-phase protein array (RPPA) profiling revealed robust expression of the melanocyte marker gp-100 (Supplementary Fig. 3) and other key uveal melanoma factors including quinone oxidoreductase-1 (NQO1)[27] and DUSP4 (refs. 23,28; Supplementary Table 2). Several of the lines showed characteristic features of uveal melanoma, including monosomy 3, and either full or partial amplification of chromosome 8, a late-occurring event associated with metastasis[29]. All cell lines showed substantial copy number alterations (Supplementary Fig. 2), which in each line involved almost all chromosomes. These data provide a platform against which to interpret the analyses outlined below.

### Combinatorial CRISPR library design and screening

Our combinatorial CRISPR library comprised 25,499 constructs containing pgRNAs targeting 514 gene pairs (Fig. 1a and Supplementary Table 4). This library consisted primarily of paralogous gene pairs ($n = 262$) and putative synthetic lethal gene pair targets ($n = 210$). Paralog pairs were selected such that they had a minimum protein sequence identity of 45% (Ensembl v92) and a single essential ortholog in *Drosophila melanogaster* or *Caenorhabditis elegans*, where disruption of this ortholog conferred a lethal phenotype in these organisms (FlyMine, FB2015_15; WormBase, WS251; Methods). We reasoned that selecting paralogs in this way, as we have described previously[30], would enrich for lethal paralog pair interactions in human cells. Additional paralogous/nonparalogous putative synthetic lethal gene pairs were identified by linear modeling and data integration with MASH-up[31], using pan-cancer TCGA expression[32] and copy number data, together with data from loss-of-function RNAi-based screens and copy number profiles from the Cancer Cell Line Encyclopedia cancer cell line dataset[33]. We also used RNA-seq and CRISPR–Cas9 data from Project Score[34,35] to systematically identify associations between gene expression levels and essentiality, thereby prioritizing candidate pairs where low expression of one gene predicted essentiality for another (Methods). These two approaches identified 115 and 95 gene pairs of the abovementioned 210 putative synthetic lethal gene pairs, respectively. In addition to these collections, we also included a subset of gene pairs relevant to uveal melanoma ($n = 42$), with one member of the pair known to be downregulated within a gene expression profile predictive of uveal melanoma outcome[36,37], while the other gene in the pair was either a paralogous gene or a candidate synthetic lethal gene based upon the database of synthetic lethal interactions (SynLethDB)[38]. Additional candidates came from mutual exclusivity analysis of TCGA uveal melanoma genome data[39] (Methods). Of note, this group of genes included drivers for uveal melanoma, such as *BAP1* paired with other genes. Ten safe-targeting guides (STGs), which had previously been designed against regions of the genome with no annotated function[40], were also used in our library (Supplementary Fig. 5). These STGs were paired with sgRNAs targeting control essential ($n = 53$) and nonessential ($n = 112$) single genes (Methods) to allow technical assessment of screen performance and with each sgRNA targeting a library gene (that is, each gene within the abovementioned 514 gene pairs) to allow the single-gene knockout effect to be computed in comparison to codisruption of both members of a gene pair.

The pgRNA construct was designed with each sgRNA placed under an independent promoter (human U6 or mouse U6) and with different tracrRNA scaffolds to reduce recombination (Fig. 1b and Supplementary Fig. 5). To compensate for any possible promoter bias, we positioned an equal number of guides for each gene behind each promoter, selecting eight individual sgRNAs per gene. This resulted in 32 unique sgRNA pairings for every gene pair (Fig. 1c and Supplementary Fig. 5). Genes with fewer than six available unique sgRNAs were excluded from the library.

Each of the ten human uveal melanoma cell lines was engineered to stably express Cas9 and validated for high activity (>85%; Methods; Supplementary Fig. 6). Screens were performed at 1,000-fold library representation in triplicate for up to 28 days post-transduction with null-normalized mean difference (NNMD), strictly standardized mean difference values and Pearson correlations between screen replicates (Fig. 1d and Supplementary Figs. 7–9) suggesting quality control (QC) concordant with Project Score/DepMap screens. We next analyzed the $\log_2$(fold change (FC)) of all double knockout pgRNAs targeting gene pairs and compared these values with the $\log_2$(FC) of single-gene knockouts where vectors contained a sgRNA targeting a gene paired with an STG sgRNA. This analysis revealed significantly lower $\log_2$(FCs) within the double knockout gene pair group ($P < 2.2 \times 10^{-16}$, Wilcoxon rank-sum test), which indicates the significantly more deleterious effect of disrupting two genes in these cells and the potential that there exist gene pairs where there is a synergistic (synthetic lethal) effect on cell fitness (Figs. 1d–e).

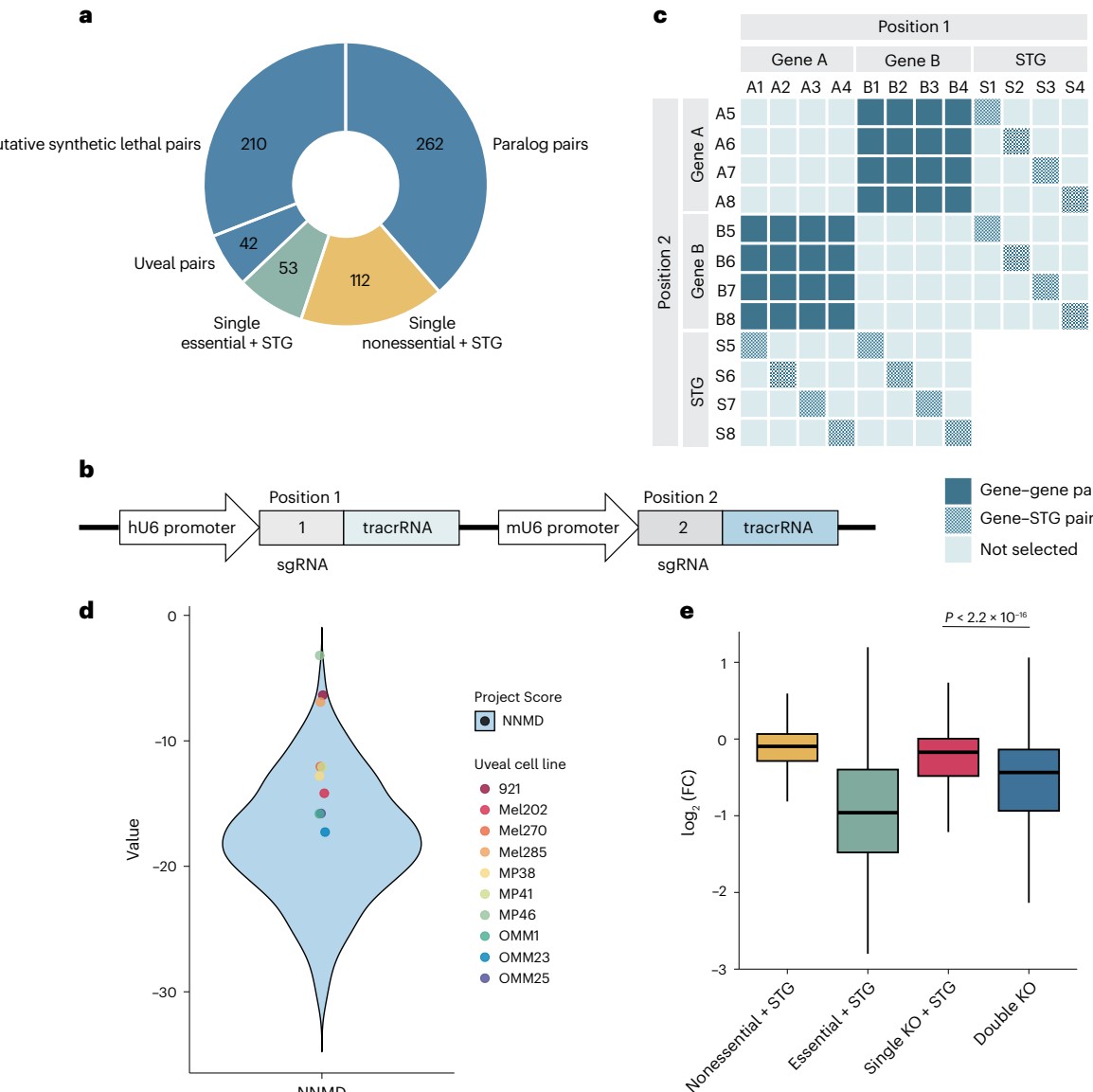

**Fig. 1 | Development of a combinatorial CRISPR library to screen for digenic dependencies in uveal melanoma. a**, Composition of the combinatorial CRISPR library. In total, the library targeted 514 gene pairs (blue), with each sgRNA targeting a gene in these pairs also found in the library with an STG, a design allowing us to compute the single versus double guide effect on cell fitness. The individual selection of sgRNAs in each category is described in the Methods. In addition to the abovementioned 514 pairs, we also included a collection of single essential (green) and single nonessential (yellow) genes paired with STGs to facilitate the calibration of our downstream analysis. **b**, Paired sgRNA construct. The first sgRNA in position 1 was placed under an hU6 promoter, and the second sgRNA in position 2 was placed under an mU6 promoter. Nonidentical tracrRNAs were used to minimize recombination (Supplementary Fig. 5). To assess single-guide activity, genes were paired with STGs placed in either position. **c**, Combinatorial sgRNA pairing strategy. For gene pairs A–B,

48 pgRNA combinations were used in the library where possible. In position 1, 12 sgRNAs were selected: 4 each against gene A (A1–A4), gene B (B1–B4) and safe-targeting regions of the genome (S1–S4). In position 2, another 12 unique sgRNAs were selected—4 each against gene A (A5–A8), gene B (B5–B8) and safe-targeting regions of the genome (S5–S8). This balanced design allowed us to account for any differences in sgRNA efficiency at either position. **d**, Screen quality was computed by calculation of NNMD values[35] for essential/nonessential genes in the library revealing a screen performance as good as DepMap/Project Score (further QC metrics are available in Supplementary Figs. 6–9). **e**, Boxplot of guide-level log$_2$(FCs) for each category of pgRNA across all ten cell lines. The colors correspond to those in **a**. The box shows the IQR; the line marks the median; whiskers extend to data within 1.5× IQR from Q1 and Q3; and points beyond are outliers. *P* values were computed using a two-tailed Wilcoxon rank-sum test (no correction was applied). IQR, interquartile range.

## Landscape of digenic dependencies in uveal melanoma

We used the Bliss independence model to identify synergistic gene pairs[41], predicting the dual knockout log$_2$(FC) to equal the sum of the individual knockout log$_2$(FCs). This allowed us to quantify the difference between the predicted and observed log$_2$(FC) as the genetic interaction (GI) score (Fig. 2a). For each cell line, we called a gene pair a 'hit' if the mean normalized GI score was less than −0.5 and the false discovery rate (FDR) below 0.01 across all guide pairs and

replicates (Supplementary Tables 5 and 6). At this threshold, we identified 105 unique gene pairs (20.4% of the library gene pairs) that were significantly depleted in at least one of the ten screened cell lines (Supplementary Table 7). Most of our hits were reproduced in at least two cell lines (>60%), indicating that there were relatively few hits that were cell line-specific in our screen. We next focused our analysis on significant digenic interactions that were shared among at least six of our ten screened cell lines, which allowed us

to consider the top 21.9% (23/105) of the total gene pair hits across all cell lines (Fig. 2b).

Of our 23 gene pair hits that were common to at least six of our screened cell lines, all had previously been screened in at least one independent combinatorial CRISPR study and all 23 gene pairs were paralog pairs, most coming from the approach of identifying conserved sequence-similar paralogs with essential orthologs in worms/flies, with some coming from the abovementioned MASH-up analysis approach. Notably, 22 of 23 pairs were identified as hits in at least one of these previous studies, with *SMARCC1/SMARCC2* as the only pair previously screened but not identified as a hit in any screen (Fig. 2c)[19–23]. However, low-throughput experiments have previously validated this gene pair as a synthetic lethal dependency[42,43]. This recovery of previously reported GIs suggests our screens are sensitive and robust.

While combinatorial screening can identify pairs of genes that represent synthetic lethal dependencies, we hypothesized that the complex genomic architecture of the uveal melanoma genome may obscure some synthetic lethal interactions. Specifically, loss-of-function mutations or transcriptional downregulation of one member of a target gene pair may hide/obscure an otherwise synthetic lethal interaction between genes, particularly if that genomic feature/expression profile is common to all/most of the uveal cell lines we screened. This phenomenon has been described previously for *VRK1/VRK2*, where *VRK1* is a single-gene dependency in glioblastoma because *VRK2* is frequently silenced[44]. Thus, we mined our combinatorial CRISPR data to identify single-gene essentialities that may have gene partners that are downregulated in uveal melanoma. To do this, we focused on paralogs and used data derived from all 514 screened pairs, as some paralog pairs were included in the library as part of the putative and uveal-specific gene sets. After removing reference essential genes[45], 19 such genes were identified to be commonly depleted across at least six of the ten cell lines. This included several known synthetic lethal gene pair members, such as *SMARCA4* and *NXT1*, which are known to have epistatic interactions with *SMARCA2* and *NXT2*, respectively (Fig. 2d)[43,46]. Using data obtained from Project Score[47], we observed negative correlations between several of our single-gene hits and the gene expression of their paralog partner, and this was particularly striking for *CDS2* and *INTS6* (Fig. 2e). Interestingly, *INTS6/INTS6L* had been screened in five cell lines across two independent published screens and was not reported to be synthetic lethal in either study[20,21], potentially suggesting the synthetic lethal interaction between this gene pair was masked by low *INTS6L* expression levels (Fig. 2e).

## CDS2 is a dependency in uveal melanoma

We next conducted genome-wide sgRNA CRISPR knockout screens in our panel of uveal melanoma cell lines with three primary objectives as follows: (1) to identify additional uveal melanoma-specific dependencies of single genes that were not assessed in the combinatorial CRISPR library; (2) to further examine digenic paralog CRISPR dependencies by perturbing each member within a paralog family and integrating

these data and (3) to provide orthogonal and complementary validation of the combinatorial CRISPR screen findings. For single gRNA library screening, each cell line was screened in technical triplicate after transduction with the Human Improved Genome-wide Knockout CRISPR Library v1.1 (ref. 47), cultured for 14 days and collected for sgRNA sequencing and quantification. A 14-day screen was performed so that the data we generated would align with data from Sanger's Project Score[34,35]. We preprocessed the screening data and corrected possible gene-independent responses to CRISPR–Cas9 targeting[48] with CRISPRcleanR[49,50] and validated screen performance by calculating the abovementioned NNMD and standardized mean difference values (Fig. 3 and Supplementary Fig. 10). This revealed high screen quality with scores like those of Project Score/DepMap. After excluding BAGEL reference essential genes (CEGv2; Methods) and filtering for genes that had a MAGeCK MLE beta score of less than −0.5 and an FDR below 0.05, a total of 861 genes were identified to be depleted/essential in at least one of the ten screened cell lines (Methods; Fig. 3a and Supplementary Tables 8–10). Notably, this gene list includes many essential genes, and although these genes could be therapeutically targeted[51], we reasoned that genes with a widespread effect on cell fitness across multiple cell and tissue types would be more likely to have effects/toxicity outside the target cancer cells. We therefore excluded all pan-cancer CRISPR-inferred common essential genes (DepMap 24Q2). From this analysis, we identified 8 genes from two-member protein-coding paralog families (Fig. 3b) and dependencies in two or more lines. The genes *CDS2(CDS1)*, *RIC8A(RIC8B)* and *SPTSSA(SPTSSB)* were a hit in ten, five and four cell lines, respectively. We next examined these pairs using TCGA uveal melanoma data[39] and the transcriptome analysis of our cell lines, observing expression profiles that align with our CRISPR results (Fig. 3c), thus revealing possible context-dependent and potentially targetable digenic vulnerabilities.

To extend this analysis further and to orthogonally computationally validate *CDS2, RIC8A* and *SPTSSA*, and to identify further candidates, including nonparalog genes, we adopted an additional approach to identify gene vulnerabilities that were specific to uveal melanoma. We again leveraged pan-cancer essentiality data derived from DepMap[52] (DepMap 22Q2) and the Mann–Whitney *U* test to calculate a *P* value using rank-normalized essentiality scores as input. Rank normalization allowed us to compare our screen results with those of DepMap and to define statistically significant uveal melanoma essential genes (Methods). Notably, before this analysis, we first removed any uveal or cutaneous melanoma lines screened as part of the DepMap resource. Following these steps, our analysis identified 76 genes as statistically significant ($\log_2(FC) > 1.80$ and $P_{adj} < 0.01$; Fig. 4a and Supplementary Table 11). Gene set enrichment analysis using gene ontology (GO) pathways was performed and showed enrichment pertaining to diacylglycerol binding and CDP-alcohol phosphatidyltransferase activity (Fig. 4b), in keeping with chronic activation of the phospholipase C (PLC) signal transduction pathway in uveal melanoma[53–55]. Our top ten uveal-specific genes identified from this analysis (Fig. 4c) had clear

**Fig. 2 | Top digenic and monogenic dependencies identified from combinatorial pgRNA CRISPR library analysis. a**, Calculation of the GI score to quantify synthetic lethality. The sum of the observed $\log_2(FC)$ for sgRNAs targeting gene A and gene B was calculated to determine the predicted $\log_2(FC)$ of that gene pair. The difference between the predicted and experimentally observed $\log_2(FC)$ was also calculated. **b**, Number of significantly depleted gene pairs across the cell lines screened. **c**, Dot plot depicting top synthetic lethal gene pairs from the combinatorial CRISPR screen that were common to at least 6 of the 10 screened cell lines, ranked (top to bottom) by descending mean GI score. Each dot represents the GI score of a given gene pair in a single cell line. All gene pairs had previously been screened across five independent CRISPR combinatorial screens and are colored by whether they were reported to be synthetic lethal in those studies[19–23]. Any pair where one of the genes was defined as lethal on its own was removed. **d**, Dot plot depicting significantly depleted

single genes (genes with an STG) that were defined by Ensembl to have a paralog and common to at least six of the ten screened cell lines, ranked (top to bottom) by descending mean normalized $\log_2(FC)$. $\log_2(FCs)$ are normalized so that the median of the STGs was 0 and the median of the reference essential guides was −1. Genes are labeled with their paralog in parentheses. Each dot represents a cell line. Color indicates the synthetic lethality across five previous combinatorial CRISPR screens[19–23]. A nonhit (gray) denotes a gene not previously reported to have a synthetic lethal interaction with its partner. Significance was defined using an FDR < 0.05 (Methods). **e**, Correlation between *INTS6L* expression and *INTS6* gene essentiality in 316 Project Score[47] cell lines, and the same for *CDS1/CDS2*. Low expressers ($\log_2(TPM + 1) < 1$) versus high expressers ($\log_2(TPM + 1) \geq 1$) were defined using expression profiles from Cell Model Passports. Significance was determined by a *t* test (two-sided, equal variance). Box and whisker plots indicate median and 5th to 95th percentiles; points are outliers.

biological roles related to melanocyte differentiation and survival (*SOX10*, *MITF*), G-protein coupled receptor (GPCR) signaling (*GNAQ*, *RIC8A*) and downstream phosphoinositide signaling (*CDS2* and *CDIPT*), and by themselves represent possible therapeutic targets. Of note, our findings of *CDS2* essentiality are consistent with our combinatorial pgRNA screen results, providing additional support for *CDS2* as a possible target in uveal melanoma, a result in keeping with several observations in the literature in other tumor types[20,21,56,57]. Intersection of our genomic data with our list of 76 uveal melanoma-specific genes revealed further driver gene candidates (Supplementary Fig. 11).

## Orthogonal validation of the *CDS1*/*CDS2* interaction

*CDS1* and *CDS2*, located on chromosome 4 (84,583,127–84,651,334) and chromosome 20 (5,126,879–5,197,887), respectively, are highly sequence-similar paralogs that encode key enzymes in the synthesis of phosphoinositides, which have key regulatory roles in the MAPK, AKT and other pathways[58,59]. *CDS1*/*CDS2* are each other's only identifiable paralog in the human genome and are widely conserved across species, with orthologs identifiable in both *D. melanogaster* and *C. elegans*, and in budding and fission yeasts. Of note, CDP-diacylglycerol synthases regulate the growth of lipid droplets and adipocyte development[60].

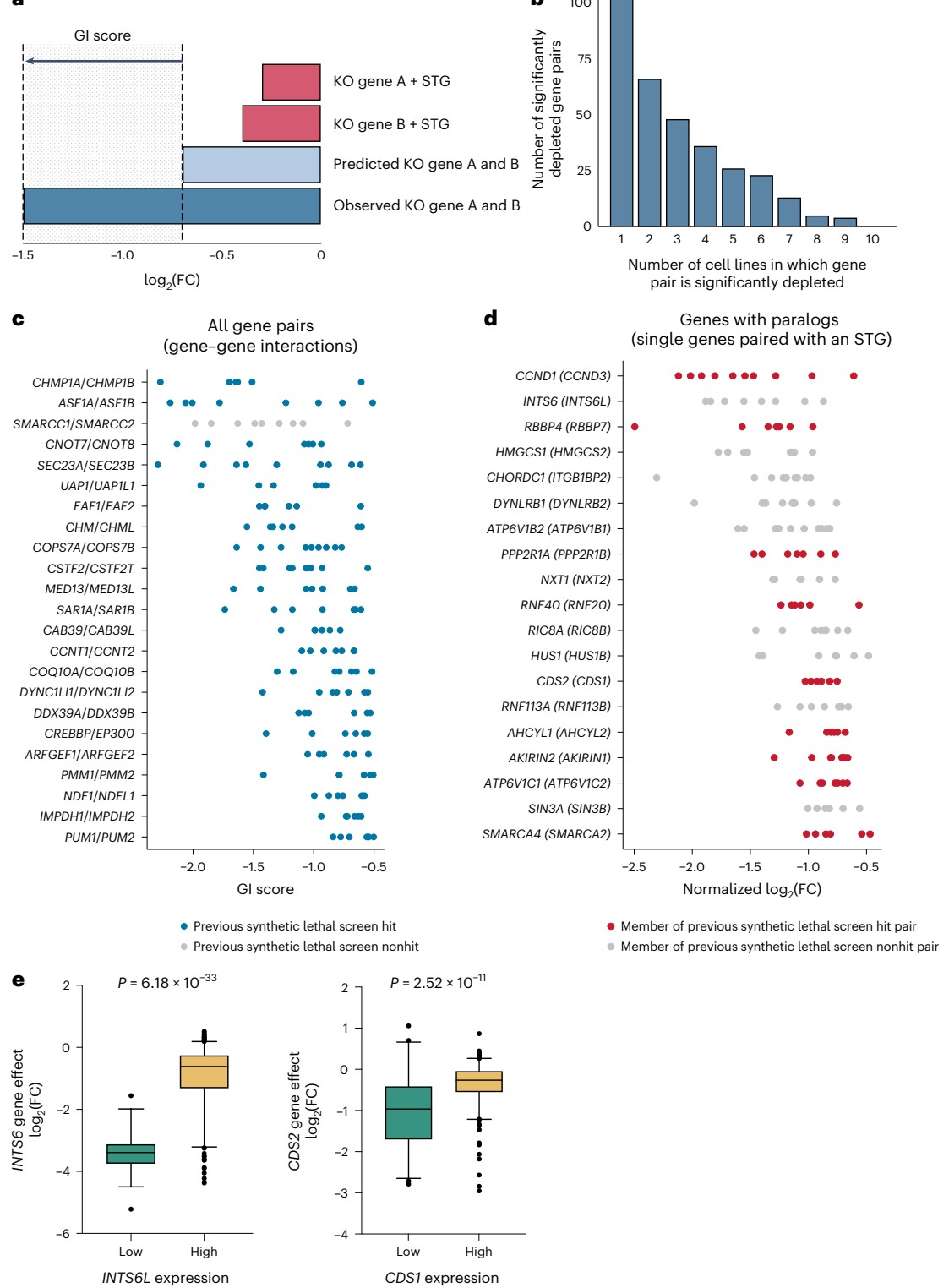

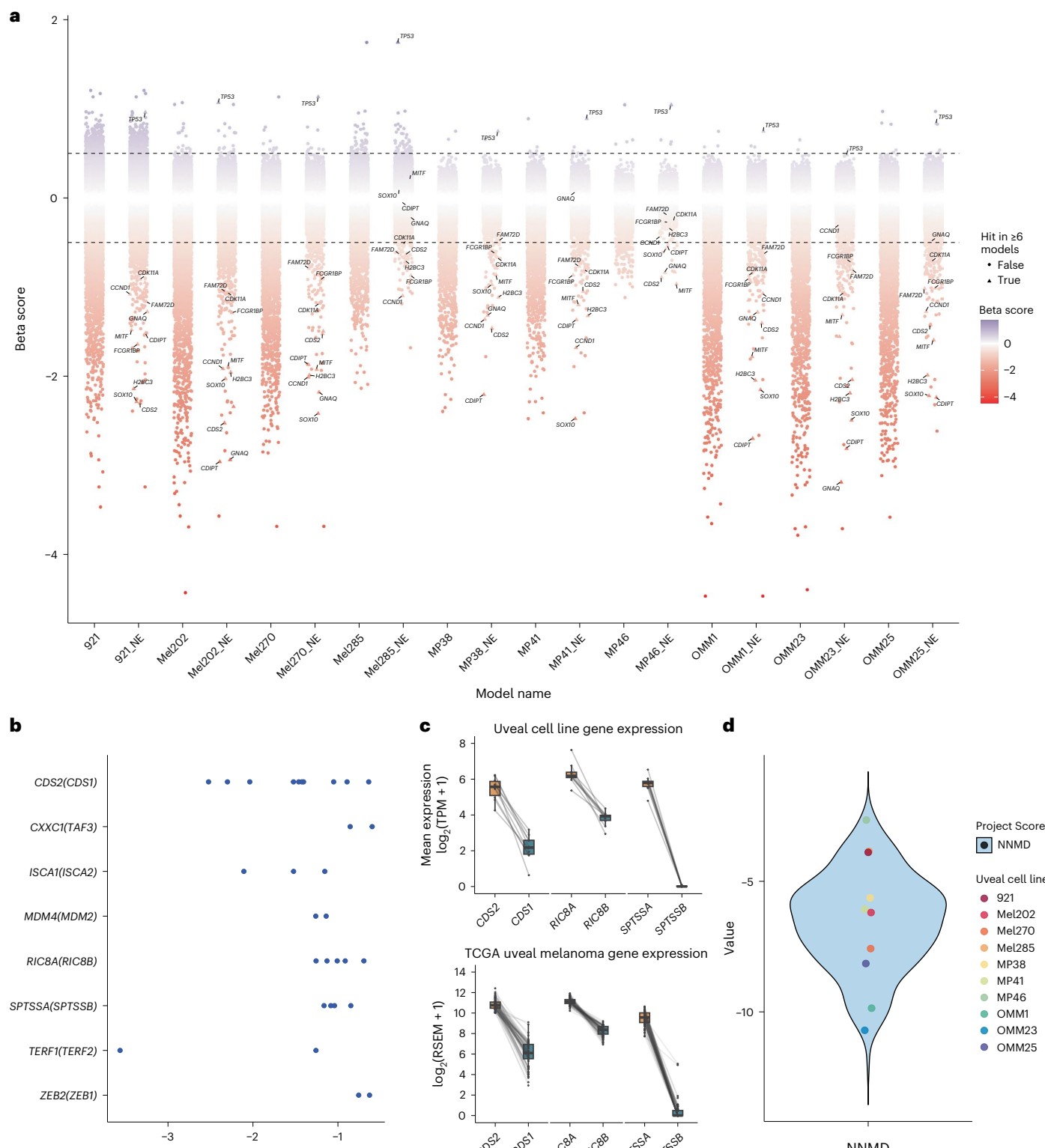

**Fig. 3 | Genome-wide single gRNA CRISPR screening identifies *CDS2* as an essential gene in uveal melanoma. a**, Plot showing the distribution of genes in each cell line following genome-wide CRISPR screening. Dotted line denotes a MAGeCK MLE beta of +0.5 or −0.5. For each cell line, '_NE' indicates scores filtered for common essential genes (DepMap 24Q2) and core essential genes (CEGv2), with genes that are significant in ≥6 lines shown as triangles (MLE beta +0.5 or −0.5 and an FDR < 0.01). The genes significant in most lines are labeled even if they are not significant in a specific line. **b**, Genes from two-gene paralog families that were a hit displayed with their beta values. The paralogous genes for these screen hits are shown in parentheses. Each dot represents a cell line. **c**, Top paralog gene pairs identified from single gRNA CRISPR data. The data shown is the relative gene expression levels of these pairs derived from our ten screened uveal cell lines (top) and the 80 tumors in the TCGA uveal melanoma dataset[38] (bottom). These data illustrate that in patient samples, the genes *CDS2*, *RIC8A* and *SPTSSA* are robustly expressed at levels much higher than their paralog gene partner. Also shown is an analogous expression pattern in our uveal melanoma models (Supplementary Table 3). The box shows the IQR; the line marks the median; whiskers extend to data within 1.5× IQR from Q1 and Q3. **d**, NNMD values for the whole-genome screens for each of the ten cell lines analyzed in comparison to Project Score screens[47].

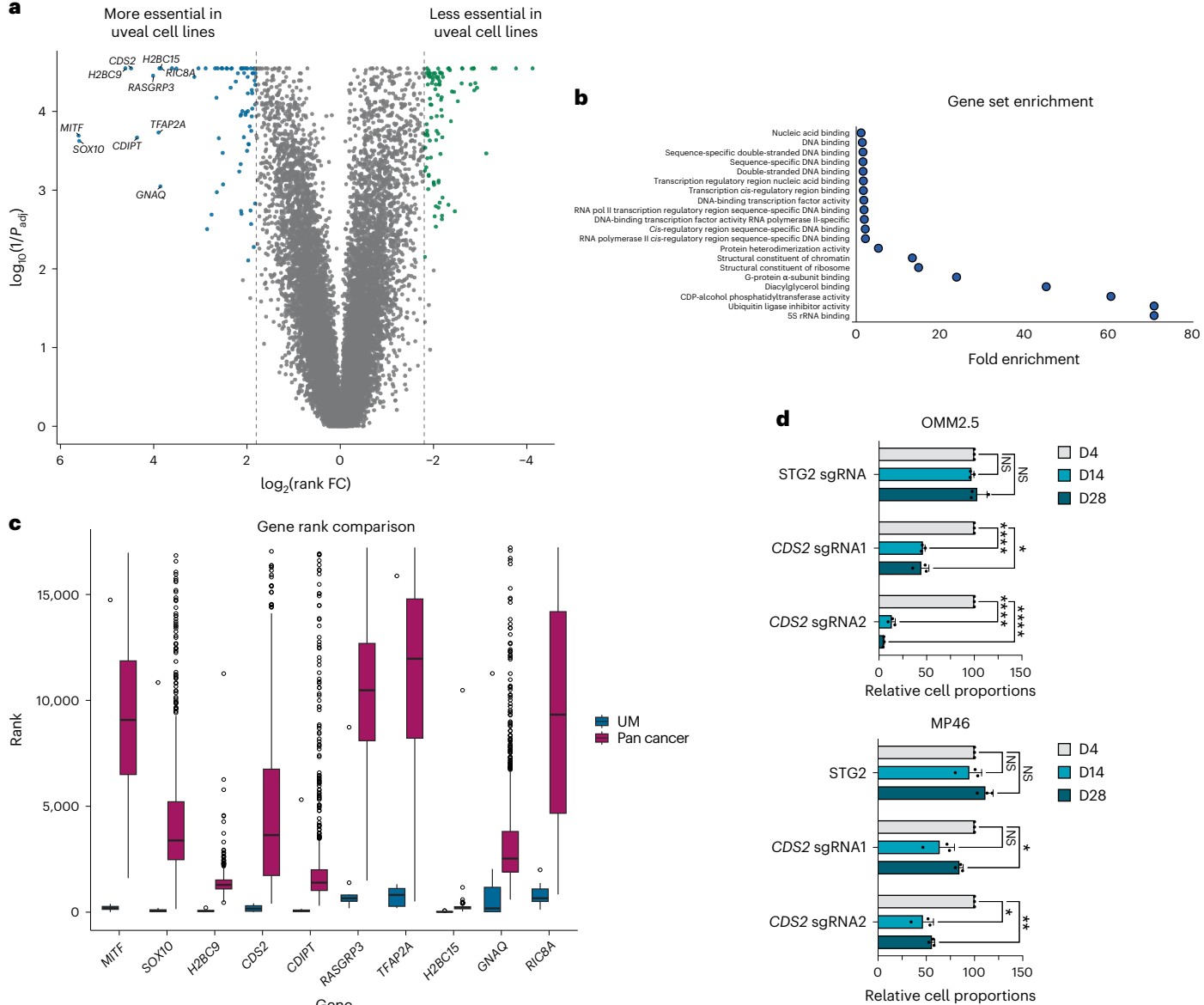

**Fig. 4 | Uveal melanoma-specific essential genes are involved in GPCR signaling and phosphoinositide signaling pathways. a**, Uveal melanoma-specific gene hits (log₂(FC) > 1.80 and $P_{adj}$ < 0.01; two-tailed Mann–Whitney $U$ test with Benjamini–Hochberg correction). Significantly differential essential genes for uveal melanoma cell lines were computed by comparing gene essentiality scores from our uveal melanoma screens to genome-wide sgRNA CRISPR screens from DepMap 22Q2 (Methods; Supplementary Table 11). Blue dots denote the top uveal-specific essential genes; green dots are genes less essential in uveal cell lines versus pan-cancer. **b**, Pathway enrichment analysis of the 76 uveal melanoma-specific genes using GO molecular function pathways. **c**, Dependency rank of top ten genes in uveal melanoma compared with pan-cancer cell lines. As above, pan-cancer data obtained from DepMap[52] 22Q2 release. The analysis compared

the 10 uveal lines to 982 pan-cancer cell lines. The box shows the IQR, the line marks the median, whiskers extend to data within 1.5× from Q1 and Q3, and points beyond are outliers. **d**, Validation of genome-wide sgRNA CRISPR screen hits with a competitive coculture assay (Methods). Cells were transduced with a lentivirus expressing an sgRNA and a BFP marker. BFP expression was measured by flow cytometry at baseline and on days 14 and 28. The proportion of surviving sgRNA-transduced cells compared with nontransduced cells is normalized to day 4. Two sgRNAs were tested against each gene, and an STG sgRNA was used as a control. Data represent three independent experiments performed in triplicate, with the mean and s.d. shown. Significance was calculated using a two-way ANOVA with Tukey's multiple test correction. ****$P$ < 0.0001, **$P$ < 0.01, *$P$ < 0.05 (exact $P$ values are provided in the source data).

Our primary objective was to further genetically validate the *CDS1/CDS2* interaction, focusing on *CDS2* as the target. *CDS2* sgRNAs were selected and cloned into a fluorescent protein-tagged vector, allowing us to quantify the dynamics of transduced cell populations over time (Methods). The results showed that by 14 days, *CDS2* was clearly required for uveal melanoma cell line fitness (Fig. 4d). Although the abovementioned sgRNAs were selected to have no off-targets, one concern of targeting genes such as *CDS2* is unwanted sgRNA-mediated *CDS1* disruption. We therefore transfected SW837 cells, a colon cancer line expressing both *CDS1/CDS2*, with our highest activity *CDS2* sgRNA

(GAGTAAAGGAAATGAACCGG) and after selection, whole-exome and transcriptome sequencing were performed. In keeping with off-target predictions, we observed only on-targeting "multi-hit" indel formation in *CDS2*. This sgRNA was used for all downstream experiments (Supplementary Fig. 12).

### *CDS1/CDS2* synthetic lethality across multiple tumor types
In both our screened uveal melanoma cell lines (*n* = 10) and the TCGA uveal melanoma cohort[39] (*n* = 80), *CDS1* gene expression was consistently lower than that of *CDS2*. In the cell lines, *CDS1* had a median

expression of 2.2 (range = 0.6–3.2; $\log_2(\text{TPM} + 1)$), compared with 5.6 (range = 4.3–6.2) for *CDS2*. Similarly, in the TCGA cohort, *CDS1* exhibited a median expression of 6.1 (2.9–9.1; $\log_2(\text{RSEM} + 1)$), compared with 10.8 (10.0–12.4) for *CDS2* (Fig. 3c). Analysis of single-cell sequencing data from 26 uveal melanomas revealed high tumor cell expression of *CDS2* and low expression of *CDS1*, further highlighting a unique opportunity to selectively target the *CDS1*/*CDS2* axis in tumors by disruption or inhibition of *CDS2* (Fig. 5a). To determine whether this expression pattern could be observed in other cancers, we analyzed TCGA pan-cancer gene expression obtained from the UCSC Xena platform[61]. While *CDS2* was widely expressed among all cancer subtypes analyzed, we found *CDS1* expression to be low or absent in cases of cutaneous melanoma, glioblastoma, hematological malignancies, hepatocellular carcinoma and sarcoma (Fig. 5b), suggesting that targeting *CDS2* in these cancers may be a viable therapeutic strategy.

To determine whether the *CDS1*/*CDS2* synthetic lethal interaction was observed in other tumor types, we extended our analysis of *CDS2* gene essentiality to include 937 cell lines from Project Achilles (DepMap 22Q2)[34]. Any uveal melanoma cell lines present within the data sets were excluded to ensure that any observed effect was not driven by uveal melanoma alone. Notably, *CDS2* was significantly more essential in *CDS1* low expressers ($\log_2$(transcripts per million (TPM) + 1) < 1; $P = 8.59 \times 10^{-9}$; $t$ test (two-sided, equal variance)), and consistent with a synthetic lethal relationship between *CDS1* and *CDS2* (Figs. 5c–d). Analysis of methylation data from these cell lines (DepMap 23Q2) revealed promoter hypermethylation in cell lines that had low *CDS1* expression (Fig. 5e), suggesting a probable mechanism of gene silencing. Finally, given that these studies revealed epistasis between *CDS1*/*CDS2*, we asked if the re-expression of *CDS1* could rescue the cell fitness defect associated with *CDS2* loss. In this way, we demonstrated that lentiviral transduction of a *CDS1* cDNA restored colony formation in vitro upon *CDS2* knockout (Fig. 5f), albeit associated with a decrease in colony formation compared to *CDS1* overexpression alone.

## Mechanistic insights into the effects of *CDS2* loss

The highly active *CDS2* sgRNA (described above) was placed under the control of a doxycycline (Dox)-inducible promoter in cell lines that constitutively expressed Cas9. We first confirmed CDS2 protein depletion and the generation of frameshift alleles upon Dox treatment (Supplementary Fig. 13 and Extended Data Figs. 1–4). In colony-forming assays, Dox treatment of *CDS2* sgRNA-inducible MP41 and OMM2.5 cells resulted in fewer and smaller colonies compared with dimethyl sulfoxide (DMSO; vehicle)-treated cells or isogenic inducible lines carrying an STG sgRNA (Fig. 6a; Methods). Of note, in apoptosis assays we observed a significant increase in apoptotic (Annexin V+/DAPI−) and dead cells (Annexin V+/DAPI+) following the induction of *CDS2* sgRNA expression, suggesting that *CDS1*/*CDS2* loss results in reduced cell fitness via apoptosis rather than cytostasis (Fig. 6b). Using these Dox-inducible OMM2.5 and MP41 cell lines, we next performed mass-spectrometry revealing significant loss of CDS2 protein in both lines, a result confirmed by Western blotting. We also collected a shared compendium of 50 upregulated and 27 downregulated proteins, with pathway analysis revealing downregulation of genes involved in DNA replication and upregulation of genes involved in cholesterol synthesis (Extended Data Fig. 2; Supplementary Table 12). We extended these analyses further using liquid chromatography–mass spectrometry (LC–MS) to quantify phosphoinositides and their metabolic precursor PA over 7 days. We observed that following disruption of CDS2, there was a progressive reduction in the phosphoinositides PI and PIP and in several phosphatidylinositol biphosphate ($PIP_2$) species (Fig. 6c and Extended Data Fig. 3). Strikingly, we observed supersized lipid droplets that were formed because of the deletion of *CDS2* (Fig. 6d), likely related to the corresponding increase in PA, the substrate for CDS enzymes. This effect has been previously reported[58,62], but in uveal melanoma cells seems particularly impressive, indicating the accumulation of

triacylglycerol due to the diversion of precursor molecules not used for PI synthesis. Because GPCR signaling is intrinsically linked to uveal melanoma development and phosphoinositide synthesis, we next asked whether there was an association between the expression of GPCR pathways involved in receptor-ligand or signal transduction and the Chronos score for *CDS2* across DepMap. This pan-cancer and unbiased analysis revealed a Kolmogorov–Smirnov $P$ value of $2.2 \times 10^{-16}$, where higher *CDS2* essentiality was associated with higher GPCR signal transduction activity and specific pathway enrichments, including those involved in Gq and PLC signaling (Extended Data Fig. 4).

Finally, to gain additional mechanistic insights, we next attempted to identify genes whose disruption could rescue the lethality observed following *CDS2* loss, reasoning that any such genes would represent potential pathways to drug resistance in patients treated with an agent that inhibited or depleted CDS2. To do this, we conducted a genetic suppressor screen using the MinLibCas9 library in an MP41-derived clonal cell line carrying a Dox-inducible *CDS2* sgRNA allele (above; Methods; Supplementary Figs. 13 and 14). Notably, we were unable to identify any genes under significant positive selection at an FDR below 0.05, suggesting limited escape to loss of *CDS2* in uveal melanoma (Supplementary Tables 13 and 14). This is notable as it suggests that an inhibitor of CDS2 may provide effective and sustained antitumor activity with minimal bypass/resistance mechanisms.

## Synthetic lethal interaction of *CDS1*/*CDS2* in vivo

Finally, given that our experiments assessing whether the effect of *CDS2* loss on cell fitness were performed in vitro, we next determined if we could extend our observations to the in vivo context. This is relevant because in vivo tumor cells need to grow in three dimensions and interact with host cells, such as stroma, which can attenuate phenotypes previously observed in vitro. To perform these experiments, we used OMM2.5 cells containing a highly active and specific Dox-inducible *CDS2* sgRNA construct (described above), selecting OMM2.5 from available lines because these cells readily and reproducibly formed xenografts. Notably, *CDS2* disruption in grafted cells resulted in significantly reduced uveal melanoma cell expansion, which prolonged the time to the tumor growth limit, when xenografted mice were fed a Dox-containing diet (625 mg kg$^{-1}$; ENVIGO). Of note, compared with control mice fed a normal diet, we saw no evidence of resistance/rebound tumor growth even up to 48 days after the experiment was initiated (Fig. 6e,f), suggesting that resistance was not acquired within this timeframe. To explore this question further and define possible mechanisms of resistance associated with *CDS2* disruption, we first performed PCR to detect whether residual cancer cells were still present at the graft site following prolonged Dox feeding. Subsequently, we sequenced RNA from these tissues using hybrid capture to enrich for human/cancer cell cDNA. It appeared that the survival of residual uveal melanoma cells was, in part, mediated by nondisruptive/in-frame events at the *CDS2* locus in our model (Supplementary Fig. 14).

## Discussion

Despite significant advances in our understanding of uveal melanoma biology and genetics, developing intuitive therapeutic strategies has been clinically challenging. In this study, we use an agnostic approach of both genome-wide sgRNA CRISPR screening and combinatorial pgRNA CRISPR screening to catalog monogenic and digenic dependencies. Combined with molecular profiling of all available uveal lines, we provide an invaluable resource for therapeutic target exploration in this disease. We observed a robust *CDS2* dependency in all ten uveal melanoma cell lines screened and, notably, demonstrated a negative impact of *CDS2* loss on phosphoinositide availability with a profound effect on cancer cell fitness, both in vitro and in vivo. Our analysis aligns with ref. 63, which independently identified the synthetic lethal interaction between *CDS1* and *CDS2* across a range of mesenchymal cancers. We extended our analysis of *CDS2* essentiality across multiple tumor

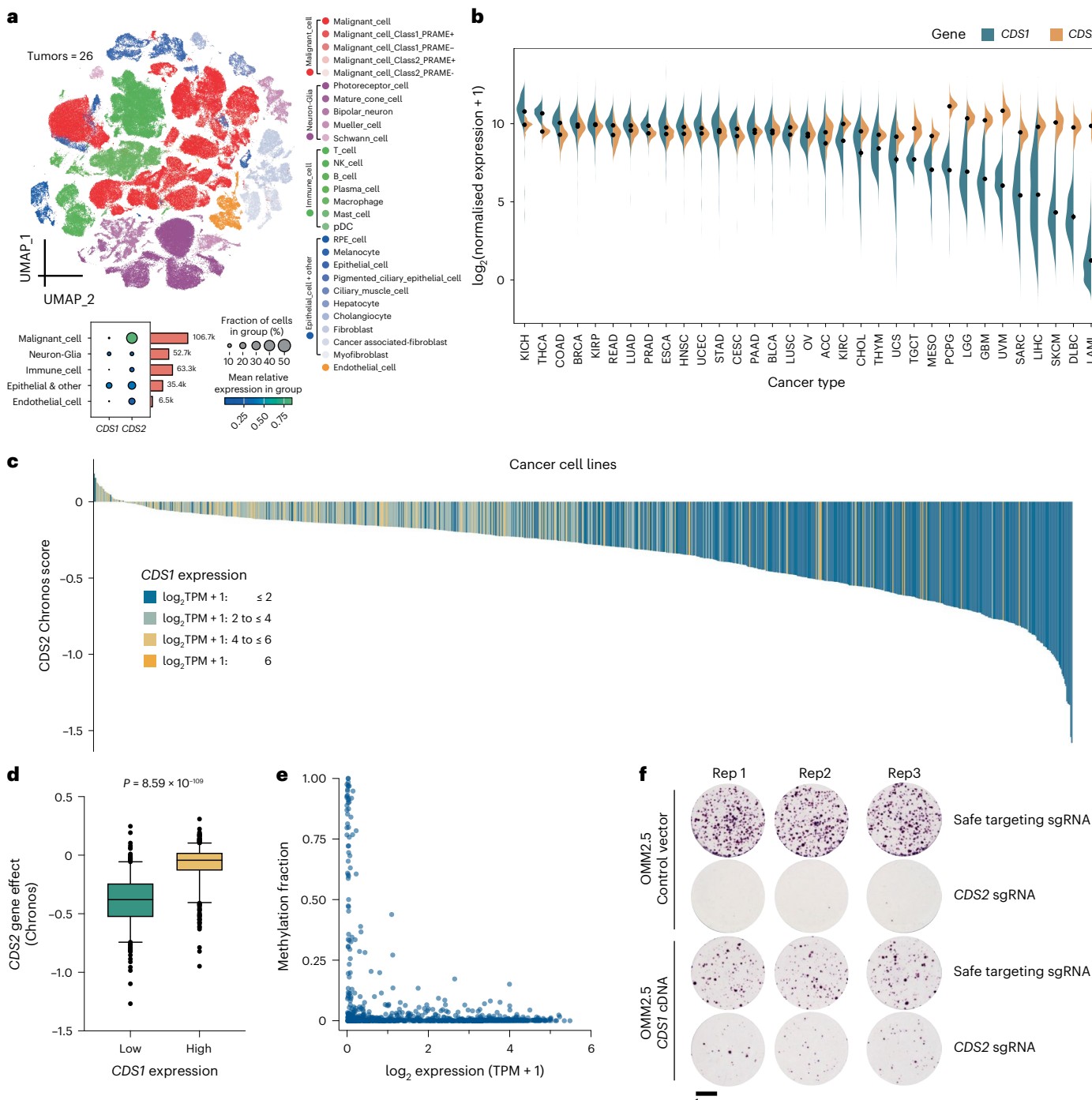

**Fig. 5 | Identification of *CDS1*/*CDS2* synthetic lethality across multiple tumor types. a**, Analysis of single-cell sequencing data from the uveal melanoma microenvironment reveals strong expression of *CDS2* in malignant cells. **b**, *CDS1* and *CDS2* gene expression in cancers based upon data generated by the TCGA Research Network and obtained from the UCSC Xena platform[61]. Shown is the data range (mean − s.e. to the mean + s.e.) with points equaling the mean. **c**, Waterfall plot depicting *CDS2* Chronos gene effect across 1013 pan-cancer cell lines from Project Achilles, DepMap 23Q4. Each column represents a cell line colored by *CDS1* expression. **d**, Comparison of *CDS2* essentiality in *CDS1* low expressers (log$_2$(TPM + 1) < 1) versus high expressers (log$_2$(TPM + 1) ≥ 1) across 937 nonuveal melanoma cell lines analyzed in Broad DepMap (22Q2). A *t* test (two-sided, equal variance) was performed to compute significance. Box and whisker plots indicate the median and the 5th to 95th percentiles. Outliers as dots. **e**, Correlation between *CDS1* methylation and expression. Methylation

fraction represents a weighted average of the methylation ratios of all CpG sites within 1,000 bp from a gene's transcriptional start site. The methylation ratio of each CpG site was determined by the number of reads where that CpG was methylated over the total number of reads covering that CpG. The weights of each CpG are calculated by the total number of reads covering that CpG overall and the total number of reads covering any CpG within 1,000 bp of the gene's transcriptional start site. Data obtained from DepMap 23Q2 (ref. 69). Each dot represents a cell line. **f**, Genetic rescue of *CDS2* lethality with a *CDS1* cDNA. OMM2.5 cells were transduced with either an sgRNA targeting *CDS2* or a control sgRNA (STG). As indicated, cells were also transduced with the *CDS1* cDNA and cultured for 14 days before staining with crystal violet. The data shown is representative of three biological replicates performed in triplicate. Each circle is a well of a six-well plate.

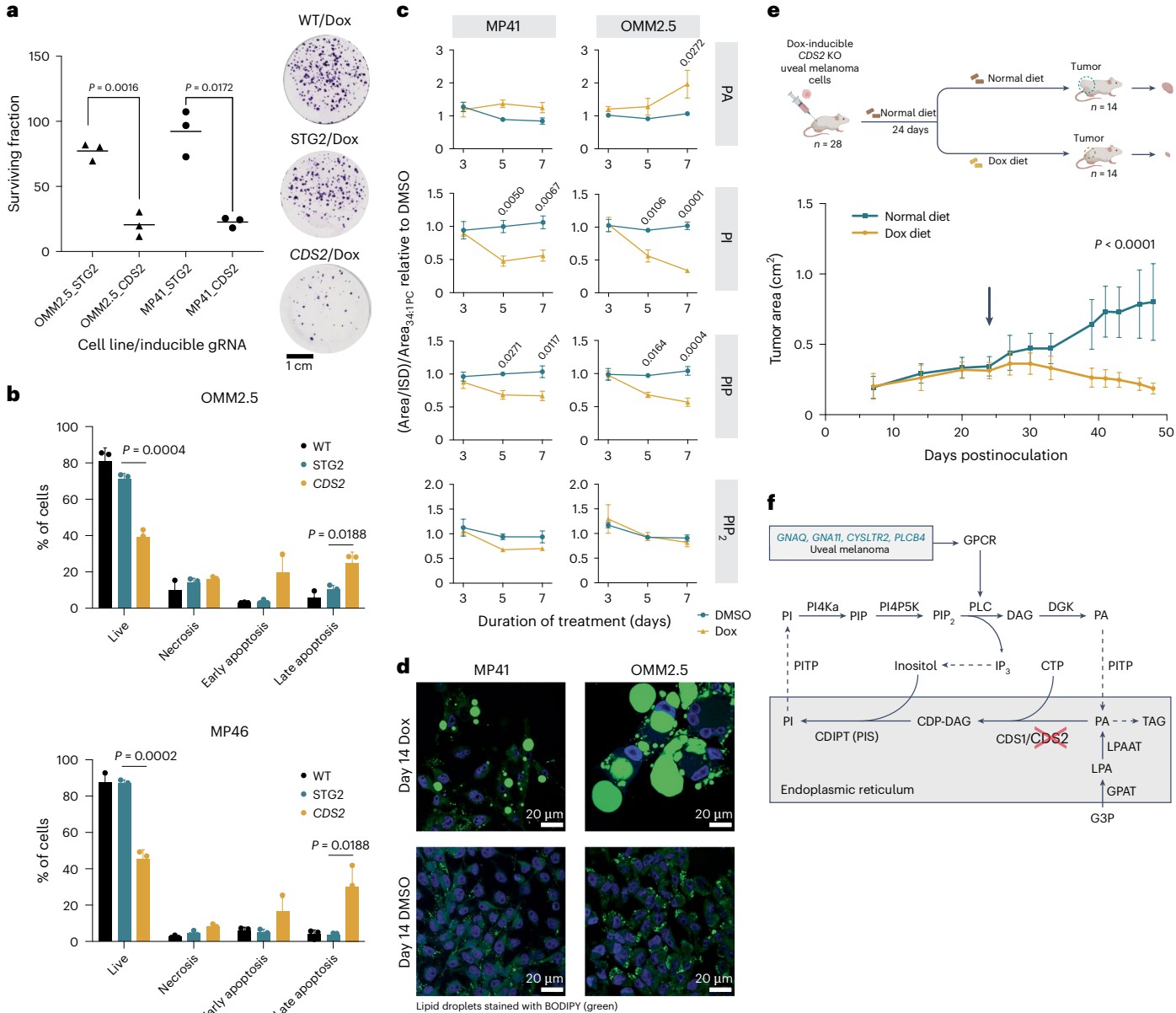

**Fig. 6 | Mechanistic insights into the effects of *CDS2* loss. a**, Clonogenic assay showing the effects of *CDS2* disruption at day 14 following the addition of Dox to Cas9-expressing cells that contain a Dox-inducible *CDS2* or an STG control sgRNA (STG2). Data were collected from three independent biological replicates for each cell line (triangles, OMM2.5; circles, MP41). Representative plates are shown for one replicate of the OMM2.5 cell line. *P* values were calculated with Welsh's two-tailed *t* test (without correction). Circles are a well of a six-well plate. **b**, Flow cytometry results from apoptosis assays at day 14 following *CDS2* sgRNA induction with Dox, compared with induction of an STG sgRNA control. Data collected from three biological replicates. Data shown are mean and s.d. *P* values were calculated using a two-tailed *t* test. Wild-type cells are shown for comparison. **c**, Disruption of *CDS2* following Dox treatment leads to a progressive reduction in the phosphoinositides PI and PIP and an increase in the precursor PA, compared with the DMSO control. Data are represented as mean ± s.e.m. from three independent MP41 clones and one OMM2.5 clone across three independent experiments. *P* values were calculated using a two-way ANOVA with Sidak's correction with significant values shown. **d**, 4,4-Difluoro-1,3,5,7,8-pentamethyl-4-bora-3a,4a-diaza-*s*-indacene (BODIPY 493/503) staining (green) of lipid droplets in cells treated with Dox or DMSO for 14 days. DAPI-stained nuclei (blue). This experiment was performed twice independently. **e**, Schematic of the in vivo experiments performed in 28 female mice at 8 weeks of age (NOD-*Prkdc*[scid]-*IL2rg*[Tm1]/Rj background). After implantation of tumor cells, mice were blindly randomized into two groups (Methods) with the switch to a Dox-containing diet occurring on day 24 (arrow). Data shown are mean ± s.d. Significance was determined using a mixed-effects model with the Geisser–Greenhouse correction. **f**, Schematic of the role of CDS1/CDS2 in phosphoinositide de novo synthesis and recycling.

types and, using available CRISPR–Cas9 screen resources, show that low *CDS1* expression predicts a cancer line's sensitivity to *CDS2* disruption. Thus, tumor *CDS1* expression may be a predictive biomarker for therapeutic CDS2 inhibition. Analysis of normal tissues from the GTEx resource suggests that *CDS2* is ubiquitously and strongly expressed. An important hurdle, however, will be to evaluate the effect of *CDS2* loss/depletion in normal cells where *CDS1* expression is low, including

hepatic and heart tissues[64] (Supplementary Fig. 15). In this regard, it is important to note that liver-specific knockout of *Cds2* from early stages of development resulted in reversible liver steatosis with PA, PI and phosphatidylglycerol levels not significantly affected in primary hepatocytes[65]. Similarly, deletion of *Cds2* in the vasculature results in reduced growth of grafted tumors with no adverse phenotypes[66]. A therapeutic window may also exist in uveal melanoma because

previous observations have shown loss of *CDS2* results in a significant reduction of GPCR-stimulated resynthesis of PI[67], with similar results observed here. Interestingly, almost all uveal melanomas harbor mutations that cause constitutive activation of GPCR signaling[39], and we provide evidence to suggest this may result in sensitivity to *CDS2* loss compared with other tissues with low *CDS1* expression, including normal tissues. Our proposition is that while low *CDS1* expression increases susceptibility to loss of cellular fitness from *CDS2* disruption, the true extent of *CDS2* dependency may be influenced by cell type and tumor-specific biological factors such as degree of chronic PLC activity, resulting in a viable therapeutic avenue for further investigation in cancers. We are intrigued by the observation that low *CDS1* expression is found in tumors, such as uveal melanoma and mesenchymal cancers. Our findings suggest that overexpression of *CDS1* impairs tumor cell fitness, suggesting that its expression is under tight physiological control. Finally, analysis of AlphaFold and AlphaFill data[68] for CDS2 suggests multiple druggable cavities, including the region containing the CDS enzyme domain, suggesting this protein is a tractable therapeutic target.

## Online content

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

¹Wellcome Sanger Institute, Hinxton, UK. ²Babraham Institute, Cambridge, UK. ³The Institute of Cancer Research, London, UK. ⁴Department of Clinical Neurosciences, University of Cambridge, Cambridge, UK. ⁵UK Dementia Research Institute, Cambridge, UK. ⁶Department of Molecular Neuroscience, Transylvanian Institute of Neuroscience, Cluj-Napoca, Romania. ⁷Human Technopole, Milan, Italy. ⁸Division of Molecular Oncology and Immunology, Netherlands Cancer Institute, Amsterdam, the Netherlands. ⁹Division of Molecular Carcinogenesis, Netherlands Cancer Institute, Amsterdam, the Netherlands. ¹⁰Oncode Institute, Utrecht, the Netherlands. ¹¹Instituto Superior Técnico (IST), Universidade de Lisboa, Lisbon, Portugal. ¹²INESC-ID, Lisbon, Portugal. ¹³Cancer Data Science Laboratory, Center for Cancer Research, National Cancer Institute, Bethesda, MD, USA. ¹⁴Laboratorio Internacional de Investigación sobre el Genoma Humano, Universidad Nacional Autónoma de México, Campus Juriquilla, Querétaro, Mexico. ¹⁵Laboratorio di Biologia Bio@SNS, Scuola Normale Superiore, Pisa, Italy. ¹⁶Conway Institute and School of Computer Science, University College Dublin, Dublin, Ireland. ¹⁷School of Medicine, University College Dublin, Dublin, Ireland. ¹⁸These authors contributed equally: Pui Ying Chan, Diana Alexander, Ishan Mehta, Larissa Satiko Alcantara Sekimoto Matsuyama. ✉e-mail: da1@sanger.ac.uk

## Methods

### Informed consent and ethics

All human datasets used in this paper were collected with the written informed consent of all patients and/or donors. The care and use of all mice in this study were in accordance with the UK Animals in Science Regulation Unit's Code of Practice for the Housing and Care of Animals Bred, Supplied or Used for Scientific Purposes, the Animals (Scientific Procedures) Act 1986, and all procedures were performed under a UK Home Office Project license (PP8090463), which was reviewed and approved by the University of Cambridge Animal Welfare and Ethical Review Body. We operate an open and inclusive research environment that is welcoming to all regardless of their background, beliefs or identity.

### Cell line whole-genome sequencing and analysis

Genomic DNA was extracted from cell lines using the Gentra Puregene Core Kit A (Qiagen), and library construction was performed using the NEBNext NEB Ultra II custom kit (New England Biolabs) according to the manufacturer's instructions. Genomes were sequenced such that, on average, 79% of the genome of each cell line was covered by >30× coverage, with the remaining 20% covered by more than 10× sequence depth. Sequencing was performed by Illumina HiSeq X Paired End Sequencing. SAMtools v1.10 was used to convert BAM files to FASTQ files. QC was performed using FastQC v0.11.8. Reads were aligned to GRCh38 using BWA mem v0.7.17. GATK v4.4.0.0 was used for germline short variant calling (SNVs, indels) with the HaplotypeCaller, as these cell lines are clonal, not tumors. Germline variants were removed with bcftools using gnomAD data, excluding all variants with an allele frequency >0.01. Identification of structural variants was performed using GRIDSS 2.0. Circos plots were generated in R v4.1.1 with StructuralVariantAnnotation v1.10.1, VariantAnnotation v1.40.0 and circlize v0.4.15 packages. Copy number calling was performed using CNVKit[70] version 0.9.8. More details are available on GitHub/Figshare along with a complete multi-QC report.

### Transcriptome sequencing

For each of the 10 uveal melanoma cell lines in this study, RNA was extracted from five technical replicate cultures using Qiagen kits (50 samples total). Illumina transcriptome libraries were prepared and sequenced as a pool on the Illumina 6000 platform to reduce possible batch effects. Reads were mapped using STAR v2.5.0c against the GRCh38 human reference genome with the ERCC spike-in sequences and ENSEMBL v103 gene annotation. To assess gene expression, reads were counted using HTSeq-count v0.7.2 with the "union" counting mode and appropriate stranded parameters. Finally, read counts were transformed to TPM. On average, each replicate was sequenced to generate 280 million reads, and on average, 20,434 genes were detected after mapping. Samples clustered tightly with high Pearson correlation coefficients. A full QC report is available via GitHub/Figshare. Transcriptome sequencing of SW837 cells, as part of *CDS2* sgRNA specificity assessment, was performed, generating at least 50 million reads per sample on the Illumina NovaSeq 6000 platform, with analysis essentially described as above.

### RPPA

RPPA analyses were performed by the RPPA core facility and MD Anderson Cancer Center[71]. Cell pellets containing $2 \times 10^6$ cells were lysed, and lysates were serially diluted and arrayed on nitrocellulose-coated slides to produce sample spots. Sample spots were probed with antibodies by a tyramide-based signal amplification approach and visualized using a 3,3′-diaminobenzidine colorimetric reaction to quantify sample spot densities. For each cell line, 50 μg of protein was used for RPPA analysis. Antibody validation was performed by Western blotting. Relative protein levels for each sample were determined by interpolating each dilution curve produced from the densities of the dilution sample spots using a standard curve for each antibody. All relative protein level data points were normalized for protein loading and transformed to linear values. Normalized linear values were transformed to $\log_2$ values and median-centered. A total of 425 antibodies were processed across all 10 human uveal melanoma cell lines.

### Cell culture and Cas9 cell line generation

All cell lines were authenticated by short tandem repeat profiling and screened negative for mycoplasma contamination. Cell-line source, culture conditions and antibiotic selection doses are provided in Supplementary Table 1. Cell culture media was replaced every 2–3 days. Cell lines were incubated in a humidified atmosphere of 5% $CO_2$ at 37 °C. To generate Cas9-expressing cell lines, all cell lines underwent lentiviral transduction with pKLV2-EF1aBsd2ACas9-W (Addgene; 68343) and blasticidin selection 72 h after transduction using a predetermined concentration according to a blasticidin dose–response assay (blasticidin range = 5–30 μg ml$^{-1}$). Cas9 activity was confirmed to be >85% for all cell lines before screening using a BFP/GFP reporter assay as previously described[72] (Supplementary Fig. 6). All Cas9-expressing cell lines were under blasticidin selection to maintain maximal Cas9 activity. For GFP/BFP assessment, we used uninfected cells to establish our control gates. First, we selected cells based on their forward scatter and side scatter properties to exclude dead cells and debris. Next, we gated for singlets and excluded doublets by analyzing the FSC-W versus FSC-A plot. The fluorescence of BFP and GFP cell populations was detected following excitation with the 405 nm and 488 nm lasers, respectively.

### Combinatorial CRISPR library design

Paralog gene pairs were included if they had >45% sequence homology (Ensembl v92) and single *D. melanogaster* (Flymine; FB2015_15) or *C. elegans* (Wormbase; WS251) essential orthologs. To computationally derive putative synthetic lethal gene pairs, we made use of the observation that GIs often occur among functionally related genes[73]. We first identified, for each gene, the top 50 functionally most similar genes using the Mashup algorithm[31] by calculating the Pearson correlation among (Mashup-based) gene-specific vectors representing an integration of the human STRING network. As demonstrated in Mashup's original publication[31], this strategy allowed the identification of gene pairs with highly similar biological functions, including—but not restricted to—gene paralogs. For these potentially interacting gene pairs, we next used ordinary least squares regression (as implemented by the Python package Statsmodels; https://www.statsmodels.org) on data from project Achilles[52] (v2.20.2) to test if copy number loss (defined as a copy number score <−0.3, GISTIC's default noise threshold for deletions) of one gene of a pair was associated with increased essentiality (which is, a lower DEMETER-based $Z$-score) of the partner gene, after adjustment for tissue of origin (503 cancer cell line essentiality scores available in the dataset were used for this analysis). Here we excluded gene pairs that resided on the same chromosome, as these follow similar copy number dynamics. Hence, a significant 'interaction' might be driven by increased sensitivity to knockdown when the gene itself is (heterozygously) deleted. For the resulting 357 significant (Benjamini–Hochberg FDR < 0.1) gene pairs, we used ordinary least squares regression to test, in a pan-cancer manner using data of 7,537 tumors/patients in TCGA (Supplementary Table 15), if the deletion of one gene (again defined as a GISTIC copy number score <−0.3) was associated with a significant upregulation of RNA expression of the partner gene. This resulted in 125 significant (Benjamini–Hochberg FDR < 0.1) gene pairs, 115 of which were taken forward and were used as our set of computationally derived putative synthetic lethal pairs for interaction testing in our combinatorial CRISPR screens. Additional putative synthetic lethal gene pairs were selected from a systematic association analysis, across 274 cancer cell lines (Supplementary Table 16), between gene expression and essentiality (Project Score[47] 2018 release). Associations

were tested using a linear regression model between gene expression (RNA-seq limma voom transformed)[74] and gene essentiality (CRISPR–Cas9 CRISPRcleanR corrected $\log_2$(FCs))[47,49]. The linear models were built using cell line growth rates and cancer type as covariates[75]. For each association, statistical significance was assessed using a log-ratio test, where the null hypothesis model does not consider the CRISPR–Cas9 gene essentiality; that is, only the covariates are used to fit the gene expression measurements. A $P$ value was then calculated using a Chi-square distribution with one degree of freedom. The top 95 gene pan-cancer associations at an FDR threshold <0.05 were selected. Uveal melanoma-specific genes were selected by identifying known tumor suppressor genes that undergo loss-of-function mutations in uveal melanoma and genes that are underexpressed based on gene expression profiling[36,37]. Corresponding gene partners were selected if they were either (1) paralogues within a two to three member paralogue family, (2) established mutually exclusive genes with loss-of-function mutations in uveal melanoma[39] or (3) synthetic lethal as defined by a database of synthetic lethal GIs (SynLethDB v1.0 (ref. [38])). There were 42 such pairs in the library.

To construct the library, sgRNAs from published genome-wide CRISPR knockout libraries were used to select eight sgRNAs per gene, comprising four sgRNAs from the Human CRISPR Knockout Pooled Library (Brunello)[76] and four sgRNAs from the Toronto KnockOut library (TKOv3)[77]. If fewer than eight unique sgRNAs were available from these two libraries, the remaining sgRNAs were selected from the Sabatini/Lander library[17] or designed using CRISPick (Genome Perturbation Platform, Broad Institute https://portals.broadinstitute.org/gppx/crispick/public). All pgRNAs included in the library are provided in Supplementary Table 4, and all oligos/sequences in Supplementary Table 17. Most gene pairs had a total of 48 unique pgRNA combinations, of which 16 comprised pairing of an sgRNA with an STG to assess single-gene knockout effects. When fewer than six unique sgRNAs were available for a gene, that gene was excluded from the final library. Ten previously published STGs designed against genomic regions with no previously defined function across 127 cell lines[40] were selected and validated by performing a surveyor assay using the Surveyor Mutation Detection Kit (IDT) according to the manufacturer's instructions (Supplementary Fig. 5). The paired gRNA construct contained a modified tracrRNA and spacer to prevent recombination (Supplementary Fig. 5 and Supplementary Table 17). The sequence of the vector, including the tracrRNA sequences, is available on Figshare[78]. As shown in Supplementary Fig. 5, the hU6 promoter in the construct is stronger than the mU6 promoter, and thus, we placed sgRNAs targeting each gene under the control of each promoter. We noted a small number of sgRNAs in this figure with strongly negative values. These are from later time points from the MEL285 and MP46 lines, where screen quality was poor. It does not reflect off-target or spurious STG sgRNAs.

### Combinatorial CRISPR library construction
The combinatorial CRISPR pgRNA library was constructed using a protocol based on the method established by Vidigal and Ventura[79]. The library oligonucleotide pool for the uveal component of the library and the remainder of the library were synthesized separately (Twist Bioscience) to allow use of the library without the uveal component in other models. Each oligonucleotide pool was PCR amplified using 25 μl of Q5 High-Fidelity 2× Master Mix (New England Biolabs), 0.05 ng of oligonucleotide pool, 5 μl of primer mix at a concentration of 10 μM and nuclease-free water to a final volume of 50 μl per reaction. The thermocycling conditions were 98 °C for 30 s, followed by 17 cycles of amplification at 98 °C for 10 s, 68 °C for 35 s and 72 °C for 30 s, and a final extension at 72 °C for 10 min. Annealed oligonucleotides for the main library and the uveal library were separately ligated into BbsI-digested pDonor_mU6 vector (Invitrogen GeneArt) in a 60 μl reaction containing 263 ng linearized pDonor_mU6 and 481 ng oligonucleotide amplicon with 30 μl Gibson Assembly Master Mix 2× (New England Biolabs) and

incubated at 50 °C for 2 h. Ligation of each insert to BsmbI-digested lentiGuide-Puro was performed using the Quick Ligation Kit (New England Biolabs) according to the manufacturer's instructions and electroporated into 10-beta electrocompetent *Escherichia coli* (New England Biolabs) and inoculated into 100 μg ml⁻¹ ampicillin-containing Luria–Bertani broth for 16 h, followed by plasmid DNA purification. Plasmid DNA from the uveal and main components of the library was combined in varying proportions to determine (via sequencing) the optimal combination with regard to pgRNA representation and correct proportion of pgRNAs from each library. Lentivirus production was carried out in $7.2 \times 10^7$ HEK293T cells transfected with 24 μg VSV-G, 40 μg pMDLg/pRRE, 20 μg pRSV-Rev, 67.2 μg pAdvantage and 69.6 μg of library plasmid with Lipofectamine 3000 (Thermo Fisher Scientific). Media was replaced 16 h after transduction, and viral supernatant was collected after 48 h, filtered through a 0.45 μM membrane (Merck; SLHP033RS), and stored at −80 °C.

### Combinatorial pgRNA CRISPR screening
For each cell line, $9 \times 10^7$ cells were transduced with the combinatorial CRISPR library at the required volume to achieve a transduction efficiency of 30% (1,000× library coverage), as determined by cell viability in puromycin (CellTiter 96 AQueous Non-Radioactive Cell Proliferation Assay (MTS), Promega). Puromycin concentrations for each cell line were determined using a dose–response assay (puromycin range = 2–4 μg ml⁻¹). Puromycin selection was performed at day 3 post-transduction for a total of 7 days. Library infections were performed in triplicate, and a minimum library representation of 3,000× was maintained at every passage throughout the screen. Cells were first collected at 28 days post-transduction. Cas9-negative cell lines were transduced under the same conditions and collected at day 7 to allow calculation of pgRNA FC. Genomic DNA was extracted for each replicate from cell pellets using a Blood & Cell Culture DNA Maxi Kit (Qiagen).

### Analysis of the combinatorial pgRNA CRISPR screen
Plasmid and genomic DNA sequencing were performed as previously described[21]. All primer sequences are included in Supplementary Table 17. Guide quantification was performed using pyCROQUET v1.5.1 (https://github.com/cancerit/pycroquet). Raw counts are provided in Supplementary Table 5. Normalized counts from Cas9-expressing lines were compared with normalized counts from control Cas9-negative lines to calculate log(FCs). sgRNAs with less than 20 read counts in the control lines were removed. Single-gene essentiality was tested through C-SAR v1.3.6 (https://github.com/cancerit/C-SAR) using BAGEL2 (ref. [80]) and MAGeCK[81]. log(FCs) were scaled so the median log(FC) of nonessential genes was 0, and known essentials were −1 ($\log_2$(FCs) are scaled $\log_2$(FCs) unless otherwise specified). The reference essential gene set (CEGv2) and reference nonessential gene set (NEGv1) were obtained from https://github.com/hart-lab/bagel. Synthetic lethal hits were called using an adapted version of the BASSIK method, as applied previously[21]. Hits were called using the Bliss independence model, where the observed reduction in fitness on knockout of both genes in combination is significantly greater than predicted from the addition of the single knockout fitness effects. Predicted dual knockout log(FCs) were adjusted using Loess regression, modeling the local behavior of the data to account for a plateau in dual KO log(FCs) when one gene is already close to essential. GI scores were calculated per pair of guides from the difference between the predicted log(FCs) for both guides in combination and the observed dual log(FCs). This was normalized by the square root of the binned variance to account for heteroscedasticity. A gene pair was called a hit if (1) the mean normalized GI score across all guide pairs and replicates was <−0.5 and (2) a one-sided $t$ test for whether the median gene pair GI score is less than the median GI score across the dataset had an FDR < 0.01 (Benjamini–Hochberg multiple testing correction). Hit pairs where either gene in the pair

was classified as singly essential by BAGEL and MAGeCK were filtered out to avoid calling hits identified because of one gene being essential.

## Assessment of *CDS1* expression and *CDS2* essentiality

To test the association between *CDS1* expression and *CDS2* essentiality across all cancer types, gene essentiality data were obtained from Project Score (https://score.depmap.sanger.ac.uk/downloads) and Project Achilles (DepMap 22Q2, https://depmap.org/portal/download/all/). For Project Score essentiality scores, we used quantile-normalized log(FC) values corrected using CRISPRcleanR[49]. Quantitative essentiality data (copy number corrected Chronos scores[82]) for the Project Achilles data set were downloaded from the DepMap portal (22Q2 release). Gene expression data were obtained from Cell Model Passports (https://cellmodelpassports.sanger.ac.uk) for Project Score cell lines and the DepMap download portal (22Q4) for Project Achilles cell lines. Cell lines with *CDS1* expression $\log_2(\text{TPM} + 1) < 1$ were annotated as low expressers, and the rest were annotated as high expressers. *t* tests (two-tailed, equal variance) were performed to test whether *CDS2* is more essential in *CDS1* low expressers.

## Whole-genome single gRNA CRISPR knockout screen and analysis

Genome-wide sgRNA CRISPR knockout screening was performed using the Human Improved Genome-wide Knockout CRISPR Library v1.1 (ref. 47). For each cell line, $3.3 \times 10^7$ cells were transduced with the required volume of Human CRISPR Library v1.1 to achieve a transduction efficiency of 30% (100× library coverage). Puromycin selection (2–4 µg ml$^{-1}$) was performed at day 3 post-transduction for a total of 7 days. Screening was performed in technical triplicate, and cells were cultured throughout the screen at a minimum library representation of 500×. Cells were collected 14 days post-transduction, pelleted and stored at −80 °C. Genomic DNA was extracted from cell pellets using a Blood & Cell Culture DNA Maxi Kit (QIAGEN). PCR amplification and sequencing were performed as previously described[72].

All samples were processed using CRISPRcleanR v3.0.1 (refs. 49,50). Only sgRNAs in common with the Human CRISPR Library v1.0 (ref. 72) were analyzed to prevent unbalanced targeting of essential genes with additional guides in version 1.1 (ref. 47). Raw counts are provided in Supplementary Table 9. sgRNAs with less than 30 read counts in the plasmid were removed. The remaining sgRNA raw counts were normalized by their total number within their replicate. Depletion and enrichment log-fold changes (logFCs) for individual sgRNAs were quantified at the individual replicate level between library plasmid read counts and postlibrary-transduction read counts. Gene-independent responses to CRISPR–Cas9 targeting were corrected using the default parameters. MAGeCK MLE v0.5.9.5 was run on the normalized corrected sgRNAs' treatment counts after applying the inverse transformation on the corrected log(FCs) as previously described[49]. Significantly depleted essential genes were identified as those with a MAGeCK MLE beta score < 0.5 and an FDR < 0.05. Previously established essential genes (CEGv2) were excluded from the list of essential genes.

## Identification of uveal melanoma-specific gene hits

To identify uveal melanoma-specific gene hits, the essentiality scores from our genome-wide sgRNA CRISPR screen and pan-cancer cell lines from DepMap 22Q2 were compared. Scores were rank normalized to allow for comparison between different screens. The Mann–Whitney *U* test was used to identify significant differences, and log2 fold changes from the ratio of median ranks were used to determine the direction of essentiality. Pan-essential genes identified using ProdeTool (https://github.com/cantorethomas/prodeTool) were removed from the results. UVM-specific vulnerabilities ($n = 76$; $\log_2(\text{FC}) > 1.80$ and $P_{\text{adj}} < 0.01$) were then used as an input for gene set enrichment analysis using the GO Molecular Function pathway gene set.

## Construction of the core UVmap reference

In developing the core UVmap (a uveal melanoma single-cell reference map), we used the GBmap pipeline approach as described in ref. 83, collecting data from 264,624 cells from 26 tumors and 29 normal tissues. We included only those samples confirmed as healthy eye or primary uveal melanoma, and healthy liver or metastatic liver, with each sample containing no fewer than 1,000 cells. The datasets for the core UVmap, which include refs. 84–87 were primarily in the form of raw count matrices. Where raw matrices were not available, BAM files were downloaded directly from the dbGaP cloud[88] (phs001861.v1.p1; approved by dbGaP on 24 May 2022) or sourced directly from the authors[89], and were then transformed into FASTQ files and re-aligned using the STARsolo v2.7.10a pipeline (https://github.com/cellgeni/STARsolo). We updated all gene names to the most current HUGO nomenclature via HGNChelper and ensured all clinical and diagnostic metadata remained consistent. Before integrating the datasets, we applied stringent filtering parameters to select only high-quality cells, excluding those with fewer than 500 genes, fewer than 1,000 unique molecular identifier counts (where applicable), and over 30% mitochondrial reads. Doublets in each droplet-based dataset were identified and removed using DoubletFinder.

To mitigate batch effects across the datasets, we employed a semi-supervised neural network model called single-cell ANnotation using Variational Inference (scANVI)[90], within the transfer-learning framework of the single-cell architectural surgery algorithm (scArches)[91]. scArches–SCANVI requires prior knowledge of cell types and labels to create a reference map. To standardize cell type labels from different sources, we annotated each dataset employing both automated and manual methods. For the automated process, we initially collected lists of melanoma and GEP markers from ref. 88, 16 cancer cell states[92], and a list of 174 adult eye and liver markers from a study published in ref. 93. We then performed UCell signature scoring[94] and applied a cutoff value of 0.2 to assign cells as state/marker positive.

Subsequently, manual cell identity was assigned based on results from the automated process, available original cell labels and specific gene expression patterns analyzed using the Wilcoxon rank-sum test. CNV analysis was conducted using the CopyKAT package[95], categorizing cells as either diploid or aneuploid. This preliminary coarse cell type labeling facilitated the training and integration of the model through scANVI-scArches.

The analysis was conducted on the raw counts from the 5,000 most variable genes, using studies as the batch variable and adhering to the recommended tool parameters. The output from the pipeline was a latent representation of the integrated data, which then served as input for clustering and dimensional reduction visualizations. We applied Leiden clustering based on a k-nearest neighbor graph (k-NNG)[96] to identify distinct cell populations, and Uniform Manifold Approximation and Projection (UMAP)[97] for data embedding and two-dimensional reduction, using the plot1cell package[98] for UMAP visualization. Post-co-embedding, cell identities were refined manually for each cluster, using our unified preliminary annotations and evaluating specific marker gene expression to accurately define each broad cell type or state.

## Validation of on-target *CDS2* sgRNA on-target activity

Because targeting closely related sequences, such as paralogs, could result in off-target cutting and paralog codisruption, we validated on-target sgRNA cutting for the sgRNA used in our validation experiments. We infected/transduced SW837_C9 cells stably expressing Cas9 and having Cas9 activity >90% with either a *CDS2* sgRNA (GAGTAAAGGAAATGAACCGG) or a safe-targeting control sgRNA (STG1; GTATCAACAGAGTGTCAGAT) at an MOI of 0.34–0.38. Cells were subsequently selected with puromycin for 7 days, and then RNA was extracted for whole transcriptome sequencing. The parental (uninfected) SW837_C9

cell line was also sequenced as a control. Analysis was performed as described above.

## Validation by sgRNA competitive proliferation assay

sgRNA competitive proliferation assays were performed using the pKLV2-U6gRNA5(BbsI)-PGKpuro2ABFP-W (Addgene; 67974) vector. To validate individual target genes, one sgRNA from the Human Improved Genome-wide Knockout CRISPR Library v1.1 (ref. 47) was used, and an sgRNA derived from the MinLibCas9 library[99]. Viral supernatants were collected 48 h after transfection into packaging lines, and transduction of *CDS1*-null uveal cell lines (OMM2.5 and MP46) was performed three times independently at an MOI between 0.5 and 0.8. Flow cytometry was performed on six-well plates using a CytoFLEX flow cytometer (Beckman Coulter) and analyzed with FCS Express v7.22.0031 to determine the proportion of BFP-positive cells at days 4, 14 and 28.

## Construction of inducible *CDS2* and STG2 knockout isogenic cell lines

A potent and specific sgRNA against *CDS2* (validated above) or a safe-targeting sgRNA (STG2) was cloned into a tet-inducible vector pRSTGEBleo-U6Tet-sg-EF1-TetRep-2A-Bleo (Cellecta) according to the manufacturer's instructions and sent for Sanger/capillary sequencing to confirm correct assembly. Vectors were transduced into MP41-Cas9 and OMM2.5-Cas9 cells. Where clones were used, each transduced cell line was sorted at single-cell density into 96-well plates containing media and then zeocin selected for 4 weeks; polyclonal lines were selected for 2 weeks. To confirm editing of the *CDS2* gene, genomic DNA was extracted from colonies using the Gentra Puregene Blood Kit (Qiagen) and sent for Sanger sequencing. Western blot analysis was performed on *CDS2* knockout clones using an anti-CDS2 antibody (Proteintech, 13175-1-AP; 1:1,000 dilution) and an anti-vinculin antibody (Thermo Fisher Scientific, MA5-11690; 1:10,000 dilution) as a loading control (Extended Data Figs. 1 and 2). Thermo Novex Sharp prestained protein markers (Thermo Fisher Scientific) were used.

## Clonogenic assay

In each 6-well plate, up to 1,300 Dox-inducible cells were seeded per well. After 24 h, media were replaced with media containing 0.1 µg ml$^{-1}$ Dox or the equivalent volume of dimethyl sulfoxide (DMSO). After 14 days, colonies were washed with ice-cold PBS, fixed with 4% paraformaldehyde (PFA) and stained with 0.1% crystal violet solution.

## Genetic rescue of *CDS2* loss with a *CDS1* cDNA

Constitutively Cas9-expressing OMM2.5 cells were transduced with a lentivirus containing a full-length *CDS1* cDNA (Origene; RC210375L3) and selected in puromycin for 7 days. Transduction with an empty vector (Origene; PS100092) followed by puromycin selection for 7 days served as a negative control. Cells were then transduced with lentivirus containing a *CDS2* sgRNA (GAGTAAAGGAAATGAACCGG) or a safe-targeting sgRNA (STG1) at a viral titer that achieved 100% transduction. Subsequently, 1,200 sgRNA-infected cells were seeded per well in a six-well plate in technical triplicate. Following culture for 18 days, colonies were washed with ice-cold PBS, fixed with 100% ethanol and stained with 1% crystal violet solution. The aforementioned experiments were performed in biological triplicate (that is, three independent trials).

## Apoptosis assessment

Apoptosis and cell death were assessed by flow cytometry using Annexin V staining. We infected the uveal melanoma cell lines with a *CDS2* sgRNA (GAGTAAAGGAAATGAACCGG) or a safe-targeting control sgRNA (STG2; GTATCAACAGAGTGTCAGAT) lentivirus at an MOI of 0.8. Cells were selected with puromycin for 4 days, and then a total of $2 \times 10^5$ cells were plated in each well with 2 ml of media. At day 14, floating and adherent cells were collected from each well. Cells were washed with

500 µl of Annexin V Binding Buffer (BD Biosciences) and resuspended in 100 µl dilution containing 5 µl PE Annexin V (BioLegend) for 15 min at room temperature. Cells were centrifuged and resuspended in 100 µl dilution with 20 µl 7-AAD (BD Biosciences) for 10 min and then pelleted and resuspended in 500 µl Annexin V Binding Buffer for flow cytometry analysis using a CytoFLEX flow cytometer (Beckman Coulter). Annexin V-/7-AAD- cells were deemed to be viable, and Annexin V+/7-AAD- cells were early apoptotic. Finally, Annexin V+/7-AAD+ cells were considered late apoptotic, and 7-AAD+ necrotic cells. Analysis was performed with FCS Express v7.22.0031.

## Proteome-wide analysis following *CDS2* disruption

These experiments were performed using clonal OMM2.5P and MP41 E cells and were performed in three biological replicate experiments with the sgRNA targeting *CDS2* activated using Dox (as above) for 7 days before cells were pelleted and snap frozen for analysis. Analysis was performed essentially as previously described[100]. Briefly, LC–MS analysis was carried out using a Dionex UltiMate 3000 UHPLC system coupled with an Orbitrap Ascend mass spectrometer (Thermo Fisher Scientific) and analyzed using a Real-Time Search-SPS-MS3 method. Approximately 3 µg of peptides from each fraction were injected onto a C18 trapping column (Acclaim PepMap 100, 100 µm × 2 cm, 5 µm, 100 Å) at a flow rate of 10 µl min$^{-1}$. The peptides were then subjected to a 120-min low-pH gradient elution on a nanocapillary reversed-phase column (Acclaim PepMap C18, 75 µm × 50 cm, 2 µm, 100 Å) at 50 °C. MS1 scans were conducted over a mass range of $m/z$ 400–1600 using the Orbitrap at 120,000 resolution, with standard AGC settings and automatic injection time. Charge states between +2 and +6 were included. Dynamic exclusion was set to 45 s with a repeat count of 1, a mass tolerance of ±10 ppm and isotopes were excluded from further analysis.

MS2 spectra were acquired in the ion trap using a Turbo scan rate with a higher energy collision dissociation energy of 32% and a maximum injection time of 35 ms. Real-time database searching was conducted against *Homo sapiens* (canonical and isoforms) using the Comet search engine with tryptic peptides, allowing a maximum of one missed cleavage. Static modifications included carbamidomethylation of C (+57.0215 Da) and TMTpro labeling on K and N termini (+304.207 Da). Variable modifications included deamidation of N/Q (+0.984 Da) and oxidation of M (+15.9949 Da), with a maximum of two variable modifications per peptide. Close-out was enabled, allowing a maximum of four peptides per protein.

Selected precursors were subjected to SPS10-MS3 scans using an Orbitrap detector at 45,000 resolution, with a higher energy collision dissociation energy of 55%, a normalized AGC target of 200% and a maximum injection time of 200 ms. Data were collected in centroid mode with a single microscan acquisition.

Proteome Discoverer 3.0 (Thermo Fisher Scientific) was used with SequestHT and Comet search engines for protein identification and quantification. Spectra were searched against Homo sapiens protein entries in UniProt, with a precursor mass tolerance of 20 ppm and a fragment mass tolerance of 0.02 Da. Fully tryptic peptides were considered, allowing up to two missed cleavages. Static modifications were TMT at N terminus/K and carbamidomethylation at C residues. Dynamic modifications included oxidation of methionine and deamidation of N/Q.

Peptide confidence was estimated using Percolator, maintaining an FDR of 0.01 with target-decoy database validation. Quantification was performed using the TMT quantifier node, with a 15 ppm integration window and the most confident centroid peak at the MS2 level. Only unique peptides were used for quantification, with a signal-to-noise ratio threshold of >3. Data were normalized to total protein loading, and relative abundances were calculated by dividing normalized values by the average abundance across all TMT channels per biological replicate. The MS proteomics data have been deposited in the ProteomeXchange Consortium

## Quantification of PA and PIPn

For PA and PI analysis, cells were cultured in 6-well plates at appropriate densities for each time point and treated with $0.1 \, \mu g \, ml^{-1}$ Dox or the equivalent volume of DMSO. At days 3, 5 and 7, culture medium was aspirated from each well and the cells were washed in ice-cold PBS before being quenched in 1 ml of 1 M ice-cold HCl. The cells from parallel wells were disaggregated in 1 ml of trypsin and counted to determine the cell number per well at each time point. The cells in the HCl wells were scraped and resuspended, and a volume of the cell suspension containing $7 \times 10^5$ cells was pelleted and frozen. A modified Folch method was used to extract their lipids and PA and PI were analyzed using LC–MS as previously described[67]. Technical duplicates were averaged for each clone. Data from three MP41 clones were pooled together. Data from one OMM2.5 clone was collected in three independent experiments and pooled. PA, PI, PIP and $PIP_2$ internal standards (ISDs) were used for correction. No ISDs were used for PC. PC was analyzed in a one-tenth aliquot of the prederivatization organic phase (Folch extraction). To compensate for the heterogeneity between clones and experiments, normalization by the area of 34:1 PC and normalization by the average values for the DMSO-treated samples at days 3, 5 and 7 were performed.

## Genetic suppressor screen

To identify genes whose deletion rescues lethality following *CDS2* loss, a genetic suppressor screen was performed in biological triplicate using the minimal genome-wide human CRISPR–Cas9 library (MinLibCas9) in a clonal population of MP41 Dox-inducible *CDS2* knockout cells. Infections were carried out in biological triplicate at an MOI of 0.3 and a library representation of 1,000×. Puromycin selection was performed from days 3 to 7 post-transduction. Cells were split and treated with $0.1 \, \mu g \, ml^{-1}$ of Dox or the equivalent volume of DMSO, replenished every 3 days, and maintained at a minimum representation of 500× throughout the screen for 21 days. Genomic DNA was extracted using a Gentra Puregene Cell Kit (Qiagen), PCR amplified and sequenced.

sgRNAs with less than 30 read counts across all samples were removed. To account for between-sample sequencing depth biases, total normalization was performed on raw sample counts by adding a pseudocount of 5 and normalizing to 10 million reads. $\log_2(FCs)$ were calculated from the normalized and filtered counts between DMSO-treated read counts and Dox-treated read counts. MAGeCK v0.5.9.3 was run to identify genes significantly under positive selection at an FDR < 0.05.

## In vivo assessment of *CDS2* essentiality in OMM2.5 xenografts

A total of 28 female nonobese diabetic-severe combined immunodeficient (NOD-SCID) γ mice (NOD-*Prkdc^scid^-IL2rg^Tm1^*/Rj) at 8 weeks of age were subcutaneously administered $2 \times 10^6$ OMM2.5 cells (in 0.1 ml 50:50 Matrigel/phosphate-buffered saline mix) into the left flank. The mice were fed standard chow (Safe-Lab, Safe105) and 24 days after dosing, were randomly assigned into two cohorts, with one cohort being fed a Dox diet (625 mg kg⁻¹; Envigo, TD.01306) and the other remaining on standard chow for the entirety of the study. The developing tumors were measured once per week initially. In week 3, after cohort assignment and Dox treatment, tumors were measured two to three times per week, and then four times per week for the last 8 weeks of the study. Individual mice reached an endpoint on the study if tumors reached 1.5 cm² (calculated by the longest length measurement × the longest width measurement) or lost 15% of their maximum body weight or 10% of their body weight in combination with other signs of ill health as determined by body condition assessments. Mice reaching study endpoints were humanely killed and the mass was excised, weighed and either snap frozen and stored at −80 °C or fixed using 10% neutral-buffered formalin for 24 h before storing in 70% ethanol. During the study, mice were maintained in a specific pathogen-free unit on a 12-h light/12-h dark cycle. The ambient temperature was 21 ± 2 °C, and the humidity was 55 ± 10%. Mice were housed using a stocking density

of three to five mice per cage in individually ventilated caging, receiving 60 air changes per hour. In addition to the bedding substrate, standard environmental enrichment was provided. Mice were given water and diet ad libitum.

## Sequence analysis of residue-grafted uveal melanoma cells

To sequence the transcriptome of tumors collected from the mice, we used exome capture sequencing with Agilent V5 baits to enrich for human reads and filtered the data to remove mouse reads using XenofilteR after mapping with the STAR aligner (v2.5.0c). Expression assessment used Kallisto (0.51.1) and Sleuth (0.30.1). For each sample, >50 million reads were generated.

## Statistics and reproducibility

Statistical methods are detailed in the figure legends. Unless specified, all tests were two-sided, and $P < 0.05$ was considered statistically significant. Multiple testing correction was applied where appropriate using FDR or Bonferroni correction. No data were excluded from the analyses. Mouse experiments aligned with the Animal Research: Reporting of In Vivo Experiments (ARRIVE) guidelines. Allocation was randomized.

## Reporting summary

Further information on research design is available in the Nature Portfolio Reporting Summary linked to this article.

## Data availability

The sequencing data generated as part of this study are available using the following European Nucleotide Accession numbers:

ERP151504: Paired guide (combinatorial) uveal melanoma screen data.

ERP151444: Single-guide (CRISPRko) uveal melanoma screen data.

ERP151445: Uveal melanoma suppressor screen.

ERP110320: Uveal melanoma cell line WGS.

ERP130186: Whole transcriptome sequencing of uveal melanoma cell lines.

ERP159012: Sequencing of SW837 cells following *CDS2* sgRNA transduction and selection. Transcriptome sequencing of mouse tumors.

ERP159013: Sequencing of SW837 cells following *CDS2* sgRNA transduction and selection.

Proteome data generated as part of this study are available via the PRIDE repository:

PXD053752: Targeting the CDS1/2 axis as a therapeutic strategy in uveal melanoma and pan-cancer.

TCGA data was downloaded from the Xena Browser: https://xena-browser.net/

All other data is available in the Supplementary Information. Source data are provided with this paper.

## Code availability

All code is available via Zenodo at https://doi.org/10.5281/zenodo.15124575 (ref. 101). The UVMap code is available via GitHub at https://github.com/jpark27/CDS1-2 (ref. 78).

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

## Acknowledgements

P.Y.C. was funded by a Wellcome Trust Clinician-Scientist PhD Fellowship. D.J.A. is funded by the Wellcome Trust (220540/Z/20/A), MRC (MR/V000292/1) and CR-UK (EDDPGM-Nov22/100004). We would like to thank the Sanger FACS facility, sequencing platform and cellular genetics team for making this project possible. E.G.'s work is partially supported by Portuguese national funds through FCT, Fundação para a Ciência e a Tecnologia, under projects UIDB/50021/2020 (https://doi.org/10.54499/UIDB/50021/2020), SARC-RON-AI (2024.07252.IACDC) and SYNTHESIS (LISBOA2030-FEDER-00868200). P.T.H., L.S. and D.B. were funded by the BBSRC (BB/Y006925/1 and BB/T002530/1). A.V. and C.J.R. were funded by Research Ireland (grants 20/FFP-P/8641 and 18/CRT/6214).

## Author contributions

P.Y.C., D.A., I.M., J.S.P., S.C., V.I., V.O., K.W., A.V., N.A.T., M.D.C.V.-H., L.W., J.v.D.H., E.G., S.S., M.E.V.-C., L.B., F.R., S.P., A.V., F.I. and C.J.R. performed computational analysis of the CRISPR screening data, genomes, transcriptome data and/or proteome data. P.Y.C., L.S.A.S.M., V.H., R.O.-L., F.G.A.-G. and L.v.D.W. performed screens and validation experiments. Z.K. and J.C. performed mass-spec experiments and analysis. G.B. and M.W. performed mouse experiments. D.B., P.T.H. and L.S. performed phosphoinositide and PA analyses. P.Y.C. and D.J.A. wrote the paper with input from the other authors.

## Competing interests

D.J.A. has received precompetitive funding from AstraZeneca and OpenTargets. F.I. provides consultancy services for the joint Cancer Research Horizons–AstraZeneca Functional Genomics Centre and for Mosaic Therapeutics. F.I. also receives funding from OpenTargets and Nerviano Medical Sciences Srl. The remaining authors declare no competing interests.

## Additional information

**Extended data** is available for this paper at https://doi.org/10.1038/s41588-025-02222-1.

**Correspondence and requests for materials** should be addressed to David J. Adams.

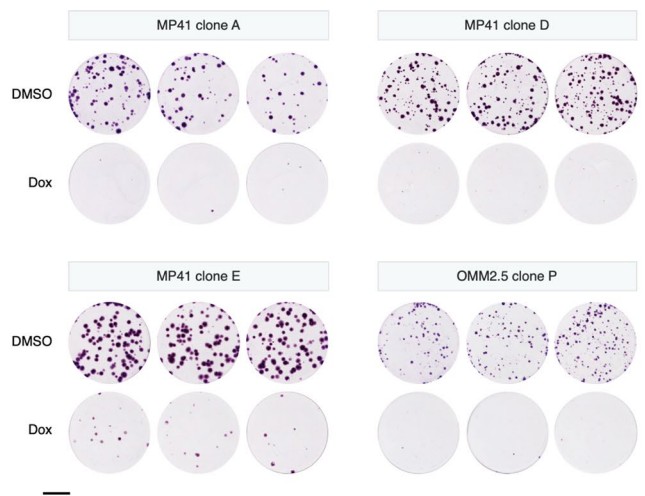

Clonal MP41-Cas9⁺ and OMM2.5-Cas9⁺ lines

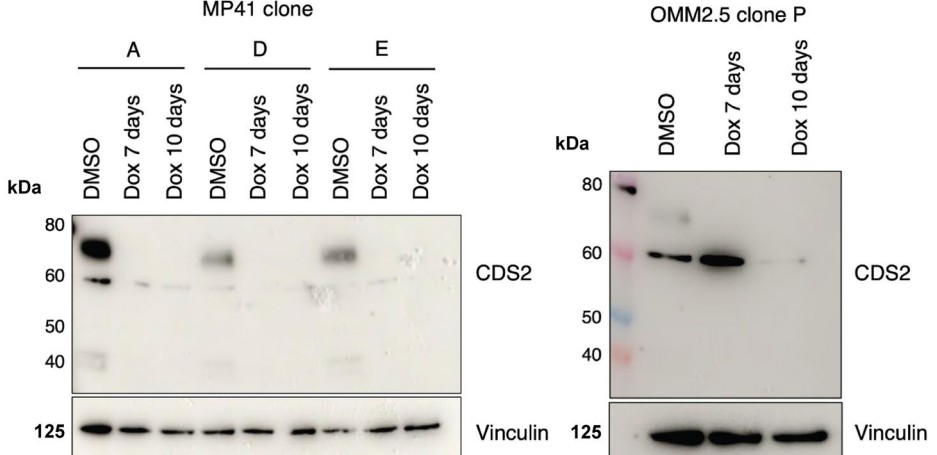

**Extended Data Fig. 1 | Effect of *CDS2* loss on colony formation and CDS2 protein expression.** Top: clonogenic assay with clones produced from uveal melanoma cells containing a doxycycline-inducible *CDS2* sgRNA construct. These data were collected from three independent experiments using different cell passages. Middle; replicate clonogenic assays using polyclonal cell lines. R refers to independent biological replicates/the experiment was performed three times independently. STG2 refers to safe-targeting sgRNA 2 (Supplementary Table 17). Cells were seeded in six-well plates for these experiments. Colony formation after 14 days of treatment with 0.1 µg ml⁻¹ doxycycline. Cells were fixed and stained with 0.1% crystal violet. Bottom: Western blots are shown which illustrate CDS2 protein depletion upon doxycycline treatment. Multiple clones of MP41 are shown (A, D, E) and one OMM2.5 clone. These blots were performed once. Days indicates the days after the addition of Dox. Vinculin was used as a loading control. Dox, doxycycline; DMSO, dimethyl sulfoxide. The higher band for CDS2 seen in MP41 has been reported previously[102].

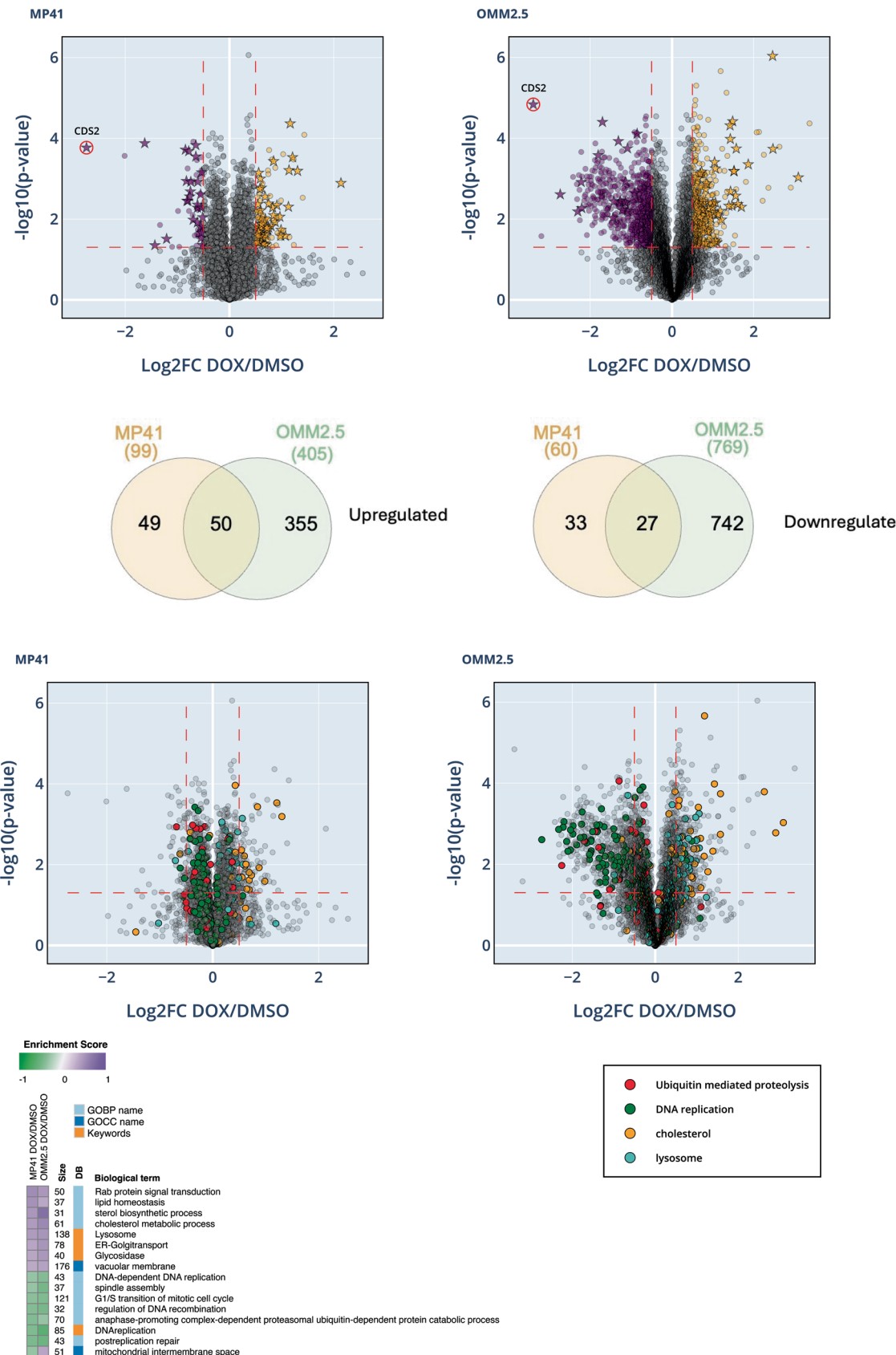

**Extended Data Fig. 2 | See next page for caption.**

**Extended Data Fig. 2 | Volcano plots for mass-spectrometry analysis following disruption of *CDS2*.** Top volcano plots showing depletion of CDS2 and other proteins in MP41 and OMM2.5 cells. These data represent analysis of three independent biological replicates (Methods). The Venn diagrams indicate the number of genes up- or downregulated in each model after Dox treatment (Supplementary Table 12). Bottom: significantly altered pathways. Proteins associated with selected enriched Gene Ontology (GO) terms are highlighted on the plots. Data visualization was performed using the Plotly package in Python. The differential expression was assessed using a paired two-sided t-test. Unadjusted *P* values < 0.05 and $\log_2$(FC) > 0.5 were considered as denoting a significant difference. Bottom right shows 1D enrichment. 1D enrichment analysis was performed using the Perseus software, which applies a two-tailed t-test with Benjamini-Hochberg (BH) correction. This method has been outlined previously[103].

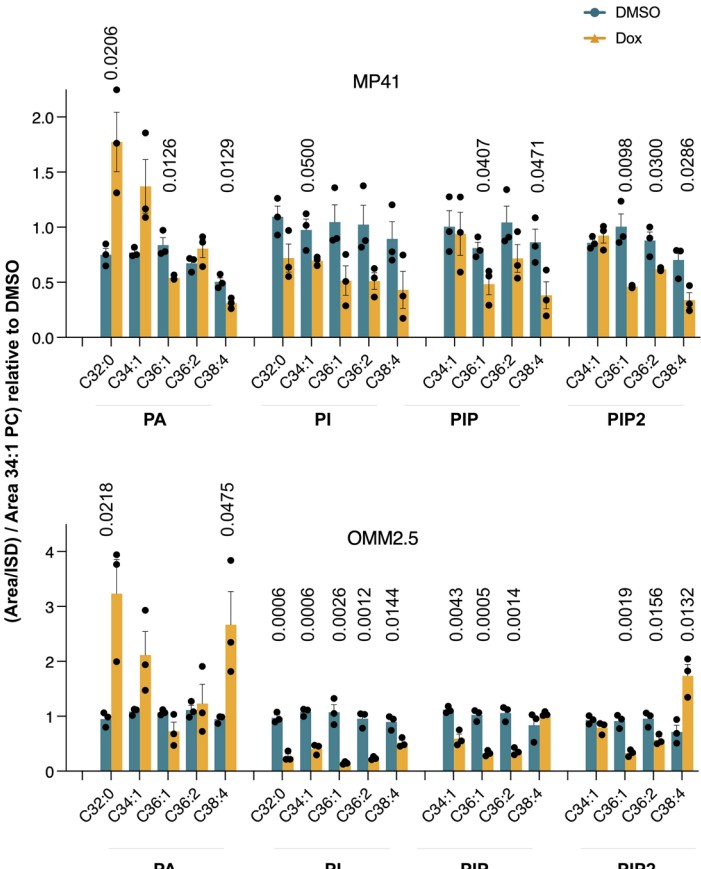

**Extended Data Fig. 3 | Effect of *CDS2* loss on PA and PIPn synthesis.** The effect of CDS2 loss following treatment with 0.1 µg ml⁻¹ of doxycycline for 7 days or the equivalent volume of DMSO (vehicle) on the synthesis of the major acyl chain species of PA and PI, PIP and PIP₂ in MP41 and OMM2.5 cells. Values were calibrated to 34:1 PC and normalized to the mean DMSO values obtained at days 3, 5 and 7. Data are represented as mean ± SEM (n = 3) from three independent MP41 clones and one OMM2.5 clone across 3 independent experiments. Significance determined from analysis of viable and combined apoptotic/dead cell populations is shown. A t-test (two-sided) was used and exact *P* values < 0.05 are shown. Dox, doxycycline; DMSO, dimethyl sulfoxide; PA, phosphatidic acid; PI, phosphatidylinositol; PIP, phosphatidylinositol monophosphate; PIP₂, phosphatidylinositol bisphosphate; PIPn, phosphoinositide.

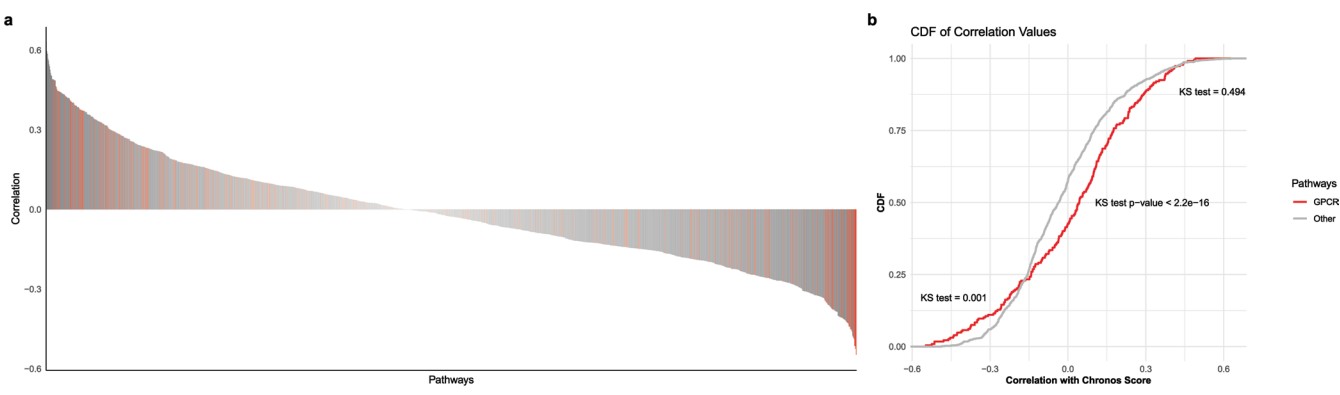

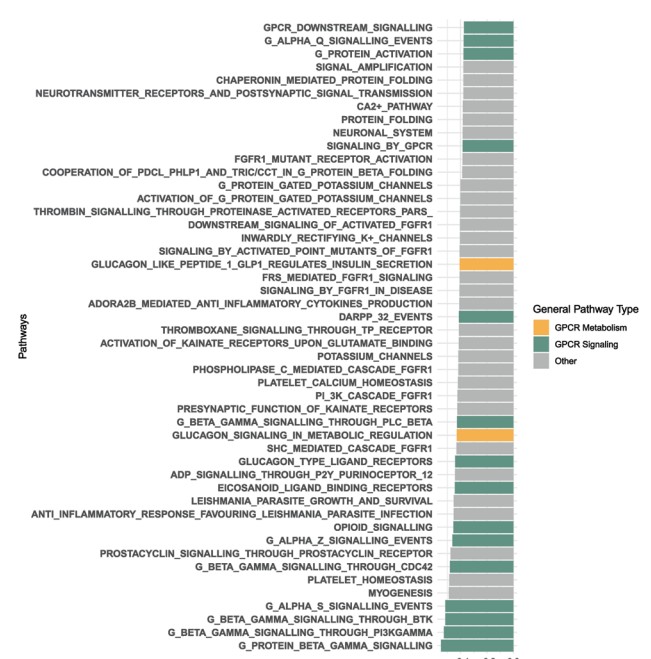

**Extended Data Fig. 4 | *CDS2* essentiality across DepMap correlates with GPCR signaling. a**, GPCR pathways, involving receptor-ligand pairs in either signal transduction or metabolism processes (red bars; ref. 104). **b**, These GPCR pathways are characterized by a significantly different cumulative distribution function (CDF) of correlation values with respect to other pathways (Kolgomorov-Smirnoff (KS) *P* value = $2.2 \times 10^{-16}$), and specifically at negative correlations (KS *P* value < 0.001). This suggests activated GPCR signaling

correlates with an increase in *CDS2* essentiality. **c**, *CDS2* Chronos/GPCR pathway most negative correlations. The correlation shown is a Spearman coefficient. As noted above, we found that multiple pathways involved in GPCR signal transduction are more frequent at negative correlation values, suggesting that higher *CDS2* essentiality (given by more negative Chronos score) is more common in those cancer cell lines displaying up-regulation of GPCR signal transduction pathways (given by higher normalized enrichment scores).

# Reporting Summary

## Statistics

For all statistical analyses, confirm that the following items are present in the figure legend, table legend, main text, or Methods section.

| n/a | Confirmed | |
|---|---|---|
| ☐ | ☒ | The exact sample size (*n*) for each experimental group/condition, given as a discrete number and unit of measurement |
| ☐ | ☒ | A statement on whether measurements were taken from distinct samples or whether the same sample was measured repeatedly |
| ☐ | ☒ | The statistical test(s) used AND whether they are one- or two-sided *Only common tests should be described solely by name; describe more complex techniques in the Methods section.* |
| ☐ | ☒ | A description of all covariates tested |
| ☐ | ☒ | A description of any assumptions or corrections, such as tests of normality and adjustment for multiple comparisons |
| ☐ | ☒ | A full description of the statistical parameters including central tendency (e.g. means) or other basic estimates (e.g. regression coefficient) AND variation (e.g. standard deviation) or associated estimates of uncertainty (e.g. confidence intervals) |
| ☐ | ☒ | For null hypothesis testing, the test statistic (e.g. *F*, *t*, *r*) with confidence intervals, effect sizes, degrees of freedom and *P* value noted *Give P values as exact values whenever suitable.* |
| ☒ | ☐ | For Bayesian analysis, information on the choice of priors and Markov chain Monte Carlo settings |
| ☒ | ☐ | For hierarchical and complex designs, identification of the appropriate level for tests and full reporting of outcomes |
| ☐ | ☒ | Estimates of effect sizes (e.g. Cohen's *d*, Pearson's *r*), indicating how they were calculated |

*Our web collection on statistics for biologists contains articles on many of the points above.*

## Software and code

Policy information about availability of computer code

| Data collection | No software was used for data collection, except for proprietary Illumina base-calling software on HiSeq platforms. |
|---|---|
| Data analysis | SAMtools v1.10 was used to convert BAM files to FASTQ files. QC was performed using FastQC v0.11.8. Reads were aligned to GRCh38 using BWA mem v0.7.17. GATK v4.4.0.0 was used for germline short variant calling (SNVs, indels) with the HaplotypeCaller. <br><br> For transcriptome data - Reads were mapped using STAR v2.5.0c against the GRCh38 human reference genome with the ERCC spike-in sequences and ENSEMBL v103 gene annotation. To assess gene expression reads were counted using HTSeq-count v0.7.2 .XenofilteR after mapping with the STAR aligner. Expression assessment used Kallisto (0.51.1) and Sleuth (0.30.1). <br><br> CRISPR analysis <br> All samples were processed using CRISPRcleanR v3.0.159 <br> C-SAR v1.3.6 (https://github.com/cancerit/C-SAR) <br> MAGeCK MLE v0.5.9.5 <br><br> Single cell analysis <br> GBmap pipeline approach from Ruiz-Moreno et al. <br> STARsolo v2.7.10a pipeline <br> semi-supervised neural network model called single-cell ANnotation using Variational Inference (scANVI), within the transfer-learning framework of the single-cell architectural surgery algorithm (scArches). <br><br> Mass-spec |

Proteome Discoverer 3.0 (Thermo Scientific) was used with SequestHT and Comet search engines for protein identification and quantification. Peptide confidence was estimated using Percolator, maintaining a false discovery rate (FDR) of 0.01 with target-decoy database validation.

Custom Code is available here: https://github.com/team113sanger/Targeting-the-CDS1-2-axis-as-a-therapeutic-strategy-in-uveal-melanoma-and-pan-cancer

For manuscripts utilizing custom algorithms or software that are central to the research but not yet described in published literature, software must be made available to editors and reviewers. We strongly encourage code deposition in a community repository (e.g. GitHub). See the Nature Portfolio guidelines for submitting code & software for further information.

## Data

Policy information about availability of data

All manuscripts must include a data availability statement. This statement should provide the following information, where applicable:

- Accession codes, unique identifiers, or web links for publicly available datasets
- A description of any restrictions on data availability
- For clinical datasets or third party data, please ensure that the statement adheres to our policy

Code Availability Statement

Github:

CRISPR Screen Analysis: https://github.com/team113sanger/Targeting-the-CDS1-2-axis-as-a-therapeutic-strategy-in-uveal-melanoma-and-pan-cancer

UVMap Data: https://github.com/jpark27/CDS1-2

All figure code is freely available for download: https://doi.org/10.5281/zenodo.15025721

Figshare: https://figshare.com/account/home#/projects/184459

Data Accessibility Statement
The sequencing data generated as part of this study is available using the following European Nucleotide Accession numbers:
ERP151504: Paired guide (combinatorial) uveal melanoma screen data.
ERP151444: Single guide (CRISPRko) uveal melanoma screen data.
ERP151445: Uveal melanoma suppressor screen.
ERP110320: Uveal melanoma cell line WGS.
ERP130186: Whole transcriptome sequencing of uveal melanoma cell lines.
ERP159012: Sequencing of SW837 cells following CDS2 gRNA transduction and selection. Transcriptome sequencing of mouse tumors.
ERP159013: Sequencing of SW837 cells following CDS2 gRNA transduction and selection.

Proteome data generated as part of this study is available via the PRIDE repository:
PXD053752: Targeting the CDS1/2 axis as a therapeutic strategy in uveal melanoma and pan-cancer.
TCGA data was downloaded from the Xena Browser: https://xenabrowser.net/
All other data is in the supplementary information.

## Research involving human participants, their data, or biological material

Policy information about studies with human participants or human data. See also policy information about sex, gender (identity/presentation), and sexual orientation and race, ethnicity and racism.

| | |
|---|---|
| Reporting on sex and gender | N/A |
| Reporting on race, ethnicity, or other socially relevant groupings | N/A |
| Population characteristics | N/A |
| Recruitment | N/A |
| Ethics oversight | N/A |

Note that full information on the approval of the study protocol must also be provided in the manuscript.

# Field-specific reporting

Please select the one below that is the best fit for your research. If you are not sure, read the appropriate sections before making your selection.

☒ Life sciences  ☐ Behavioural & social sciences  ☐ Ecological, evolutionary & environmental sciences

For a reference copy of the document with all sections, see nature.com/documents/nr-reporting-summary-flat.pdf

# Life sciences study design

All studies must disclose on these points even when the disclosure is negative.

| Sample size | No explicit sample size was decided upon a priori, rather we used standardized workflows for CRISPR screening and analysis. For example the CRISPR screening approach we used was that used by DepMAP/ProjectScore (PMID: 33712601) i.e. 3 replicates per cell line. For the mouse tumour growth experiments we defined significance using a Mixed-effects model with the Geisser-Greenhouse correction. With 16 mice per group and with at least 5 measurements per mouse/tumour we estimate power of ~70% to detect an effect of genotype on tumour growth. For all other experiments we either used all available data to make comparisons (for example all CRISPR screened cell lines) or independently replicated the experiments at least 3 times and an alpha value of 0.05. |
|---|---|
| Data exclusions | Only QC failed data was excluded and these parameters are clearly outlined in the paper. For example if NNMD values for CRISPR screens were not met or insufficient sequence data was generated. |
| Replication | All experiments where statistics were applied were repeated at least three times independently as indicated in the figure legend. This included separate days of transfection and distinct cell cultures. All replication attempts were successful. |
| Randomization | For the mouse experiments shown in figure 6 mice were randomised for tumour growth studies with an equal number of animals being assigned to each group. For CRISPR validation experiments gRNAs were randomly assigned to wells/cultures to avoid any biases. CRISPR screens were performed in pools with random virus infection so these experiments were randomised internally. |
| Blinding | Blinding was not performed but all key experiments were replicated by independent individuals in the lab. The CRISPR screens were performed en masse so there was no selection for screen outcomes. |

# Reporting for specific materials, systems and methods

We require information from authors about some types of materials, experimental systems and methods used in many studies. Here, indicate whether each material, system or method listed is relevant to your study. If you are not sure if a list item applies to your research, read the appropriate section before selecting a response.

## Materials & experimental systems

| n/a | Involved in the study |
|---|---|
| ☐ | ☒ Antibodies |
| ☐ | ☒ Eukaryotic cell lines |
| ☒ | ☐ Palaeontology and archaeology |
| ☐ | ☒ Animals and other organisms |
| ☒ | ☐ Clinical data |
| ☒ | ☐ Dual use research of concern |
| ☒ | ☐ Plants |

## Methods

| n/a | Involved in the study |
|---|---|
| ☒ | ☐ ChIP-seq |
| ☐ | ☒ Flow cytometry |
| ☒ | ☐ MRI-based neuroimaging |

## Antibodies

| Antibodies used | anti-CDS2 antibody (Proteintech 13175-1-AP, 1:1000 dilution) and an anti-vinculin antibody (ThermoFisher MA5-11690, 1:10,000 dilution) |
|---|---|
| Validation | The CDS2 antibody was validated by western blotting of KO cells. i.e. to show protein loss. The vinculin antibody has been extensively validated by the manufacturer using null cell lines (https://www.thermofisher.com/antibody/product/Vinculin-Antibody-clone-VLN01-Monoclonal/MA5-11690). |

## Eukaryotic cell lines

Policy information about cell lines and Sex and Gender in Research

| Cell line source(s) | These are stated in Supplementary Table 1 and include ATCC, ECACC and the University of Liverpool Ocular melanoma tisseu bank. |
|---|---|
| Authentication | STR profiling was performed on all lines |
| Mycoplasma contamination | All lines were screened and found negative for mycoplasma. The were also routinely tested. |
| Commonly misidentified lines (See ICLAC register) | No commonly misidentified lines were used in this study. |

# Animals and other research organisms

Policy information about studies involving animals; ARRIVE guidelines recommended for reporting animal research, and Sex and Gender in Research

| | |
|---|---|
| Laboratory animals | NOD-Prkdcscid-IL2rgTm1/Rj background. 8 weeks. Female mice. Mice were maintained in a specific pathogen-free unit on a 12h light:12h dark cycle. The ambient temperature is 21±2°C, and the humidity is 55±10%. Mice were housed using a stocking density of 3–5 mice per cage (overall dimensions of caging: 365×207×140mm3 (length×width×height), floor area 530cm2) in individually ventilated caging receiving 60 air changes per hour. In addition to Aspen bedding substrate, standard environmental enrichment of two Nestlets, a cardboard fun tunnel, and three wooden chew blocks are provided. Mice were given water and diet ad libitum. |
| Wild animals | No wild animals were used in the study. |
| Reporting on sex | Yes - we only used female mice. |
| Field-collected samples | No field collected samples were used in the study. |
| Ethics oversight | The care and use of all mice in this study were in accordance with the UK Animals in Science Regulation Unit's Code of Practice for the Housing and Care of Animals Bred, Supplied or Used for Scientific Purposes, the Animals (Scientific Procedures) Act 1986, and all procedures were performed under a UK Home Office Project license (PP8090463), which was reviewed and approved by the University of Cambridge Animal Welfare and Ethical Review Body. |

Note that full information on the approval of the study protocol must also be provided in the manuscript.

# Plants

| | |
|---|---|
| Seed stocks | *Report on the source of all seed stocks or other plant material used. If applicable, state the seed stock centre and catalogue number. If plant specimens were collected from the field, describe the collection location, date and sampling procedures.* |
| Novel plant genotypes | *Describe the methods by which all novel plant genotypes were produced. This includes those generated by transgenic approaches, gene editing, chemical/radiation-based mutagenesis and hybridization. For transgenic lines, describe the transformation method, the number of independent lines analyzed and the generation upon which experiments were performed. For gene-edited lines, describe the editor used, the endogenous sequence targeted for editing, the targeting guide RNA sequence (if applicable) and how the editor was applied.* |
| Authentication | *Describe any authentication procedures for each seed stock used or novel genotype generated. Describe any experiments used to assess the effect of a mutation and, where applicable, how potential secondary effects (e.g. second site T-DNA insertions, mosiacism, off-target gene editing) were examined.* |

# Flow Cytometry

## Plots

Confirm that:

☒ The axis labels state the marker and fluorochrome used (e.g. CD4-FITC).

☒ The axis scales are clearly visible. Include numbers along axes only for bottom left plot of group (a 'group' is an analysis of identical markers).

☒ All plots are contour plots with outliers or pseudocolor plots.

☒ A numerical value for number of cells or percentage (with statistics) is provided.

## Methodology

| | |
|---|---|
| Sample preparation | As above cell lines were used for analysis. These cells were simply dissociated in trypsin, resuspended in PBS and then subjected to FACS. GFP/BFP was excited with 488nm lazers. |
| Instrument | CytoFLEX flow cytometer (Beckman Coulter) and analysed with FCS Express v7.22.0031 |
| Software | FCS Express v7.22.0031 |
| Cell population abundance | Cell populations were always higher than 10% making for easy and accurate quantification. |
| Gating strategy | This is provided in Supplementary Fig 6. We used uninfected cells to establish our control gates. First, we selected cells based on their forward scatter and side scatter properties to exclude dead cells and debris. Next, we gated for singlets and excluded doublets by analysing the FSC-W vs. FSC-A plot. The fluorescence of BFP and GFP cell populations was detected following excitation with the 405 nm and 488 nm lasers, respectively. |

☒ Tick this box to confirm that a figure exemplifying the gating strategy is provided in the Supplementary Information.

