## [Peer Review File · Nature Genetics]

The synthetic lethal interaction between CDS1 and CDS2 is targetable across multiple tumor types

Corresponding Author: Dr David Adams

Version 0:

Decision Letter:

23rd Jan 2024

Dear Dr Adams,

First, please accept my apologies for the delay in returning this decision to you. Thank you for bearing with me.

Your Letter, "Targeting the CDS1/2 axis as a therapeutic strategy in uveal melanoma and pan-cancer" has now been seen by 4 referees. Please note that reviewers #2,3 reviewed together. Reviewer #4 has signed their report. You will see from their comments copied below that while they find your work of considerable potential interest, they have raised quite substantial concerns that must be addressed. In light of these comments, we cannot accept the manuscript for publication, but would be very interested in considering a revised version that addresses these serious concerns.

We hope you will find the referees' comments useful as you decide how to proceed. If you wish to submit a substantially revised manuscript, please bear in mind that we will be reluctant to approach the referees again in the absence of major revisions.

To guide the scope of the revisions, the editors discuss the referee reports in detail within the team, including with the chief editor, with a view to identifying key priorities that should be addressed in revision and sometimes overruling referee requests that are deemed beyond the scope of the current study. We hope that you will find the prioritised set of referee points to be useful when revising your study. Please do not hesitate to get in touch if you would like to discuss these issues further.

If you choose to revise your manuscript taking into account all reviewer and editor comments, please highlight all changes in the manuscript text file. At this stage we will need you to upload a copy of the manuscript in MS Word .docx or similar editable format.

*2) If you have not done so already please begin to revise your manuscript so that it conforms to our Letter format instructions, available here. Refer also to any guidelines provided in this letter.

*3) Include a revised version of any required Reporting Summary: <https://www.nature.com/documents/nr-reporting-summary.pdf>

Please be aware of our guidelines on

digital image standards.

Link Redacted

If you wish to submit a suitably revised manuscript we would hope to receive it within 6 months. If you cannot send it within this time, please let us know. We will be happy to consider your revision so long as nothing similar has been accepted for publication at Nature Genetics or published elsewhere. Should your manuscript be substantially delayed without notifying us in advance and your article is eventually published, the received date would be that of the revised, not the original, version.

Nature Genetics is committed to improving transparency in authorship. As part of our efforts in this direction, we are now requesting that all authors identified as 'corresponding author' on published papers create and link their Open Researcher and Contributor Identifier (ORCID) with their account on the Manuscript Tracking System (MTS), prior to acceptance. ORCID helps the scientific community achieve unambiguous attribution of all scholarly contributions. You can create and link your ORCID from the home page of the MTS by clicking on 'Modify my Springer Nature account'. For more information please visit please visit www.springernature.com/orcid.

Thank you for the opportunity to review your work.

Sincerely,

Safia Danovi
Editor
Nature Genetics

Referee expertise:

Referee #1: functional genomics, SL, cancer

Referee #2,3: functional genomics, SL, cancer

Referee #4: functional genomics

Reviewers' Comments:

Reviewer #1:

Remarks to the Author:

Overall, this work is a quite interesting study that seems like a major advance in the uveal melanoma field by adding in these new models and generating data sets for further characterizations. Impressive they seemed to find a pan-uveal melanoma target in CDS2 – curious why CDS1 is so low in these cells/ generally across cell lines – authors don't discuss this much.

Major comments:

1. Authors note at the beginning of results that there are few uveal melanoma models but then are able to source about 10 cell line models, some of which also appear to be represented in DepMap. DepMap appears to have about 12. So is it true that "none have been comprehensively characterized" as the authors state?
2. Figure 2d – It is mentioned in the text that there are negative correlations between gene expression of CDS2 and paralog and INTS6 and paralog, but only INTS6 data is shown, no data from CDS2. Should at least add to supplement, or maybe show CDS1/2 here because its followed up on and add INTS6 to supplement
3. Authors should provide more information for the 10 melanoma cell lines in terms of paralog expression? No RNA-seq or even qPCR based expression testing for the CDS1 and others. Cell lines are referred to being low CDS1 expressers, but there is no data to support this (generated by authors or indicated from DepMap data).
4. Figure 4d. Authors chose ITBG5 and ZEB2 as controls due to some cell lines not requiring these genes, while CDS2 is required in all cell lines tested. But, extending the assay to 28d shows requirement for these genes. So, are they pan essential in the uveal melanoma lines? Or, were the three lines selected chosen because requirement for these genes was demonstrated in the sgRNA CRISPR screen? It is unclear why these were chosen.
5. Figure 4D - Further, major activity differences between CDS2 gRNAs is evident in the data represented. Is there any information on gRNA activity – western blot, sanger sequencing to understand this discrepancy better? Was the more potent gRNA based on phenotype moved forward to the remaining assays using CDS2 depletion? Authors should comment on the specificity of the gRNAs used in their screen and in validation for each paralog. It is possible that some gRNAs would target

both paralogs.

6. Figure 4D –the experiment should also be performed with safe-targeting or non-targeting guides as control
7. Figure 5A – it would be helpful to understand the relative levels of both CDS1 and CDS2 in each tumor type (currently the figure shows only CDS1)
8. Figure 6A. Again, there is no data to represent the editing activity or specificity of this gRNA system. It is unclear if the gRNA is definitely cutting CDS2 or also has activity against CDS1
9. Figure 6A - Further, DMSO is not the appropriate control for this assay. A safe targeting gRNA should have been used in the same dox-inducible system to generate an appropriate control, both for dox treatment as well as gRNA cutting/ damage to the DNA induced by this. This is also missing from the in vivo study, and a limitation of the findings. Further, authors say the colonies are both fewer and smaller, but there is no quantification of this represented, or statistical analyses.
10. The in vivo study in Fig 6g would be strengthened by showing that re-expression of CDS2 of CDS1 can rescue this effect
11. Bottom of page 19. Authors refer to this suppressor screen, but don't present any data from it beyond data in a supplement and also don't describe much of the experimental method (timing?, how was coverage maintained CDS2 depletion is fully lethal at 14d but screen went for 21d?). What if FDR is less stringent? Authors go on to say their lack of findings suggests that there may be few resistance mechanisms to treatment with a CDS2 inhibitor, but this is a shallow analysis and these genetic based screens do not recapitulate what would happen with an inhibitor (ie direct selection of mutations that bypass inhibitor binding, etc).
12. Supplementary Figs 6-7 – The MP46 screen is clearly an outlier in terms of quality. How did authors determine whether the screen is of sufficient performance to include results?
13. Methods
 - o – please provide more information on the WGS including coverage, criteria for variant calling, and exclusion of germline variants
 - o Please discuss in more detail the source and characteristics of the uveal melanoma cell lines

Minor comments/text issues:

14. It could be helpful to the reader to note the genomic position of CDS1 and CDS2 in the human genome
15. On Page 7, the authors state that their analysis “revealed significantly lower log2FCs within the double knockout gene pair group” – is this referring to the data in Fig 1D? If so a figure callout can be added for clarity. If not, a supplemental figure could be added to show this
16. Figure 4A – It is a bit counter intuitive to show the x axis with the more-essential in melanoma genes on the right with positive LFC. Minor suggestion to flip the sign of the LFC comparison.
17. Figure 6D: Labelling the images with what fluor is being used (DAPI, BIODIPY) will help readers
18. Supplementary Fig 11 – the text is not legible
19. The authors allude to targeting CDS2 but provide no discussion around how targetable this protein is or if other classes of these enzymes have drugs/inhibitors in development or as tool compounds – some discussion of feasibility of this would be nice.

Reviewer #2:

None

Reviewer #3:

Remarks to the Author:

In this study, the authors have aimed to identify novel therapeutic targets in uveal melanoma (UVM). They have employed an extensive characterization of 10 UVM cell lines and utilized a small ~570 gene-pair combinatorial CRISPR library in addition to a whole-genome CRISPR library for screening. The manuscript's key observation is the inverse correlation between sensitivity to CDS2 knockout (KO) and CDS1 expression levels in cancer cells, suggesting functional redundancy between these two paralogues. Based on this observation, the authors pose CDS2 as a potential therapeutic target in those cancers, such as UVM. While the study represents an extensive body work, it has several notable caveats.

The observation that CDS1 and CDS2 are potentially functionally redundant paralogue pairs is partially novel, although this has previously been suggested through analysis of existing whole-genome single-gene CRISPR KO screens (PMID: 35417719). However, the central claim of the manuscript that CDS2 is a therapeutic target in UVM is not adequately supported by data.

Specifically, the authors identify (1) low CDS1 expression in various cancer cell line lineages other than UVM as well as (2) significant enrichment of CDS1 expression in tumors compared to normal tissues (Fig. 5a, Sup. Fig. 15). Together these data suggest broad toxicity concerns for targeting CDS2 in UVM, which the authors briefly mention but do not explore. The authors' hypothesis that mutations contributing to increased GPCR signaling could lead to higher CDS2 dependency in UVM warrants extensive validation. For instance, sensitivity comparisons to CDS2 KO in other low CDS1-expressing cell lines from different lineages (i.e. blood and liver), and the impact of altering GPCR and/or PLC signaling, should be demonstrated.

The manuscript also lacks important rescue experiments and several detailed methodological descriptions:

- Functional redundancy of CDS1/2 should be shown through rescue experiments, such as using exogenous CDS1 expression to compensate for CDS2-KO in UVM cells.

- The mechanism of CDS2-KO mediated cell death is unclear. Rescue experiments involving supplementation of phosphoinositides or depletion of intracellular triacylglycerol accumulation could provide insights.
- The CRISPR screen quality assessment is insufficient (Sup. Figs 8-9). Separation metrics of pan-essential and non-essential genes should be shown (e.g., SSMD and NNMD scores). In addition, benchmarking of these screens against previous published screens would add context to the screening performance.
- Details on sgRNA activities from the hU6 and mU6 promoters in the combinatorial screens are missing (showing whether both sides were balanced in their ability to achieve the desired phenotypic effect). In addition, it is unclear from Sup. Table 13 which specific alternative tracrRNA sequences used for sgRNA expression under the hU6 and mU6 promoters, and this should be clearly shown in the (Sup.) text or figures.
- It is unclear how the 72 gene hits in UVM cells were defined; if a $\text{Log}_2\text{FC} > 1.5$ and $\text{P-adj} < 0.01$ was applied to define this, this should be clearly stated in the main text as well as Fig. 4a, and Fig. 4a should be corrected to reflect this cutoff.

In conclusion, the manuscript contributes valuable data towards understanding the genetic dependencies in UVM. The data presented fall short of establishing the necessary specificity and safety profile required for a viable therapeutic strategy in targeting CDS2, particularly given the potential for broad toxicity. To advance this hypothesis, comprehensive validation through additional well-designed experiments and more rigorous data analysis is crucial. The inclusion of these elements would greatly enhance the robustness and credibility of the proposed therapeutic implications.

Reviewer #4:

Remarks to the Author:

Metastatic uveal melanoma can be devastating to patients and new therapies are desperately needed. In this study the authors use a functional genomic approach to investigate essential genes and synthetic lethal (SL) interactions across a panel of uveal melanoma cell lines. The authors focus on one SL interaction, CDS1/2, and establish functional relevance in vitro and in vivo. The design and screening approaches are well described and effectively applied in this system, and appear to reveal new molecular insight into molecular mechanisms controlling uveal melanoma. Overall the study is original and effectively uses powerful and new functional genomics techniques. CDS2 is convincingly shown to be required for uveal melanoma, given the importance of this pathway, its not clear if it is also essential for many other cancers, or healthy cells, but surely with this knowledge strategies can be developed to generate novel chemotherapies with some level of selectivity.

I have included major and minor comments below that may help improve this manuscript.

Major comments

1. Concerning the uveal melanoma cell lines used in this study, its not clear how removed they are from primary patient material, and from this what changes reflect bona fide cancer signatures vs changes that have accumulated from in vitro propagation. How do these genomic and protein signatures compare to those from primary tumour material? Use of fresh patient derived cell lines that recapitulate synthetic lethal results, at least for CDS1/2, would strengthen the confidence in this approach.
2. An assumption with this work is that the screening results will specifically inform on the biology of uveal melanoma. However by focusing on pairs that were shared among at least 6/10 cell lines screened, this could actually favour general (vs. uveal melanoma) SL combinations. To confirm there is a cancer type molecular specificity to this approach, one could also screen non-uveal melanoma cell lines and show there is a lack of correlation for SL hits. Alternatively, if sufficient data exists, comparing the current SL hits (or imputed SL hits found in single KO screens + RNA seq etc) with previous screens of cells from different cancer origins could also be informative, with either general or cancer specific SL interactions being interesting. To some extent this type of comparison is included in fig 3 and 5, but for slightly different purposes.
3. The inclusion of single gene KO screening is useful, but in some way disrupts the study flow. This is primarily b/c CDS2 on its own is lethal, since CDS1 expression is reduced in these cells and presumably there is a SL-like interaction. This complicates the study and makes the SL screen component seem less significant, since really screening single KO libraries and comparing essential genes with cell expression profiles could reveal the same SL interactions at the genome level, and from a cancer therapeutic perspective a single target may still be sufficient depending on gene expression.
4. While CDS2 essentiality in a low CDS1 expressing cell is presumably via a CDS1/2 SL interaction, other more indirect mechanisms are also possible. These competing indirect mechanisms could be ruled out by inducing CDS1 ectopic expression in CDS2 KO lethal lines. i.e. can one rescue lethality with dox inducible CDS1?
5. The genetic suppressor screen is interesting, however loss of function is only one way that cells could change to circumvent CDS2 dependence, and gain of function changes could also help cells escape from loss of CDS2, so its not clear if one can claim targeting CDS2 therapeutically would not lead to cells that may become resistant through additional changes.
6. From my experience, 3 biological replicates performed in a single experiment (e.g. Fig 6b) does not always lead to definitive results, and to be certain, reproduction of results through multiple independent experiments performed on different days is more robust.

7. The in vivo experiments are compelling, can you evaluate KO efficiency or gene expression in the surviving tumors over time? Presumably these cell have intact CDS2? Or have they upregulated CDS1 or other factors? Optimally these data could also be confirmed with xenografts if such systems are available.

Minor comments

Fig 1a. Its not clear in the text where the 1061 single genes came from. Presumably these are guides targeting only one of each paralog or synthetic lethal gene pairs? This could be clarified in the text.

Fig 1c. If there are 8 individual guides per gene and two positions, shouldn't this be 64 combinations? It seems that only 4 of the 8 are selected for each position, but the rationale for this wasn't completely clear and could be further clarified for the general readership?

Greg Neely

Version 1:

Decision Letter:

25th Feb 2025

Dear Dr Adams,

Please accept my apologies for the delay in returning this decision to you. Thank you for your patience.

Your Letter, "Targeting the CDS1/2 axis as a therapeutic strategy in uveal melanoma and pan-cancer" has now been seen by your original referees (as before, Reviewers #3,#4 reviewed together). You will see from their comments below that while they find your work of interest, some important points are raised. We are interested in the possibility of publishing your study in Nature Genetics, but would like to consider your response to these concerns in the form of a revised manuscript before we make a final decision on publication.

We therefore invite you to revise your manuscript taking into account all reviewer and editor comments. Our plan is to assess your revisions in-house, so we'll only return to Reviewers #3,#4 if absolutely necessary.

Please highlight all changes in the manuscript text file. At this stage we will need you to upload a copy of the manuscript in MS Word .docx or similar editable format.

*2) If you have not done so already please begin to revise your manuscript so that it conforms to our Letter format instructions, available

http://www.nature.com/ng/authors/article_types/index.html here

*3) Include a revised version of any required Reporting Summary: <https://www.nature.com/documents/hr-reporting-summary.pdf>

Please be aware of our <https://www.nature.com/nature-research/editorial-policies/image-integrity> guidelines on digital image standards.

EXTENDED DATA FIGURES

When re-submitting your manuscript, please ensure that any supplementary figures and tables that are crucial to the manuscript's conclusions are converted into Extended Data figures and tables to increase visibility of these data. Extended

Data figures and tables are online-only (present in the online PDF and full-text HTML versions of the paper), peer-reviewed display items that provide essential background to the article but are not included in the main article due to space constraints. A maximum of ten Extended Data display items (figures and tables) is permitted.

Link Redacted

We hope to receive your revised manuscript within four to eight weeks. If you cannot send it within this time, please let us know.

Nature Genetics is committed to improving transparency in authorship. As part of our efforts in this direction, we are now requesting that all authors identified as 'corresponding author' on published papers create and link their Open Researcher and Contributor Identifier (ORCID) with their account on the Manuscript Tracking System (MTS), prior to acceptance. ORCID helps the scientific community achieve unambiguous attribution of all scholarly contributions. You can create and link your ORCID from the home page of the MTS by clicking on 'Modify my Springer Nature account'. For more information please visit please visit www.springernature.com/orcid.

Sincerely,

Safia Danovi, PhD
Senior Editor, Nature Genetics
ORCID: 0009-0007-7822-5479

Referee expertise:

Referee #1:

Referee #2:

Referee #3:

Reviewers' Comments:

Reviewer #1 (Remarks to the Author):

The authors have thoughtfully addressed my prior concerns. In particular the off-target analysis, rescue experiment, and additional methodological details are appreciated. Congratulations!

Reviewer #1 (Remarks on code availability):

The github repo appears to be a great resource.

Reviewer #3 (Remarks to the Author):

In this study, the authors aim to identify novel therapeutic targets in uveal melanoma (UVM). The manuscript combines data from 10 UVM cell lines, including a small ~570 gene-pair combinatorial CRISPR library, whole-genome CRISPR screens, RNA-seq, and proteomics. The study highlights the inverse correlation between CDS2 knockout sensitivity and CDS1 expression, suggesting functional redundancy between these paralogues and proposing CDS2 as a therapeutic target in cancers such as UVM.

This manuscript has been co-submitted with a complementary study by Arnoldus et al (Daniel Peeper lab), which leverages DepMap and TCGA data with experimental validation to propose CDS2 as a therapeutic target in mesenchymal-like cancers. Both manuscripts were reviewed simultaneously.

The authors have made substantial revisions and incorporated new experimental data in response to reviewer feedback, effectively addressing many initial concerns. However, several critical questions remain unresolved. Below, a detailed

analysis is provided along with key areas of the manuscript requiring amendments or additional data for clarification and improvement.

1. Proposed novelty and contribution to the field

The manuscript provides valuable data on UVM, including combinatorial and single-gene CRISPR KO dependency data, RNA-seq, and proteomics datasets, while also characterizing the CDS1/CDS2 synthetic lethal interaction (SLI). However, the usefulness of these datasets in identifying the CDS1/CDS2 SLI is limited, as this interaction could also be deduced from existing single-gene CRISPR screens, as demonstrated by the Arnoldus et al co-submitted paper. Moreover, several prior studies, including PMID: 35417719, PMID: 35194081, PMID: 34469736, and the authors' own prior work (PMID: 33637726), have reported or inferred this SLI. While the authors argue that these findings are obscure, explicit acknowledgment of these prior works in the manuscript is fundamentally essential.

2. CDS2 as a therapeutic target in UVM

We appreciate the additional analysis supporting a therapeutic window for CDS2 in UVM, specifically the correlation between GPCR signaling and CDS2 essentiality, and the inclusion of references showing that Cds2 deletion in mice is well-tolerated in liver and blood. However, the manuscript does not adequately address how their findings relate to the co-submitted paper by Arnoldus et al. The authors should clarify whether their data suggest that UVM represents a particularly favorable or distinct context for CDS2 dependency, potentially due to its biology, or if CDS2 dependency in UVM aligns with the broader mesenchymal cancer subtype described by Arnoldus et al. Establishing this link or distinction is critical for readers to fully contextualize CDS2 as a therapeutic target in UVM and other cancers.

3. Combinatorial CRISPR screen quality assessment

The authors have included extensive additional metrics, such as NNMD, SSMD, ROC curves, and Cas9 activity scores, to support the quality of their screens. However, some critical caveats remain unaddressed:

- In Sup. Fig. 5d, sgRNA activity under hU6 and mU6 promoters is clearly not balanced. As shown in Fig. 1c, each unique sgRNA is expressed by only one promoter, either hU6 or mU6, but never both. This raises questions about whether the authors could distinguish between effects arising from true sgRNA activity/quality and those influenced by promoter positioning. The authors should explain how they accounted for this potential confounding factor in their analysis.

- Also in in Sup. Fig. 5d, a group of non-essential genes (and essential genes) scores strongly in both promoter positions (LFC -5.0 to -7.5). Could this result from unintended activity, such as off-target effects or an STG (Safe Targeting sgRNA) that is not truly safe targeting? The authors should describe how such artifacts were identified and mitigated (e.g. how sgRNAs paired with this "bad" STG were handled in downstream analyses). These points are currently absent in the text and need to be addressed.

4. Low CDS1 expression in UVM and other Cancers

We appreciate the inclusion of the CDS1 rescue experiment (Fig. 5f), which demonstrates that ectopic CDS1 expression can partially rescue lethality associated with CDS2 loss. However, the observed toxicity of CDS1 expression in OMM2.5 cells raises questions about its role in tumor biology. The authors should discuss why certain cancers might downregulate CDS1, and provide a hypothesis for why CDS1 expression might be detrimental to these cells.

Minor Comments and Corrections

Sup. Fig. 5C : There is an inconsistency in the tracrRNA sequence reported in Sup. Fig. 5C (5'-gttcagagctatgctggaaacagcatagcaagttgaaataaggcta-3') versus Sup. Table 16 and Fig. R20 (5'-gttcagagctatgctggaaactgcatagcaagttgaaataaggcta-3'). Please confirm and update the correct sequence in all relevant figures, tables, and methods.

Line 140: The term "pgRNA" is introduced without definition. Please define this term in the text.

Figure 5a: The figure and its legends are too small and difficult to read. Enlarging and reformatting for clarity would greatly enhance readability.

Reviewer #4 (Remarks to the Author):

The authors have addressed my concerns and I am satisfied with the revised manuscript. Greg Neely

Version 2:

Decision Letter:

Our ref: NG-A64078R1

18th Mar 2025

Dear Dr Adams,

Thank you for submitting your revised manuscript "Targeting the CDS1/2 axis as a therapeutic strategy in uveal melanoma and pan-cancer" (NG-A64078R1). Your revisions were assessed by the team and I'm delighted to say that we'll be happy in principle to publish it in Nature Genetics, pending minor revisions to comply with our editorial and formatting guidelines.

Please note that we have decided to upgrade the manuscript to an Article which will give you an allowance of 8 main display items and 4400 (max) words of main text.

Sincerely,

Safia Danovi, PhD
Senior Editor, Nature Genetics
ORCID: 0009-0007-7822-5479

Dear Dr. Danovi,

We are pleased with the reviewers' comments on our paper which we have exhaustively responded to in below.

With best wishes,

David Adams PhD DSc FMedSci FRCPATH
Senior Group Leader & Head of the Cancer, Ageing and Somatic Mutation Programme
Experimental Cancer Genetics
Wellcome Sanger Institute
Hinxton, Cambs, CB10 1SA
Ph: +44 (0) 1223 834 244

Reviewers' Comments:

Reviewer #1:

Remarks to the Author:

Overall, this work is a quite interesting study that seems like a major advance in the uveal melanoma field by adding in these new models and generating data sets for further characterizations. Impressive they seemed to find a pan-uveal melanoma target in *CDS2* – curious why *CDS1* is so low in these cells/ generally across cell lines – authors don't discuss this much.

We thank reviewer 1 for their constructive comments and insights. At present uveal melanoma (UM) remains an extremely difficult malignancy to treat and unlike cutaneous melanoma where checkpoint inhibitors have been a revolution, advanced uveal melanoma remains incurable. We provide a dataset of unparalleled depth and scale and show how these data can be used to find new therapeutic targets such as *CDS2*. Regarding the point about why *CDS1* is lowly expressed in UM, in Figure 5 we show a strong correlation between *CDS1* methylation and low gene expression.

Major comments:

1. Authors note at the beginning of results that there are few uveal melanoma models but then are able to source about 10 cell line models, some of which also appear to be represented in DepMap. DepMap appears to have about 12. So is it true that “none have been comprehensively characterized” as the authors state?

As the reviewer states there are some cell lines listed as uveal melanoma cell lines in DepMap, but as described below the data on these lines is far from complete. The situation is as follows:

1. There is no CRISPR screen data for 921, MP38, MP41, MP46, OMM1 or OMM2.3. MP38 and OMM2.3 are not in DepMap at all. In fact, there are only 4 lines (MEL202, OMM2.5, MEL270 and MEL285) where DepMap provides CRISPR data (just 4 of the 10 lines we screened). Where whole genome CRISPR screening has been performed on the same lines as we screened (MEL202, OMM2.5, MEL270 and MEL285), a different library was used, and thus our data will allow robust comparisons and replication between datasets.

2. None of the datasets in DepMap include whole genome sequencing or proteomics.
3. None of the uveal lines in DepMap have not been screened with a combinatorial library.
4. There is no integration between available CRISPR data with, for example, expression or other data.

Importantly, we note that several of the lines in DepMap categorised as “uveal” are of unclear provenance and in this context, we highlight the note (point 3) made by Reviewer 1 below. Throughout the paper we do highlight the amazing DepMap resource and have tweaked our narrative based on this comment.

2. Figure 2d – It is mentioned in the text that there are negative correlations between gene expression of CDS2 and paralog and INTS6 and paralog, but only INTS6 data is shown, no data from CDS2. Should at least add to supplement, or maybe show CDS1/2 here because its followed up on and add INTS6 to supplement.

Reviewer 1 makes a good point, and we appreciate this comment – we previously included this comparison in Figure 5A-C to fit with the flow of the manuscript but appreciate that we should amend the narrative to make this clearer and to bring together the *INTS6/6L* and *CDS1/2* analysis, and we have therefore amended the text/figure accordingly.

3. Authors should provide more information for the 10 melanoma cell lines in terms of paralog expression? No RNA-seq or even qPCR based expression testing for the CDS1 and others. Cell lines are referred to being low CDS1 expressers, but there is no data to support this (generated by authors or indicated from DepMap data).

As part of our revised manuscript, we generated and analysed deep transcriptome data for all of the cell line models we used in our study (5 replicates per cell line for all 10 lines. The data is released into the ENA: ERP130186 and TPMs in Supplementary Table 3). This allowed us to perform the *CDS1/CDS2* comparison suggested by Reviewer 1. This can now be found in Figure 3 and below. As detailed in the QC report (available in the Github/Figshare for this project) these data are of extremely high quality. As shown below the expression of *CDS1/2* and other paralog candidates closely mirrors the expression profile seen in uveal melanomas in the TCGA.

C.

Figure R1: Transcriptome analysis of paralog gene pairs in uveal melanoma and uveal melanoma cells lines. We generated deep transcriptome data (average 100Million reads/5 replicates per cell line) to assess the expression of genes across uveal melanoma cell line models. Shown above is the expression of *CDS1/CDS2*, *RIC8A/RIC8B* and *SPTSSA/SPTSSB*, essential paralog gene pairs revealed in our study. As shown, all cell lines show lower expression of *CDS1* compared to *CDS2*, a pattern that parallels what is seen in uveal melanomas from the TCGA.

4. Figure 4d. Authors chose *ITGB5* and *ZEB2* as controls due to some cell lines not requiring these genes, while *CDS2* is required in all cell lines tested. But, extending the assay to 28d shows requirement for these genes. So, are they pan essential in the uveal melanoma lines? Or, were the three lines selected chosen because requirement for these genes was demonstrated in the sgRNA CRISPR screen? It is unclear why these were chosen.

We thank Reviewer 1 for really thinking about our data. With this analysis we aimed to achieve two things 1). Orthogonally validate the *CDS1/2* interaction and 2). Show, using *ITGB5* and *ZEB2*, that essentialities revealed from our CRISPR screens could be recapitulated in the competitive growth assay. Of note, we show in our screens that *ITGB5* is a specific vulnerability in *OMM2.5* and *ZEB2* in *MP41*, a result in complete agreement with the validation studies we present in the initial paper submission. We selected *ZEB2* because we (PMID: 22000016) and others have shown it plays an important role in melanoma development. Similarly, *ITGB5*, and integrins in general (PMID: 34660316), have been implicated in melanoma development. On reflection we appreciate that this might be confusing, so we have removed this analysis from the paper so-as-to focus the reader on the key message i.e. that *CDS1/2* is a robust and validated synthetic lethal interaction. Of note, we have repeated all of the experiments in this figure again and included a safe-targeting

gRNA control (STG). These studies completely replicated the previous experiments as expected.

5. Figure 4D - Further, major activity differences between CDS2 gRNAs is evident in the data represented. Is there any information on gRNA activity – western blot, sanger sequencing to understand this discrepancy better? Was the more potent gRNA based on phenotype moved forward to the remaining assays using CDS2 depletion? Authors should comment on the specificity of the gRNAs used in their screen and in validation for each paralog. It is possible that some gRNAs would target both paralogs.

We have included on-target and off-target scores for all gRNAs used in this study (please see supplementary table 4). Of note, we calculated these values using 3 different algorithms (CRISPRater, MIT and we also generated cfd scores). Importantly, none of the *CDS1* or *CDS2* gRNAs in our library or the *CDS2* gRNAs used for the validation experiments were predicted to cut off-target. Indeed, below we show the gRNA alignment for the *CDS2* gRNA we used for all of our validation experiments against the *CDS1/2* cDNA alignment (**Figure R2**). Importantly, the sequence identity is only 50% making off-target cutting at *CDS1* extremely unlikely.

```

CDS1      TTGTTCAAAGAGAAGAACAACCTTCAGTTCCTCATTGCTACCATAGATTTATATCATTTG 840
CDS2      TGGTCCAGAGAGAAGAGCCTTTGCGGATTCTCAGTAAATACCA CCGGTTCATTTCTTTA 599
          * ** ** ***** * * * * * ***** * ***** * ** ** ** **
CDS1      CCCTCTATCTGGCAGGTTTCTGCATGTTTGTACTGAGTTTGGTGAAGAAACATTATCGTC 900
CDS2      CTCTCTATCTAATAGGATTCTGCATGTTTGTACTGAGTCTGGTCAAGAAGCATTATCGAC 659
          * ***** ** ***** ***** ***** ***** ***** *

```

gRNA relative to the genome: AGTAAAGGAAATGAACCGG

gRNA binding site on the cDNA: CCGTTCATTTCTTTACT

Figure R2: Alignment of the *CDS1/2* DNA sequence and the location of the gRNA used for validation. The red box shows the binding site for the *CDS2* gRNA. Stars indicate identical nucleotides.

To further explore the off-target question, we rigorously tested the *CDS2* gRNA (AGTAAAGGAAATGAACCGG) we used for our validation experiments by transfecting and selecting SW837 cells (which do not require *CDS2* for their fitness) with our *CDS2* gRNA expression vector or a safe targeting control gRNA (STG) – we did this in three replicate experiments. Selection was performed for 7 days to enrich for cells which had editing events at target loci. Cells were subsequently collected for RNA and DNA extraction with these nucleic acids analysed by whole exome and transcriptome sequencing, respectively. Notably, with our *CDS2* gRNA no recurrent off-target disruptive events were found, not at any site across the genome or transcriptome. Indeed, we only saw 2 missense changes (below), which could be cell culture artefacts.

Question: Does the gRNA used generate off-target cutting?

Figure R3: Analysis of on/off-target cutting with the *CDS2* and *STG* gRNAs. The experimental design is shown with the FACS plot showing the transduction efficiency of the lentiviral vectors with the *CDS2* and *STG1* (safe; middle) gRNAs prior to 7 days of selection in puromycin. Uninfected (control; Cas9-expressing SW837_C9) cells were used as a control. The *CDS2* gRNA generated multiple disruptive indel events (top; left panel). Two potential (non-recurrent) missense events (in *CAPNS1* and *DOCK4*) which might be the result of off-target cutting or could be cell culture artefacts were also observed. A single potential off-target *STG1* gRNA event in the *WASHC4* gene was found (right panel). The tile plot (middle) shows a summary of all results. In the tileplot the left 3 samples were from SW837_C9 cells transfected with a *CDS2* gRNA. Right 3 samples were from SW837_C9

cells transfected with the *STG1* gRNA. Of note since the *STG1* gRNA cutting site is not in an exon we did not detect cutting with this gRNA, just the one potential off-target event in *WASHC4*. Importantly, in addition to exome sequencing of the CRISPR *CDS2* transduced cell lines, we also performed transcriptome sequencing of each replicate (bottom). This revealed, as expected, disruption of *CDS2* transcripts which were significantly differentially expressed by DE-Seq, q-val 3.42415949565055e-65. Log2Fc -1.715871. For brevity we have not provided the other 9 differentially expressed genes in these responses but we will include these in the Github/Figshare. These genes did not include *CDS1*. The data accession number is: ERP159012.

Thus, from the above experiment we can definitively confirm that the gRNA against *CDS2* which we used in all our validation experiments is both specific and highly active. Similarly, our control *STG* gRNA cuts specifically.

To directly address the point about protein expression (below) we used our inducible system (which carries the same *CDS2* gRNA as validated above) and disrupted *CDS2*. Equal amounts of protein were loaded, and we very clearly see *CDS2* protein depletion in both of the cell lines we tested (OMM2.5 and MP41).

Figure R4: Validation of the inducible gRNA against *CDS2* as depleting *CDS2* protein. To further validate the depletion of *CDS2* by the potent and specific gRNA we used (AGTAAAGGAAATGAACCGG) we induced its expression with dox in both MP41 and OMM2.5 cells, engineered to stably express Cas9. As shown, *CDS2* disruption resulted in *CDS2* protein depletion. The higher band for *CDS2* seen in MP41 has been reported previously (PMID: 29253589). OMM2.5 cells were analysed as a polyclonal pool while clones (A, D, E) were generated for MP41.

We further extended this analysis with mass-spec.

MP41

OMM2.5

Stars are proteins that are commonly up- or down-regulated upon CDS2 KD, with $p < 0.05$ and $\log_2FC > |0.5|$

Figure R5 – Mass-spec analysis of CDS2 protein before and after induction of a CDS2 gRNA expression with dox in cells stably expressing Cas9. In this experiment we clearly show significant depletion of CDS2 protein in both cell lines. This experiment was performed 3 times

independently. Of note, we were not able to detect *CDS1* protein either before or after activation of the *CDS2* gRNA. This likely reflects the low expression of the gene, which is about 10-fold lower in transcript expression than *CDS2*.

6. Figure 4D –the experiment should also be performed with safe-targeting or non-targeting guides as control

Reviewer 1 is completely correct, and this is what we did when we performed these experiments – we apologise for not being clear and have modified the figure accordingly. The safe-targeting control gRNAs (*STGs*) were used to account for the effects of CRISPR-mediated double strand breaks. *STGs* cut the genome but do not disrupt a gene (i.e. cuts an insert locus) and have no/little effect on the fitness of either OMM2.5 or MP46 cells. We thank Reviewer 2 for making this important point.

7. Figure 5A – it would be helpful to understand the relative levels of both *CDS1* and *CDS2* in each tumor type (currently the figure shows only *CDS1*)

We have performed the analysis suggested by Reviewer 1 and the figure is shown below. As can clearly be seen, low levels of *CDS1* are found across a range of TCGA tumour types including uveal melanoma (UVM).

Figure R5 – Expression of *CDS1*/*CDS2* across TCGA tumour types. TPM+1 values were downloaded from the Xena Browser (<https://xenabrowser.net/>). *CDS1* and *CDS2* gene expression in cancers based upon data generated by the TCGA Research Network and obtained from the UCSC Xena platform. Shown is the data range [mean minus the SE to the mean plus the SE] with points equalling the mean. The X-axis shows the TCGA tumour codes. UVM refers to uveal melanoma. A full list of tumour codes is available here: <https://gdc.cancer.gov/resources-tcga-users/tcga-code-tables/tcga-study-abbreviations>.

We thank Reviewer 1 for making this suggestion and we think this figure enhances the story. In revising the paper, we also used single cell sequencing data to examine at single cell resolution the expression of *CDS1/2* in the uveal melanoma microenvironment. In keeping with the bulk analysis, we see high *CDS2* expression in malignant cells and low expression of *CDS1*.

8. Figure 6A. Again, there is no data to represent the editing activity or specificity of this

gRNA system. It is unclear if the gRNA is definitely cutting *CDS2* or also has activity against *CDS1*

As above we have performed experiments to assess the on/off-target activity of the *CDS2* and *STG* gRNAs we have used. This revealed no evidence of disruptive off-target cut sites and high on-target activity. Further we show using both western blotting and mass-spec that activation of the *CDS2* gRNA (with dox treatment in a cell line constitutively expressing Cas9) potentially disrupts *CDS2* protein levels. Of note, we did not see any cutting at the *CDS1* locus in the abovementioned experiments. We also thought to determine if there was altered *CDS1* mRNA expression after disruption of *CDS2*, albeit in the SW837 line, but we did not observe this following analysis of the abovementioned RNA-seq data (these data/analysis are in the Github/ENA). Finally, we sequenced the residual tumours from the mice carrying uveal melanoma cells with an inducible *CDS2* gRNA generating both exome and transcriptome data of these tumours (detailed below in the responses for Reviewer 3). Critically, we did not see any cutting at *CDS1*. Collectively, these results are in keeping with the on-target/off-target scores for the gRNAs we selected (Supplementary Table 4). Finally the rescue experiments we have performed (discussed below) clearly show that we can rescue cells following *CDS2* disruption with a *CDS1* cDNA. If there was off-target cutting of *CDS1* in the coding region this experiment would fail (and the cDNA would be disrupted). We have included the extensive experiments outlined above in the revised manuscript.

We thank Reviewer 1 for asking these important questions.

9. Figure 6A - Further, DMSO is not the appropriate control for this assay. A safe targeting gRNA should have been used in the same dox-inducible system to generate an appropriate control, both for dox treatment as well as gRNA cutting/ damage to the DNA induced by this. This is also missing from the *in vivo* study, and a limitation of the findings. Further, authors say the colonies are both fewer and smaller, but there is no quantification of this represented, or statistical analyses.

Using our potent and specific *CDS2* and *STG* gRNAs under the control of a dox inducible promoter we re-made the inducible vectors and replicated our experiments with these data shown in Figure 6A of the revised paper (we did this 3 times independently and in technical triplicate). These data are also shown below in Figure R7. As the cells used in the abovementioned experiments are essentially the same as those transplanted into mice and the new *in vitro* experiments completely replicated the initial results we were advised that it would be unethical to repeat the *in vivo* transplants.

OMM2.5-Cas9

	WT Mock		STG2		CDS2	
	DMSO	DOX	DMSO	DOX	DMSO	DOX
R1						
R2						
R3						

Figure R7: *CDS2* disruption is potent and specific when compared to a safe-targeting control (*STG2*) in uveal melanoma cell lines carrying the dox inducible gRNA system. We have performed these experiments in both OMM2.5 and MP41 cells by engineering these cell lines to carry dox inducible vectors expressing a gRNA against *CDS2* (as validated above) or a safe-targeting control gRNA (*STG2*). For each line we have performed the study three times independently (R1-R3) obtaining the same result. i.e. disruption of *CDS2* results in reduced colony numbers (blue box) while the *STG* gRNA (red box) has a minimal/no effect on cell fitness (when compared to DMSO). Top: these data are plotted as surviving fraction. For each experiment 1300 cells/well were plated. The P-value was computed using a two-tailed T-test. Of note, these were polyclonal lines hence the slightly higher background of colonies when compared to the clonal lines shown in the bottom panel, but the data above suggests that there is minimal effect of the safe gRNA (*STG2*) cutting on the fitness of these cells. Colony counting was performed using ImageJ.

10. The in vivo study in Fig 6g would be strengthened by showing that re-expression of CDS2 of CDS1 can rescue this effect.

Figure R8: *CDS1* cDNA expression rescues the cell fitness phenotype associated with *CDS2* loss in uveal melanoma cells. As shown above expression of the *CDS1* cDNA in uveal melanoma cells rescues cell fitness phenotypes associated with loss of *CDS2* in uveal cell lines.

The result shown above is in complete agreement with the other experiments we outline in the paper. We have performed this experiment 3 times independently and in technical triplicate. These data are included in the revised manuscript.

11. Bottom of page 19. Authors refer to this suppressor screen, but don't present any data from it beyond data in a supplement and also don't describe much of the experimental method (timing?, how was coverage maintained *CDS2* depletion is fully lethal at 14d but screen went for 21d?). What if FDR is less stringent? Authors go on to say their lack of findings suggests that there may be few resistance mechanisms to treatment with a *CDS2* inhibitor, but this is a shallow analysis and these genetic based screens do not recapitulate what would happen with an inhibitor (ie direct selection of mutations that bypass inhibitor binding, etc).

We thank Reviewer 1 for making this point – we provide a diagram/schematic outlining how we did this screen (see below and Supplementary Figure 17). Of note, the analysis provided in the supplementary tables allows readers to select different FDRs based on their risk appetite – most screens of this type use a cut-off of FDR <0.05 so in keeping with this we used this threshold also. We are aware that resistance mechanisms identified by CRISPR screening vs mediated following use of a *CDS2* inhibitor (which is currently not available) may be different. For this reason, we were very careful with our language in the paper being

mindful not to overstate the meaning of the results of this experiment. That said, we have amended the text to make clear that resistance to an inhibitor may have a different profile than those revealed by our rescue screen. The reviewer is correct that we did this screen for 21 days. It is likely that a 14 day screen would have been sufficient but we felt that a longer screen would further enrich for resistant clones (if such clones could be revealed by a CRISPR loss-of-function screen) – importantly in the Github/Supplementary Figures we provide all of the screen QC metrics and these screens passed the DepMAP QC cut-offs and are thus of high quality. Of relevance to the discussion around resistance it is important to note that the long-term xenograft experiments we performed allow the cells to evolve as a grafted tumour in mice and thus in this system cells have the potential to acquire all types of resistance mutations/events. As shown in figure 6 we did not observe xenograft re-growth out to 50 days. Reviewer 3 asked specifically about possible resistance mechanisms from the xenograft experiments – we have therefore provided an extensive analysis of residual tumours in the responses to their comments (below). This includes exome and transcriptome sequencing of tumours. Of note, we did not observe any unifying mechanisms of resistance following sequence analysis, but our experiment was limited to 50 days to align with our animal ethics, and we might require longer for resistant clones to emerge.

Figure R9 - Schematic showing the protocol for the genetic suppressor screen. A clonal population of doxycycline-inducible *CDS2* knockout MP41 cells underwent genome-wide CRISPR knockout screening using the MinLibCas9 library (PMID: 33478580). Following puromycin selection, half of the cells received doxycycline for the duration of the screen, to induce *CDS2* disruption, and the remaining half were treated with the equivalent volume of DMSO. After 21 days, the surviving cells in each population were harvested. Genomic DNA was extracted and underwent library preparation for sequencing of the single gRNA region to determine whether any enriched genes were present within the doxycycline-treated cohort with these genes being possible mediators of resistance/rescue associated with *CDS2* loss. Cells marked with dotted red line were sequenced (1, pre-doxycycline samples; 2, DMSO-treated samples, 3, doxycycline-treated samples).

12. Supplementary Figs 6-7 – The MP46 screen is clearly an outlier in terms of quality. How did authors determine whether the screen is of sufficient performance to include results?

We have addressed this question by comparing our screen data to that of ProjectScore (below).

Figure R10 – Analysis of screen quality. To perform this analysis we compared the data from our screens to the screens performed as part of DepMap/Project Score (n= 325 screens) – this was performed essentially as described previously (PMID: 33712601 and PMID: 28993443). Null-normalized mean difference (NNMD). Strictly standardized mean difference (SSMD). See also supplementary Figures.

As noted by Reviewer 1, our screens are of very high quality but the MP46 screens did not perform as well as the others. That said, the results for MP46 passed the Project Score/DepMap screen metric cut-offs and thus can be considered successful screens (these criteria are found in Dempster *et al.*, PMID: 34930405 & PMID: 39030569).

Importantly, in our study we focussed our conclusions on robust hits shared between 6 or more uveal melanoma cell line that we screened. Despite less strong essential/nonessential separation for MP46, many hits shared by other lines were still called from this cell line, including *CDS2*. For a comparison of the NNMD of these screens see below.

Figure R11 - Synthetic lethal hits called at each time point in MP46 (Day 28 and Day 64). Hits shown in blue are called below the mean normalised genetic interaction (GI) score of -0.5 and FDR below 0.01. Area of dot represents how frequently the hit is shared with screens in other cell lines and time points.

The analysis above clearly illustrates that the MP46 data is of sufficient quality to remain part of our story and resource.

13. Methods

o – please provide more information on the WGS including coverage, criteria for variant calling, and exclusion of germline variants

We have provided an exhaustive (MultiQC) report of the genome sequence QC in the Github/Figshare directory. This analysis suggests that our genomic data is of extremely high quality and has a median coverage of >30x which is more than sufficient since we sequenced clonal cell lines. We also provide a summary of exactly how the data was processed to variant calls.

o Please discuss in more detail the source and characteristics of the uveal melanoma cell lines

We provide this information in Supplementary Figure 2 and Supplementary Table 1. The cell lines were purchased from either ATCC or ECACC, with the exception of OMM1, OMM2.3 and OMM2.5 which were obtained from the Liverpool ocular melanoma cell line/biobank resource (<http://www.loorg.org/>). All cell lines were validated by STR profiling. We are not sure which specific characteristics we should add to the paper but provide genomes, proteomes and CRISPR screening data. We now also provide RNA-Seq data on each line.

We have further analysed these lines using both CELLector [PMID: 32437684] (which uses genomic features of cell lines) to align them with TCGA tumours and CellAligner [PMID: 33397959] which uses gene expression profiles. Unequivocally the cell lines we used represent models that carry the key drivers of uveal melanoma and have a gene expression programme that aligns them with cancers of this type.

Minor comments/text issues:

14. It could be helpful to the reader to note the genomic position of CDS1 and CDS2 in the human genome

We thank Reviewer 1 for this comment – we have added these details to the revised manuscript. Of note, *CDS1* is located at 4:84583127-84651334:1 and *CDS2* is located at 20:5126879-5197887:1. This is important because some paralogs are close to each other on the same chromosome which means that the sequence between them is deleted when both genes are targeted by CRISPR at the same time. Importantly, we specifically examined this as part of our library design strategy to ensure the robustness of our screen results.

15. On Page 7, the authors state that their analysis “revealed significantly lower log₂FCs within the double knockout gene pair group” – is this referring to the data in Fig 1D? If so a figure callout can be added for clarity. If not, a supplemental figure could be added to show this.

The Reviewer is correct, we did make this statement. To clarify what we have done we provide the figures below, which have been included in the revised manuscript.

Figure R12 – shown is the distribution of log₂-fold change values for elements of the combinatorial library. This analysis reveals that double KO vectors, on average, have a bigger negative effect on cell fitness than single knockout pairs (where a gene is paired with a safe-targeting control).

This analysis reveals, as expected, an enrichment for genetic interactions with gene co-disruption. $P < 2.2e-16$, Wilcoxon rank sum test.

16. Figure 4A – It is a bit counter intuitive to show the x axis with the more-essential in

melanoma genes on the right with positive LFC. Minor suggestion to flip the sign of the LFC comparison.

We thank Reviewer 1 for this point – we have amended the figure accordingly to flip the axis. We did, however, keep the symbol the same because if we changed the sign it would no longer align with the output of the statistical analysis in Supplementary Table 10.

17. Figure 6D: Labelling the images with what fluor is being used (DAPI, BODIPY) will help readers

We have made this amendment – thanks for improving the clarity of our paper.

Figure R13 – Lipid accumulates in melanoma cells following disruption of *CDS2*. Green denotes lipids stained within the cells, while blue represents the nuclear stain DAPI.

18. Supplementary Fig 11 – the text is not legible

This is likely a file conversion issue. Thank you for making this point. We have increased the font sizes as suggested and made this a vector image.

19. The authors allude to targeting CDS2 but provide no discussion around how targetable this protein is or if other classes of these enzymes have drugs/inhibitors in development or as tool compounds – some discussion of feasibility of this would be nice.

We have added a discussion about druggability to the revised manuscript. It is notable that CDS2 is an enzyme so is likely tractable using a range of approaches. Indeed it contains several druggable pockets.

We thank Reviewer 1 for their extremely helpful comments.

Reviewer #2 & #3:

Remarks to the Author:

In this study, the authors have aimed to identify novel therapeutic targets in uveal melanoma (UVM). They have employed an extensive characterization of 10 UVM cell lines and utilized a small ~570 gene-pair combinatorial CRISPR library in addition to a whole-genome CRISPR library for screening. The manuscript's key observation is the inverse correlation between sensitivity to CDS2 knockout (KO) and CDS1 expression levels in cancer cells, suggesting functional redundancy between these two paralogues. Based on this observation, the authors pose CDS2 as a potential therapeutic target in those cancers, such as UVM. While the study represents an extensive body work, it has several notable caveats.

The observation that CDS1 and CDS2 are potentially functionally redundant paralogue pairs is partially novel, although this has previously been suggested through analysis of existing whole-genome single-gene CRISPR KO screens (PMID: 35417719). However, the central claim of the manuscript that CDS2 is a therapeutic target in UVM is not adequately supported by data.

We thank Reviewer 2 for their comments which we have addressed in full below. The paper that Reviewer 2 cites (PMID: 35417719) is one that we are very familiar with. The authors of this paper applied a computational approach to publicly available CRISPR/RNAi screens to identify associations between the expression (mRNA/protein) of one gene and the essentiality of its paralog(s). Using this approach, they identified 2,040 candidate associations, published in the supplementary materials, of which one was *CDS1/2*. Select associations were then validated with additional experiments, but *CDS1/2* was not among them. We do not think that one association published alongside thousands of others with no further validation detracts from our study, and indeed do not consider the many paralog dependency associations we have previously published in supplementary materials to be validated (PMID: 31652272). In our work we move beyond correlative computational analysis and demonstrate the CDS1/CDS2 synthetic lethality using both single and double gene perturbations. This, combined with our mechanistic studies make our work novel. We do believe that *CDS1/2* is targetable therapeutically, and we have made this message more prominent in the revised manuscript.

Specifically, the authors identify (1) low CDS1 expression in various cancer cell line lineages other than UVM as well as (2) significant enrichment of CDS1 expression in tumors compared to normal tissues (Fig. 5a, Sup. Fig. 15). Together these data suggest broad

toxicity concerns for targeting CDS2 in UVM, which the authors briefly mention but do not explore.

The low expression of *CDS1* in a range of cancers suggests, as we describe in our paper, that targeting CDS2 might be a useful approach beyond uveal melanoma. We see this as an advantage and suggests our findings are of general interest as they highlight a potentially widely applicable approach to controlling tumour growth. In our paper we clearly state, as noted by the reviewer, that some normal tissues express low levels of *CDS1* and suggest that if there isn't a therapeutic window with systemic therapy then potentially locoregional administration of a "CDS2 inhibitor" would be an option – for example into the eye to "mop up" errant tumour cells not removed by radiotherapy.

Reflecting on this point further we would like to direct the reviewer to GTEX and the Chan Zuckerberg (Genestocells resource) which show that there are various tissues that do not express *BRCA1* or *BRCA2* or express it below a TPM of 1. Following the rationale behind the comment above would preclude the use of PARP inhibitors as therapy in patients with *BRCA1/2* mutant cancers, yet these drugs have saved the lives of thousands of people across a range of tumour types. Further, non-replicating cells that don't express *BRCA1/2* are not sensitive to PARP inhibitors because the BRCA-PARP lethality occurs in the context of replication. Similar context specificities will need to be explored for *CDS1/2*. Importantly, one tissue (in both mouse and human) that expresses low levels of *CDS1* is liver. Deletion of *Cds2* in the mouse, using *Alb-Cre* which is expressed in hepatocytes from the earliest stages of liver development, does not result in the death of the animal, rather adult animals on a diet that promotes liver steatosis showed progressive liver disease that can be attenuated with a *PPAR γ* -agonist (PMID: 36546079). In the same way, deletion of *Cds2* in the vasculature, as part of another study (PMID: 32139674), using a tamoxifen inducible Cre driven by a *Cad5* promoter, reduced the growth of xenografted tumours but did not otherwise result in a phenotype. An analogous vasculature specific experiment was also performed by another group with no overt phenotypes reported (PMID: 31501519). Finally, it should also be noted that we recently showed that under conditions of sustained GPCR-stimulation of PLC, CDS2-deficient mouse macrophages were unable to maintain enhanced rates of PI synthesis via the 'PI cycle', leading to a substantial loss of PI (PMID: 39312194). This is in keeping with the idea that there exists a difference between normal and GPCR activated tissues such as uveal melanoma. It therefore does not necessarily hold that a CDS2 inhibitor administered to adult humans would be associated with organ limiting toxicity, but clearly this is a question that will require many years of investigation before a CDS2 inhibitor is deployed clinically.

The authors' hypothesis that mutations contributing to increased GPCR signaling could lead to higher CDS2 dependency in UVM warrants extensive validation. For instance, sensitivity comparisons to CDS2 KO in other low CDS1-expressing cell lines from different lineages (i.e. blood and liver), and the impact of altering GPCR and/or PLC signaling, should be demonstrated.

In answering this question, we decided that rather than testing specific cell lines we would leverage the entire DepMap and a recently published analysis of GPCR pathway associated expression profiles (PMID: 38723607).

We calculated the enrichment of reactome pathways on z-score transformed cancer cell line RNAseq data. We then correlated the normalized enrichment scores with the *CDS2* Chronos score across cell lines.

Figure R14 – *CDS2* essentiality across DepMap correlates with GPCR signalling. We found that GPCR pathways, involving receptor-ligand pairs in either Signal Transduction or Metabolism processes (Arora *et al.*, PMID: 38723607), are characterized by a significantly different distribution of correlation values (CDF) with respect to other pathways (Kolmogorov-Smirnoff P-value=2.2 E-16). This suggests activated GPCR signalling correlates with an increase in *CDS2* essentiality.

We found that multiple pathways involved in GPCR signal transduction are more frequent at negative correlation values, suggesting that higher *CDS2* essentiality (given by more negative Chronos score) is more common in those cancer cell lines displaying up-regulation of GPCR signal transduction pathways (given by higher normalized enrichment scores). Pathway instances such as “G_PROTEIN_BETA_GAMMA_SIGNALLING” or “G_BETA_GAMMA_SIGNALLING_THROUGH_PI3KGAMMA” are in absolute the most negatively correlated pathways. Among other negatively correlated GPCR pathways we also found instances associated to Gq signaling, such as “G_BETA_GAMMA_SIGNALLING_THROUGH_PLC_BETA” and “G_ALPHA_Q_SIGNALLING_EVENTS”.

Figure R15 – *CDS2* Chronos/GPCR pathway correlations. The correlation shown is a Spearman's coefficient. As noted above, we found that multiple pathways involved in GPCR signal transduction are more frequent at negative correlation values, suggesting that higher *CDS2* essentiality (given by more negative Chronos score) is more common in those cancer cell lines displaying up-regulation of GPCR signal transduction pathways (given by higher normalized enrichment scores).

Finally, it should also be noted that we recently showed that that under conditions of sustained GPCR-stimulation of PLC, *CDS2*-deficient mouse macrophages were unable to maintain enhanced rates of PI synthesis via the 'PI cycle', leading to a substantial loss of PI (PMID: 39312194). This is in keeping with the idea that there exists a difference between normal and GPCR activated tissues such as uveal melanoma.

In summary, *CDS2* essentiality is higher in cell lines with activated GPCR signalling.

The manuscript also lacks important rescue experiments and several detailed methodological descriptions:

As shown, we have performed these rescue experiments, and the figure shown below is representative of three biological replicates performed in triplicate.

Figure R8: *CDS1* cDNA expression rescue the cell fitness phenotype associated with *CDS2* loss in uveal melanoma cells.

- Functional redundancy of *CDS1/2* should be shown through rescue experiments, such as using exogenous *CDS1* expression to compensate for *CDS2*-KO in UVM cells.

We have performed the rescue experiments suggested. These experiments clearly show that ectopic expression of *CDS1* rescues *CDS2* loss in keeping with the hypothesis we described in our manuscript (above). Of note, overexpression of *CDS1* by itself did appear to have an effect on cell fitness, but nonetheless genetic rescue was clear.

- The mechanism of *CDS2*-KO mediated cell death is unclear. Rescue experiments involving supplementation of phosphoinositides or depletion of intracellular triacylglycerol accumulation could provide insights.

We have performed the exact experiment suggested supplementing cells with phosphoinositides (PI or PA) and found that uveal melanoma cells in which *CDS2* has been disrupted can be rescued by treatment, but that they are hypersensitive to PI. As shown below we could reproducibly rescue MP41 cells with PI but OMM2.5 cells died, even with low dose treatment. Thus, we can confirm that PI does rescue uveal melanoma cells following *CDS2* loss, but that this phenotype is context dependent. Importantly in the paper co-submitted with this paper (from Daniel Peepers' group at the NKI), PI is shown to rescue lung and other cancer cells following *CDS2* loss. These cells are not as acutely sensible to PI as our uveal lines.

*PI was complexed with lipid-free BSA and added at 0.25 mM.

In this experiment cells were transduced with a *CDS2* gRNA lentivirus and another virus expressing mCherry. The dynamics of the population was measured over time by flow cytometry with the data above collected on day 10. Viability was rescued with PI in MP41 cells but not in OMM2.5 cells because of massive cell death in response to PI.

To address the question about pathways/ways of cells death - we used our proteomic data to ask what processes are altered following loss of *CDS1* (see below) in uveal melanoma lines. Analysis in this way revealed four pathways as differentially regulated namely; ubiquitin mediated proteolysis, DNA replication, cholesterol regulation and lysosomes. We also show an increase in apoptosis in Figure 6 suggesting that loss of *CDS1/2* promotes this mode of cell death.

Figure R16 – Volcano plots illustrating differentially expressed proteins between DOX-induced knockout of *CDS2* and the corresponding parental cell lines. The differential expression was assessed using a paired two-sample *t*-test. Proteins associated with selected enriched Gene Ontology (GO) terms are highlighted on the plots. Data visualization was performed using the Plotly package in Python.

- The CRISPR screen quality assessment is insufficient (Sup. Figs 8-9). Separation metrics of pan-essential and non-essential genes should be shown (e.g., SSMD and NNMD scores). In addition, benchmarking of these screens against previous published screens would add context to the screening performance.

As detailed below in Figures R17 & R18 we have performed the analyses suggested by Reviewer 3.

Figure R17 – Combinatorial screens: Displays NNMDs (Null normalised mean differences) derived from guide-level LFCs of essentials and nonessentials in each replicate.

Figure R18 – Combinatorial screens: This figure displays NNMDs derived from mean gene-level LFCs of essential and non-essential genes in each cell line, compared to data from 325 Project Score screened lines (there are more negative values than guide-level replicate-level NNMDs above due to less variation). Of note, we re-computing NNMDs using the subset of essential and nonessential control genes in the combinatorial library to draw this figure.

Thus, the combinatorial screens in the uveal melanoma cell lines are of high quality and comparable to those in Project Score and DepMap. In the paper we also provide screen metrics for the single whole genome gRNA screens which we find to be as good as those from Project Score/DepMap.

- Details on sgRNA activities from the hU6 and mU6 promoters in the combinatorial screens are missing (showing whether both sides were balanced in their ability to achieve the desired phenotypic effect).

As shown in Figure 1C, to account for potential promoter bias, we positioned an equal number of guides for every gene behind each promoter, with 4 sgRNAs in either orientation, and a total of 8 different sgRNAs per gene.

```

hU6 Tracr: gtttcagagctatgctggaaactgcatagcaagttgaaataaggcta
mU6 Tracr: gttttagagcta      gaa      atagcaagttaaaataaggcta

```

Figure R20 – Design/sequence of the tracrRNAs used in this study. We have drawn the figure above and included it in supplementary Figure 5. Of note, we have also provided the full sequence of the CRISPR vector, should investigators wish to examine it (see Figshare).

- It is unclear how the 72 gene hits in UVM cells were defined; if a $\text{Log}_2\text{FC} > 1.5$ and $\text{P-adj} < 0.01$ was applied to define this, this should be clearly stated in the main text as well as Fig. 4a, and Fig. 4a should be corrected to reflect this cutoff.

We have added these details to the figure legend to align with the details provided in the methods. Reviewer 2 is absolutely correct on how these genes were identified. Of note, in the revised manuscript we elected to use a slightly more stringent cut-off of $\text{Log}_2\text{FC} > 1.83$. Figure 4a has been amended accordingly.

In conclusion, the manuscript contributes valuable data towards understanding the genetic dependencies in UVM. The data presented fall short of establishing the necessary specificity and safety profile required for a viable therapeutic strategy in targeting CDS2, particularly given the potential for broad toxicity. To advance this hypothesis, comprehensive validation through additional well-designed experiments and more rigorous data analysis is crucial. The inclusion of these elements would greatly enhance the robustness and credibility of the proposed therapeutic implications.

We thank Reviewer 2 for these thoughtful comments – it should be noted that this paper was co-submitted with a study from Daniel Peepers' lab that describes exactly the same mechanism and also the *CDS1/CDS2* essentiality/synthetic lethal interaction we describe in our manuscript. As above, the evidence from mouse studies suggests that loss of *Cds2* does not disrupt homeostasis, at least in the liver and vascular, and thus we remain positive that a CDS2 inhibitor could be used as a systemic therapy and if not delivered in a localised manner.

Reviewer #4:

Remarks to the Author:

Metastatic uveal melanoma can be devastating to patients and new therapies are desperately needed. In this study the authors use a functional genomic approach to investigate essential genes and synthetic lethal (SL) interactions across a panel of uveal melanoma cell lines. The authors focus on one SL interaction, CDS1/2, and establish functional relevance in vitro and in vivo. The design and screening approaches are well described and effectively applied in this system, and appear to reveal new molecular insight into molecular mechanisms controlling uveal melanoma. Overall the study is original and effectively uses powerful and new functional genomics techniques. CDS2 is convincingly shown to be required for uveal melanoma, given the importance of this pathway, its not clear if it is also essential for many other cancers, or healthy cells, but surely with this knowledge strategies can be developed to generate novel chemotherapies with some level of selectivity.

I have included major and minor comments below that may help improve this manuscript.

We thank Reviewer 4 for these extremely useful comments which we have responded to in full below. The first author of this paper is a full-time uveal melanoma oncologist/physician and we (as a team) have seen first-hand just how devastating a diagnosis of this disease can be.

Major comments

1. Concerning the uveal melanoma cell lines used in this study, it's not clear how removed they are from primary patient material, and from this what changes reflect bona fide cancer signatures vs changes that have accumulated from in vitro propagation. How do these genomic and protein signatures compare to those from primary tumour material? Use of fresh patient derived cell lines that recapitulate synthetic lethal results, at least for *CDS1/2*, would strengthen the confidence in this approach.

We have used two methods to examine the alignment of the cell models we used to uveal melanomas collected and analysed from patients. These methods, CellAligner and CELLector, use RNA expression and genomic features/driver mutations, respectively. Of note, the models we used are clearly derived from uveal melanomas showing genomic features and expression profiles that align with the TCGA uveal melanoma dataset. Further, some of the cell lines we use are PDX-derived lines (for example MP46). Remarkably, it is currently not possible to propagate primary uveal melanomas (as for example organoids), largely because these cultures tend to become overrun with contaminating fibroblasts. Of note, to perform the CellAligner analysis we generated deep transcriptome data of each cell line, which we provide with the revised version of the manuscript (The data accession number is: ERP130186).

2. An assumption with this work is that the screening results will specifically inform on the biology of uveal melanoma. However by focusing on pairs that were shared among at least 6/10 cell lines screened, this could actually favour general (vs. uveal melanoma) SL combinations. To confirm there is a cancer type molecular specificity to this approach, one could also screen non-uveal melanoma cell lines and show there is a lack of correlation for SL hits.

Our original focus, and the premise by which we started these experiments, was to define new targets for uveal melanoma – that we also show (using DepMap genome-wide CRISPR data in cell lines derived from over 30 tumour types; Figure 5) that the genetic interaction between *CDS1/2* is also relevant across a range of cancers is extremely interesting and rather than complicating our story, we see it as a compelling outcome. Of note, a paper from Daniel Peepers' group at the NKI was co-submitted with our study and shows that *CDS2* is essential in other tumour types, such as lung, in which *CDS1* is expressed at a low level and these results align perfectly with those described in our paper.

Alternatively, if sufficient data exists, comparing the current SL hits (or imputed SL hits found in single KO screens + RNA seq etc) with previous screens of cells from different cancer origins could also be informative, with either general or cancer specific SL interactions being interesting. To some extent this type of comparison is included in fig 3 and 5, but for slightly different purposes.

This is a very good point, and we have performed this analysis as suggested – this question/comment suggested to us that we need to make the fact that the *CDS1/2* genetic interaction is operative in a range of tumour types more prominent in the paper. We have amended the narrative accordingly.

3. The inclusion of single gene KO screening is useful, but in some way disrupts the study flow. This is primarily b/c *CDS2* on its own is lethal, since *CDS1* expression is reduced in these cells and presumably there is a SL-like interaction. This complicates the study and makes the SL screen component seem less significant, since really screening single KO libraries and comparing essential genes with cell expression profiles could reveal the same SL interactions at the genome level, and from a cancer therapeutic perspective a single target may still be sufficient depending on gene expression.

We appreciate this comment and on reflection we have modified our manuscript to improve the flow. We were trying to illustrate that by looking at single CRISPR screening data alone it is not possible to find the interaction between *CDS1/2* because *CDS2* by itself appears essential. Even if expression data and CRISPR data were available, and we deduced the *CDS1/2* interaction by analysing these data we feel that this does not obviate the requirement to validate and contextualise CRISPR screen results. Of note, previous studies have not described epistasis between *CDS1/2*, as we did, likely missing it for the reasons detailed here.

4. While *CDS2* essentiality in a low *CDS1* expressing cell is presumably via a *CDS1/2* SL interaction, other more indirect mechanisms are also possible. These competing indirect mechanisms could be ruled out by inducing *CDS1* ectopic expression in *CDS2* KO lethal lines. i.e. can one rescue lethality with dox inducible *CDS1*?

This is an important point and in response to this and the same point made by the other Reviewers, we have performed genetic rescue experiments. As show in Figure 5 (and in Figure R8, above) we found that introduction of the *CDS1* cDNA rescues the cellular lethality associated with *CDS2* loss. We thank Dr. Heely for making this important suggestion leading to this elegant experiment.

5. The genetic suppressor screen is interesting, however loss of function is only one way that cells could change to circumvent *CDS2* dependence, and gain of function changes could also help cells escape from loss of *CDS2*, so its not clear if one can claim targeting *CDS2* therapeutically would not lead to cells that may become resistant through additional changes.

Reviewer 1 also made this important point which we appreciate. In addition to the CRISPR “rescue” screen we aged mice in which *CDS2* has been disrupted and we do this over 50 days without seeing any escape/resistance. *In vivo* experiments performed in this way should capture all/most possible modes of tumour evolution including gain- and loss- of function mutations. We have worked on the narrative of this section of the paper to make this message clearer. Below we describe an exhaustive analysis of tumours from the mice using exome and transcriptome sequencing with the aim being to identify possible resistance mechanisms.

6. From my experience, 3 biological replicates performed in a single experiment (e.g. Fig 6b) does not always lead to definitive results, and to be certain, reproduction of results through multiple independent experiments performed on different days is more robust.

This was 3 independent experiments/biological replicates that were analysed on the same day, akin to sequencing biological replicates together to reduce possible batch effects. We apologise for the confusion and Prof. Heely is completely right that this was not clear. In revising the experiments for the paper, we also did these experiments (again) completely independently and obtained the same result. These data are provided in the revised manuscript.

7. The *in vivo* experiments are compelling, can you evaluate KO efficiency or gene expression in the surviving tumors over time? Presumably these cell have intact *CDS2*? Or have they upregulated *CDS1* or other factors? Optimally these data could also be confirmed with xenografts if such systems are available.

We have addressed this question in several ways. Firstly, we exome sequenced tumours collected from mice with and without dox treatment (i.e. *CDS2* disruption) to call somatic variants and we also subjected these tumours to RNA-Seq. These tumours were collected at the terminal timepoint. To make the analysis possible we developed a pipeline that allowed

us to separate reads from the human uveal melanoma cell lines and those from the mouse stroma. This pipeline is shown below.

Figure R21 – Analysis of xenografted tumours collected from the mouse experiments shown in Figure 6. Tumours were collected when they reached the ethical limit or at the end of the experiment.

Since we could not detect visible tumours when the dox-treated mice were sacrificed (i.e. tumours in which *CDS2* was disrupted) we first used human specific primers to see if we could detect any residual human/uveal melanoma cells in DNA extracted from tissues collected from the injection site of the xenografted mice. Analysis in this way revealed that there were human uveal melanoma cell remaining in the dox treated mice i.e. those mice exposed to dox so that the implanted human uveal melanoma cells would be disrupted at the *CDS2* locus. As shown below the dox-treated tumours had more stroma than untreated tumours.

Figure R22 – Detection of xenografted cells and stromal context of uveal melanoma cells grafted into mice. The PCR shown (left) was performed to detect the OMM2.5 cells at the graft site. As shown, the PCR primers amplified a band in most samples suggesting that even though no tumours could be palpated in the dox-treated animals, residual tumour cells remained. We

transcriptome sequenced these tumours revealing (on average) more mouse stroma following *CDS2* loss (right). The primers used for the PCR amplify a 823bp band flanking the *CDS2* target site: F=GGAATGCTACAAAAGGGGC; R=TCCCAATTCCTACCTGACAG.

We used the extracted RNA from the tumours shown in Figure R22 for transcriptome sequencing to explore Prof. Neely's question about resistance. Firstly, we did not see lower levels of *CDS2* expression when we analysed the RNA-Seq data from QC passed tumours (below), which prompted us to look for *CDS2* cutting from the gRNA at the target locus (Figure R23). In one tumour we observed in-frame cutting where 6 nucleotides were deleted, which likely explains why these cells survived. In other cases, we saw indels at the target locus but also intact reads with mutations in them that might be wildtype but no longer a substrate for the *CDS2* gRNA. Alleles such as these likely explain why we see no differential expression of *CDS2* because these editing outcomes at the *CDS2* locus mean that the gene is not disrupted. Finally, we performed differential expression analysis comparing the treated vs untreated tumours. In this way we identified 125 differentially expressed protein coding genes. Of note, *CDS1* was not differentially expressed and GSEA analysis of these genes did not reveal any significantly altered pathways. Collectively, the analysis outlined above suggests that, at least in the time course of this experiment, that surviving uveal melanoma cells are those in which *CDS2* was not completely disrupted, but we are mindful that a larger cohort of tumours will need to be analysed to explore this question further.

Figure R23 – Exploring the mechanisms of resistance in surviving uveal melanoma cells in xenograft models. Differential expression analysis of wildtype vs dox-treated *CDS2* disrupted uveal xenografted OMM2.5 cells reveals no differential expression of *CDS2* (top). Analysis of *CDS2* cut sites revealed non-disruptive edits likely explaining why these cells have survived *in vivo* (middle). Differential expression analysis of wildtype vs dox-treated *CDS2* disrupted uveal xenografted OMM2.5 cells reveals 125 protein coding genes as differentially expressed but GSEA analysis of these genes did not reveal any common pathways. Of note, *CDS1* was not differentially expressed. These data and analysis are available in the Github for the project.

Minor comments

Fig 1a. Its not clear in the text where the 1061 single genes came from. Presumably these are guides targeting only one of each paralog or synthetic lethal gene pairs? This could be clarified in the text.

We thank Reviewer 4 for this comment. We have clarified this point in the revised manuscript – basically for each gene in a candidate SL pair we included vectors in the library where there was a non-targeting control paired with a gRNA targeting that gene. During the analysis this allows us to compute the single gene effect and thus identify gene-gene synergy (i.e. the combinatorial effect of gene disruption). To simplify the narrative, we have re-drawn figure 1 and improved the figure legend.

Fig 1c. If there are 8 individual guides per gene and two positions, shouldn't this be 64 combinations? It seems that only 4 of the 8 are selected for each position, but the rationale for this wasn't completely clear and could be further clarified for the general readership?

Figure 1c. 8 individual single gRNAs were selected for each gene, allowing 4 to be placed under each promoter, resulting in a total of 32 unique gRNA pairings for every gene pair. For each gene pair, an additional 16 combinations were included, from pairing each single gRNA with an STG to assess individual guides, and single gene knockout effects. This results in each gene pair having a total of 48 unique gRNA combinations.

We thank Prof. Neely for his very constructive comments.

Dear Dr. Danovi,

We are delighted that Reviewers 1 & 4 are fully satisfied with the revisions we have made to our paper and that we have also satisfied the initial set of comments from Reviewers 2/3. We thank Reviewers 2/3 for their additional comments which we have addressed in full below.

With best wishes,

David Adams PhD DSc FMedSci FRCPATH
Senior Group Leader & Head of the Cancer, Ageing and Somatic Mutation Programme
Experimental Cancer Genetics
Wellcome Sanger Institute
Hinxton, Cambs, CB10 1SA
Ph: +44 (0) 1223 834 244

Reviewers' Comments:

REVIEWER #1

The authors have thoughtfully addressed my prior concerns. In particular the off-target analysis, rescue experiment, and additional methodological details are appreciated. Congratulations!

It is so pleasing for an author to read a comment like this! We are delighted that Reviewer 1 is happy with our revisions, and we thank them for their kind words.

REVIEWER #2,#3

In this study, the authors aim to identify novel therapeutic targets in uveal melanoma (UVM). The manuscript combines data from 10 UVM cell lines, including a small ~570 gene-pair combinatorial CRISPR library, whole-genome CRISPR screens, RNA-seq, and proteomics. The study highlights the inverse correlation between CDS2 knockout sensitivity and CDS1 expression, suggesting functional redundancy between these paralogues and proposing CDS2 as a therapeutic target in cancers such as UVM.

We thank REVIEWERS #2,#3 for their thoughtful comments which we have responded to below. As we detail in the manuscript, we screened all available uveal melanoma cell lines with a focused and rationally designed library.

This manuscript has been co-submitted with a complementary study by Arnoldus *et al* (Daniel Peeper lab), which leverages DepMap and TCGA data with experimental validation to propose CDS2 as a therapeutic target in mesenchymal-like cancers. Both manuscripts were reviewed simultaneously.

It is fortuitous that Daniel Peepers' group identified the CDS1/2 interaction independently – we are pleased that we could co-submit both papers simultaneously as the two studies cross validate each other.

The authors have made substantial revisions and incorporated new experimental data in response to reviewer feedback, effectively addressing many initial concerns. However, several critical questions remain unresolved. Below, a detailed analysis is provided along

with key areas of the manuscript requiring amendments or additional data for clarification and improvement.

We are pleased that REVIEWERS #2,#3 are satisfied by the significant revisions we made to the original submission. We have addressed their final points below.

1. Proposed novelty and contribution to the field

The manuscript provides valuable data on UVM, including combinatorial and single-gene CRISPR KO dependency data, RNA-seq, and proteomics datasets, while also characterizing the CDS1/CDS2 synthetic lethal interaction (SLI). However, the usefulness of these datasets in identifying the CDS1/CDS2 SLI is limited, as this interaction could also be deduced from existing single-gene CRISPR screens, as demonstrated by the Arnoldus et al co-submitted paper. Moreover, several prior studies, including PMID: 35417719, PMID: 35194081, PMID: 34469736, and the authors' own prior work (PMID: 33637726), have reported or inferred this SLI. While the authors argue that these findings are obscure, explicit acknowledgment of these prior works in the manuscript is fundamentally essential.

In performing these studies, we had two aims. 1). To generate a resource of synthetic lethal interactions and genetic dependencies for uveal melanoma that the community could mine and 2). To identify SL interactions that we could explore functionally. Had we not generated the data we describe in the paper we would not have homed in on the *CDS1/2* interaction among a sea of other candidates. In fact we pointed Arnoldus et al., at *CDS1/2* from a long list, which they then focused on in their paper. Indeed, despite intense interest in synthetic lethal interactions and wide access to DepMap, we are the first to credential *CDS1/2* as an SL interaction. Of note, while we focused on the *CDS1/2* interaction there are many other interactions that we report (most not evident in DepMap) that can be examined and explored functionally using our data. As suggested, we have made the abovementioned studies more visible in the revised manuscript.

2. CDS2 as a therapeutic target in UVM

We appreciate the additional analysis supporting a therapeutic window for CDS2 in UVM, specifically the correlation between GPCR signaling and CDS2 essentiality, and the inclusion of references showing that *Cds2* deletion in mice is well-tolerated in liver and blood. However, the manuscript does not adequately address how their findings relate to the co-submitted paper by Arnoldus et al. The authors should clarify whether their data suggest that UVM represents a particularly favorable or distinct context for CDS2 dependency, potentially due to its biology, or if CDS2 dependency in UVM aligns with the broader mesenchymal cancer subtype described by Arnoldus et al. Establishing this link or distinction is critical for readers to fully contextualize CDS2 as a therapeutic target in UVM and other cancers.

We thank REVIEWERS #2,#3 for this thoughtful comment. We think that UVM is a particularly amenable tumour type to treatment with a (potential) CDS2 inhibitor. We think, as we suggest in the paper, that the activation of GPCR signalling in UVM makes this tumour hypersensitive to CDS2 inhibition and the new analysis we provided in supplementary Figure 17 and in Figure 5 support this view. In our revised manuscript we have added a few sentences to link the excellent work from Arnoldus et al. to our study with a focus on the broader implications.

3. Combinatorial CRISPR screen quality assessment

The authors have included extensive additional metrics, such as NNMD, SSMD, ROC curves, and Cas9 activity scores, to support the quality of their screens. However, some critical caveats remain unaddressed:

- In Sup. Fig. 5d, sgRNA activity under hU6 and mU6 promoters is clearly not balanced. As

shown in Fig. 1c, each unique sgRNA is expressed by only one promoter, either hU6 or mU6, but never both. This raises questions about whether the authors could distinguish between effects arising from true sgRNA activity/quality and those influenced by promoter positioning. The authors should explain how they accounted for this potential confounding factor in their analysis.

REVIEWERS #2,#3 are correct that hU6 is stronger than the mU6 promoter. Although the exact same gRNA is not, as they say, in both positions in the vector we do include an equal number of the gRNAs against each gene behind each promoter. This balanced design was chosen to mitigate for the possibility of differences in promoter strength impacting the results at library scale, avoiding instances of one gene being subject to more efficient editing as a consequence of guides only being placed behind the stronger promoter.

In terms of accounting for the differences in promoter strength in the analysis, in an early version of the analysis using this library we separated out guides targeting single genes in either orientation and compared this to the combined data from both orientations. We found there was no impact on the key results so decided to take the combined approach in the analysis for this paper. Of note, for each gene we examined in this study we have 8 gRNAs (4 behind each promoter) which is higher coverage than virtually any other combinatorial library. We have made these points clearer in the revised manuscript.

- Also in in Sup. Fig. 5d, a group of non-essential genes (and essential genes) scores strongly in both promoter positions (LFC -5.0 to -7.5). Could this result from unintended activity, such as off-target effects or an STG (Safe Targeting sgRNA) that is not truly safe targeting? The authors should describe how such artifacts were identified and mitigated (e.g. how sgRNAs paired with this “bad” STG were handled in downstream analyses). These points are currently absent in the text and need to be addressed.

Following further analysis, we can report that in Sup. Fig. 5d nonessential controls with negative unscaled LFCs (< -2.5 in both promoter orientations) are accounted for by being data points for either day 49 of the screens in cell line Mel285 or day 64 of the screens in the cell line MP46. These points show reduced cell survival because the cell lines did not survive well at these later time points. Instances of these nonessential genes displaying an essential phenotype are a consequence of the cell lines' poor overall survival, rather than a feature of those guides specifically. While Sup Fig 5d shows unscaled logFCs, in the core analysis to identify synthetic lethal hits LogFCs were scaled so the median logFC of non-essential genes was 0, and known essentials was -1 to account for variable growth rates and cell survival across lines. As discussed in our previous set of reviewer responses, while we are aware these cell lines did not perform as well as the others the results still passed the Project Score screen metric cut-offs and thus can be considered successful screens (Supplementary Figure 8). Of note, in our study we focussed our conclusions on the earlier time point of Day 28 for the dual guide screens, and screen hits shared between 6 or more uveal melanoma cell lines. Collectively, this addresses the concern of off target effects or the possibility of a ‘safe targeting’ guide targeting an important region. To comprehensively illustrate what we have outlined above we have included a similar plot (below) highlighting the points from Mel285 Day 49 and MP46 Day 64. There are no nonessential genes in other cell line time points with such negative lfc (see yellow dots).

Plot displays the mean unscaled LFCs per gene in a particular orientation for each sample. Cell lines Mel285 at Day 49 and MP46 at Day 64 are highlighted to illustrate extreme negative LFCs for 'nonessential' genes are a consequence of poor cell line survival rather than other factors (such as off-target cutting).

4. Low CDS1 expression in UVM and other Cancers

We appreciate the inclusion of the CDS1 rescue experiment (Fig. 5f), which demonstrates that ectopic CDS1 expression can partially rescue lethality associated with CDS2 loss. However, the observed toxicity of CDS1 expression in OMM2.5 cells raises questions about its role in tumor biology. The authors should discuss why certain cancers might downregulate CDS1, and provide a hypothesis for why CDS1 expression might be detrimental to these cells.

REVIEWER #2,#3 raises an very interesting question, and one we have been pondering ourselves. Firstly, it is possible that ectopically over expressing CDS1 decreases cell fitness because we are likely expressing it at an abundance above physiological levels i.e. there is a "just-right" level of expression. The interesting question for us is why do so many cancers (including uveal melanoma and others) downregulate CDS1, and why do they want to keep its expression under such tight control? We posit that it may be because it is involved in their development, potentially as a tumour suppressor or that the level of expression is controlled to sustain tissue homeostasis. In keeping with this gene expression of *CDS1* has a narrower expression variance range across normal tissues as shown in GTEx (See Sup. Fig19), in comparison to *CDS2*. In our view the tight regulation of CDS1 denotes its physiological importance, as explored by us, and also in the Arnoldus *et al.* paper. We have added a note on this in the revised manuscript.

Minor Comments and Corrections

Sup. Fig. 5C : There is an inconsistency in the tracrRNA sequence reported in Sup. Fig. 5C (5'-gtttcagagctatgctggaacagcatagcaagttgaaataaggcta-3') versus Sup. Table 16 and Fig.

R20 (5'-gtttcagagctatgctggaaactgcatagcaagttgaaataaggcta-3'). Please confirm and update the correct sequence in all relevant figures, tables, and methods.

This shows amazing attention to detail. The sequences in the figures and tables are correct. There are two tracer sequences in the vector – in the supplementary table we had only shown one but for the inducible CDS2 vector. We have made this clearer.

Line 140: The term “pgRNA” is introduced without definition. Please define this term in the text.

We thank REVIEWERS #2,#3 for noticing this and have amended the text accordingly.

Figure 5a: The figure and its legends are too small and difficult to read. Enlarging and reformatting for clarity would greatly enhance readability.

We thank REVIEWERS #2,#3 for noticing this and have amended the text/figure accordingly.

We thank REVIEWERS #2,#3 for their extremely careful review and we are grateful for their efforts to enhance our manuscript.

REVIEWER #4 (signed report)

The authors have addressed my concerns and I am satisfied with the revised manuscript.
Greg Neely

We thank Professor Neely for his excellent comments and for his contribution to improving our manuscript.